# FlashSinkhorn: IO-Aware Entropic Optimal Transport on GPU

**Felix X.-F. Ye** [1]  **Xingjie Li** [2]  **An Yu** [3]  **Ming-Ching Chang** [3]  **Linsong Chu** [4]  **Davis Wertheimer** [4]

## Abstract

Entropic optimal transport (EOT) via Sinkhorn iterations is widely used in modern machine learning, yet GPU solvers remain inefficient at scale. Tensorized implementations suffer quadratic HBM traffic from dense $n \times m$ interactions, while existing online backends avoid storing dense matrices but still rely on generic tiled map-reduce reduction kernels with limited fusion. We present **FlashSinkhorn**, an IO-aware EOT solver for squared Euclidean cost that rewrites stabilized log-domain Sinkhorn updates as row-wise LogSumExp reductions of biased dot-product scores, the same normalization as transformer attention. This enables FlashAttention-style fusion and tiling: fused Triton kernels stream tiles through on-chip SRAM and update dual potentials in a single pass, substantially reducing HBM IO per iteration while retaining linear-memory operations. We further provide streaming kernels for transport application, enabling scalable first- and second-order optimization. On A100 GPUs, FlashSinkhorn achieves up to $32\times$ forward-pass and $161\times$ end-to-end speedups over state-of-the-art online baselines on point-cloud OT, improves scalability on OT-based downstream tasks. For reproducibility, we release an open-source implementation at `https://github.com/ot-triton-lab/flash-sinkhorn`.

---

[1]Department of Mathematics & Statistics, University at Albany, Albany, NY, USA [2]Department of Mathematics and Statistics, University of North Carolina at Charlotte, Charlotte, NC, USA [3]Department of Computer Science, University at Albany, Albany, NY, USA [4]IBM T. J. Watson Research Center, Yorktown Heights, NY, USA. Correspondence to: Felix X.-F. Ye <xye2@albany.edu>.

*Proceedings of the 43rd International Conference on Machine Learning*, Seoul, South Korea. PMLR 306, 2026. Copyright 2026 by the author(s).

## 1. Introduction

Optimal transport (OT) is a principled way to compare probability measures and has become a standard tool across modern machine learning, from distributional objectives in generative modeling (Arjovsky et al., 2017) to alignment and matching problems (Solomon et al., 2015; Schiebinger et al., 2019; Bunne et al., 2024). Entropic regularization (Cuturi, 2013) makes OT differentiable and computationally tractable via Sinkhorn iterations, and GPU implementations of Sinkhorn-based losses are now widely used (Feydy et al., 2019; Cuturi et al., 2022). However, scalability remains a persistent challenge in practice: OT is often considered too expensive to use extensively at evaluation time. For instance, Kangin & Angelov (2024) omit confidence intervals for their Sinkhorn setting due to computational cost, and Kokot & Luedtke (2025) note that repeatedly evaluating Sinkhorn divergence at MNIST scale ($n \approx 70000$) can be computationally prohibitive. These observations point to a systems bottleneck. Tensorized GPU Sinkhorn solvers are dominated by quadratic HBM traffic: materializing and repeatedly traversing the $n \times m$ interaction structure across iterations induce large reads/writes. Online backends avoid $O(nm)$ storage by evaluating interactions on-the-fly via generic GPU map-reduce (Charlier et al., 2021), but can still be slow because each Sinkhorn update is executed as generic reduction kernels with limited fusion/tiling and limited opportunity to exploit tensor-core GEMM structure. As a result, OT often fails to scale to repeated large point-cloud solves in downstream pipelines, especially in high dimensions.

We address this bottleneck with a systems-level observation: for squared Euclidean cost, stabilized log-domain Sinkhorn updates can be rewritten as row-wise LogSumExp (LSE) reductions of a biased dot-product score matrix—the same normalization that underlies scaled dot-product attention. FlashAttention (Dao et al., 2022; Dao, 2024) showed that this normalization can be computed exactly with substantially fewer HBM accesses by streaming tiles through on-chip SRAM and maintaining online max/sumexp statistics. We introduce **FlashSinkhorn**, which transfers this IO-aware strategy to EOT: fused Triton kernels stream tiles, compute costs on-the-fly, and write only updated dual potentials—avoiding $n \times m$ intermediates while, crucially, reducing HBM IO volume via fusion

and tiling. Our contributions are:

- **Attention-form Sinkhorn update.** We express stabilized Sinkhorn updates as biased dot-product LSE reductions with an iteration-dependent dual bias, enabling exact FlashAttention-style streaming.

- **IO-aware EOT.** We fused Triton kernels for streamed Sinkhorn iterations, along with transport matrix-free streaming operators for differentiation of an EOT loss.

- **Large-scale evaluation.** We demonstrate substantial speedups for forward, backward, and HVP computations, improving scalability on point-cloud OT and OT-based workloads including OTDD and shuffled regression.

## 2. Background

### 2.1. EOT and Sinkhorn Algorithm

Given source points (row vectors) $X = \{\mathbf{x}_i\}_{i=1}^n \subset \mathbb{R}^d$ with weights $\mathbf{a} \in \Delta^n$ and target points (row vectors) $Y = \{\mathbf{y}_j\}_{j=1}^m \subset \mathbb{R}^d$ with weights $\mathbf{b} \in \Delta^m$, define discrete distributions $\boldsymbol{\mu} = \sum_{i=1}^n a_i \delta_{\mathbf{x}_i}$ and $\boldsymbol{\nu} = \sum_{j=1}^m b_j \delta_{\mathbf{y}_j}$, where $\delta_{\mathbf{x}}$ denotes the Dirac measure at $\mathbf{x}$. Entropic optimal transport (EOT) (Peyré & Cuturi, 2019) is

$$\mathrm{OT}_\varepsilon(\boldsymbol{\mu}, \boldsymbol{\nu}) = \min_{P \in \Pi(\mathbf{a}, \mathbf{b})} \langle C, P \rangle + \varepsilon \mathrm{KL}\left(P \| \mathbf{a} \otimes \mathbf{b}\right), \quad (1)$$

where $\Pi(\mathbf{a}, \mathbf{b}) = \{P \in \mathbb{R}_{\geq 0}^{n \times m} : P\mathbb{1}_m = \mathbf{a}, P^\top \mathbb{1}_n = \mathbf{b}\}$ is the transport polytope, $C \in \mathbb{R}^{n \times m}$ is the cost matrix with entries $C_{ij} = \|\mathbf{x}_i - \mathbf{y}_j\|_2^2$, i.e., the squared Euclidean distance, and $\mathrm{KL}\left(P \| \mathbf{a} \otimes \mathbf{b}\right) := \sum_{i=1}^n \sum_{j=1}^m \left(P_{ij} \log \frac{P_{ij}}{a_i b_j} - P_{ij} + a_i b_j\right)$, denotes the relative entropy with $\varepsilon > 0$ the regularization strength.

Let $\mathbf{f} \in \mathbb{R}^n$ and $\mathbf{g} \in \mathbb{R}^m$ be the dual potentials. In the stabilized log-domain, Sinkhorn's fixed-point alternating iterations update $(\mathbf{f}, \mathbf{g})$ via log-sum-exp reductions (Altschuler et al., 2017; Schmitzer, 2019; Feydy, 2020):

$$f_i \leftarrow -\varepsilon \mathrm{LSE}_j \left[(g_j - C_{ij})/\varepsilon + \log b_j\right], \quad (2)$$
$$g_j \leftarrow -\varepsilon \mathrm{LSE}_i \left[(f_i - C_{ij})/\varepsilon + \log a_i\right], \quad (3)$$

where $\mathrm{LSE}(x) = \log \sum_k \exp(x_k)$ and $\mathrm{LSE}_j(\cdot)$ (resp. $\mathrm{LSE}_i(\cdot)$) denotes reduction over $j$ (resp. $i$). The optimal transport matrix is recovered as $P_{ij}^* = a_i b_j \exp\left[(f_i^* + g_j^* - C_{ij})/\varepsilon\right]$. A common symmetrized variant computes the two half-steps in parallel from the same $(\mathbf{f}^{\mathrm{old}}, \mathbf{g}^{\mathrm{old}})$ and then averages (Feydy, 2020):

$$f_i^{\mathrm{new}} \leftarrow \frac{f_i^{\mathrm{old}}}{2} - \frac{\varepsilon}{2} \mathrm{LSE}_j \left[(g_j^{\mathrm{old}} - C_{ij})/\varepsilon + \log b_j\right], \quad (4)$$
$$g_j^{\mathrm{new}} \leftarrow \frac{g_j^{\mathrm{old}}}{2} - \frac{\varepsilon}{2} \mathrm{LSE}_i \left[(f_i^{\mathrm{old}} - C_{ij})/\varepsilon + \log a_i\right]. \quad (5)$$

Unlike the alternating updates eq. (2)–(3), the two reductions are independent and therefore more amenable to parallel evaluation. Detailed derivation is in appendix B.

### 2.2. Differentiation of EOT with respect to data

The gradient of the EOT loss with respect to source points is (Feydy et al., 2019; Pooladian & Niles-Weed, 2021):

$$\nabla_{\mathbf{x}_i} \mathrm{OT}_\varepsilon = 2a_i\left(\mathbf{x}_i - T_\varepsilon(\mathbf{x}_i)\right), \; T_\varepsilon(\mathbf{x}_i) = \frac{1}{a_i}\sum_{j=1}^m P_{ij}^* \mathbf{y}_j.$$

where $T_\varepsilon$ is the barycentric projection.

The Hessian $\nabla_X^2 \mathrm{OT}_\varepsilon$ can be viewed as a 4th-order tensor $\mathcal{T} \in \mathbb{R}^{n \times d \times n \times d}$ with entries (Li et al., 2025b),

$$\mathcal{T}_{ktsl} = \frac{1}{\varepsilon} \sum_{i,j=1}^{n+m} \mathcal{R}_{ikt} \, H_{ij}^{*\dagger} \, \mathcal{R}_{jsl} \; + \; \mathcal{E}_{ktsl}, \quad (6)$$

where $H^{*\dagger}$ denotes the Moore-Penrose pseudoinverse of the sensitivity matrix $H^* = \begin{pmatrix} \mathrm{diag}(\mathbf{a}) & P^* \\ (P^*)^\top & \mathrm{diag}(\mathbf{b}) \end{pmatrix} \in \mathbb{R}^{(n+m) \times (n+m)}$. Define $\mathcal{B} \in \mathbb{R}^{n \times d \times m}$ by $\mathcal{B}_{ktj} := 2(x_{kt} - y_{jt}) P_{kj}^*$, and $\mathcal{R} \in \mathbb{R}^{(n+m) \times n \times d}$ slice-wise by

$$\mathcal{R}_{:,:,t} := \begin{pmatrix} \mathrm{Diag}\left(\mathcal{B}_{:,t,:} \mathbb{1}_m\right) \\ (\mathcal{B}_{:,t,:})^\top \end{pmatrix}.$$

The term $\mathcal{E}$ is block-diagonal across points: $\mathcal{E}_{k,:,s,:} = 0$ for $k \neq s$, and

$$\mathcal{E}_{k,:,k,:} = 2a_k \mathbb{I}_d - \frac{4}{\varepsilon} \sum_{j=1}^m P_{kj}^* (\mathbf{x}_k - \mathbf{y}_j)^\top (\mathbf{x}_k - \mathbf{y}_j). \quad (7)$$

If Sinkhorn is terminated before full convergence, the resulting coupling $\tilde{P}$ typically satisfies the marginal constraints only approximately. Let $\tilde{\mathbf{a}} := \tilde{P}\mathbb{1}_m$ and $\tilde{\mathbf{b}} := \tilde{P}^\top \mathbb{1}_n$ be its actual marginals. Then the expressions above remain exact when interpreted as derivatives of $\mathrm{OT}_\varepsilon(\tilde{\boldsymbol{\mu}}, \tilde{\boldsymbol{\nu}})$ with $\tilde{\boldsymbol{\mu}} = \sum_{i=1}^n \tilde{a}_i \delta_{\mathbf{x}_i}$ and $\tilde{\boldsymbol{\nu}} = \sum_{j=1}^m \tilde{b}_j \delta_{\mathbf{y}_j}$ (Feydy, 2020). The detailed derivation is in appendix C.

### 2.3. FlashAttention and Online LSE

Modern GPUs feature a memory hierarchy where on-chip static random-access memory (SRAM) offers an order of magnitude higher bandwidth than high-bandwidth memory (HBM), but with far smaller capacity. As computing speeds have outpaced memory bandwidth, many operations, including softmax and reductions, are memory-bound: their runtime is dominated by HBM accesses rather than arithmetic. Kernel fusion addresses this by loading inputs once from HBM and performing multiple operations in SRAM before writing final outputs to HBM.

---

**Algorithm 1** FlashSinkhorn: streaming $\hat{\mathbf{f}}$-update

---

1: **Input:** $X \in \mathbb{R}^{n \times d}, Y \in \mathbb{R}^{m \times d}, \hat{\mathbf{g}} \in \mathbb{R}^m, \mathbf{b} \in \Delta^m$
2: **Output:** $\hat{\mathbf{f}} \in \mathbb{R}^n$
3: $\boldsymbol{\delta} \leftarrow \varepsilon \log \mathbf{b}$           // *precompute bias*
4: **for** each row block $I$ of size $B_N$ **do**
5:     Load $X_I$ to on-chip SRAM
6:     $\mathbf{m}_I \leftarrow -\infty, \mathbf{s}_I \leftarrow 0$     // *running max / sumexp*
7:     **for** each column block $J$ of size $B_M$ **do**
8:        Load $Y_J, \hat{\mathbf{g}}_J$ and $\boldsymbol{\delta}_J$ to on-chip SRAM
9:        $S \leftarrow (2X_I Y_J^\top + \hat{\mathbf{g}}_J + \boldsymbol{\delta}_J)/\varepsilon$ // *score tile + bias*
10:       $\tilde{\mathbf{m}} \leftarrow \mathrm{rowmax}(S)$           // *tile max*
11:       $\mathbf{m}_{\mathrm{new}} \leftarrow \max(\mathbf{m}_I, \tilde{\mathbf{m}})$   // *update max row-wise*
12:       $\mathbf{s}_I \leftarrow e^{\mathbf{m}_I - \mathbf{m}_{\mathrm{new}}} \odot \mathbf{s}_I + \mathrm{rowsum}(e^{S - \mathbf{m}_{\mathrm{new}}})$
13:       $\mathbf{m}_I \leftarrow \mathbf{m}_{\mathrm{new}}$
14:     **end for**
15:     $\hat{\mathbf{f}}_I \leftarrow -\varepsilon (\mathbf{m}_I + \log \mathbf{s}_I)$       // $\hat{\mathbf{f}} = -\varepsilon \mathrm{LSE}_{\mathrm{row}}$
16:     Write $\hat{\mathbf{f}}_I$ to HBM
17: **end for**

---

FlashAttention (Dao et al., 2022) applies this principle to attention by avoiding materialization of the $n \times n$ score matrix $S = QK^\top$. The key is online log-sum-exp (LSE) (Milakov & Gimelshein, 2018). The detailed derivation is in appendix D.3. This fuses "score $\rightarrow$ softmax $\rightarrow$ aggregation" into one kernel that updates per-row and output accumulator tile-by-tile, writing only final results to HBM.

## 3. FlashSinkhorn: Algorithm and Analysis

### 3.1. Streaming Sinkhorn Iterations

For EOT with squared Euclidean cost, the log-domain Sinkhorn updates can be written as row-wise LSE reductions of a biased dot-product score matrix. This is exactly the same per-row normalization statistic that underlies scaled dot-product attention. As a result, each Sinkhorn iteration can be computed by FlashAttention-style streaming without materializing the cost matrix $C \in \mathbb{R}^{n \times m}$.

**Proposition 1** (Sinkhorn iteration as biased dot-product LSE). *Define* $\boldsymbol{\alpha} \in \mathbb{R}^n, \boldsymbol{\beta} \in \mathbb{R}^m$ *by* $\alpha_i = \|\mathbf{x}_i\|_2^2$ *and* $\beta_j = \|\mathbf{y}_j\|_2^2$. *Set* $Q := \sqrt{2}X$ *and* $K := \sqrt{2}Y$ *and define the precomputable row vectors* $\boldsymbol{\delta} := \varepsilon \log \mathbf{b} \in \mathbb{R}^m, \boldsymbol{\gamma} := \varepsilon \log \mathbf{a} \in \mathbb{R}^n$. *Define the shifted potentials* $\hat{\mathbf{f}} := \mathbf{f} - \boldsymbol{\alpha}$ *and* $\hat{\mathbf{g}} := \mathbf{g} - \boldsymbol{\beta}$. *Define the row-logits and column-logits*

$$S_X(\hat{\mathbf{g}}) := \left(QK^\top + \mathbb{1}_n(\hat{\mathbf{g}} + \boldsymbol{\delta})\right)/\varepsilon, \quad (8)$$

$$S_Y(\hat{\mathbf{f}}) := \left(KQ^\top + \mathbb{1}_m(\hat{\mathbf{f}} + \boldsymbol{\gamma})\right)/\varepsilon. \quad (9)$$

*Then the stabilized log-domain alternating Sinkhorn up-*

dates in eq. (2)–(3) are equivalently

$$\hat{\mathbf{f}} \leftarrow -\varepsilon \mathrm{LSE}_{\mathrm{row}}(S_X(\hat{\mathbf{g}})), \quad (10)$$

$$\hat{\mathbf{g}} \leftarrow -\varepsilon \mathrm{LSE}_{\mathrm{row}}\left(S_Y(\hat{\mathbf{f}})\right). \quad (11)$$

The proof is in appendix D.1. Consequently, Proposition 1 reduces each stabilized Sinkhorn half-step to a LSE of a biased dot-product score matrix. This is the same normalization that appears in IO-aware exact attention kernels: a tiled GEMM produces blocks of logits, while online LSE accumulators update per-row statistics on the fly, so the full $n \times m$ score matrix never needs to be materialized in high-bandwidth memory (HBM) (Dao, 2024).

To make this precise, we adopt a two-level memory model: inputs reside in slow HBM; each thread block has fast on-chip SRAM of size $M$; and we measure cost by the number of scalars transferred between HBM and SRAM. Under this model, a materialized Sinkhorn half-step writes the score matrix to HBM and then applies a row-wise LSE, incurring $\Theta(nd + md + nm)$ HBM accesses.

**Theorem 2.** *We consider the exact update in eq.* (10) *and assume a tiling with row blocks of size $B_N$ and column blocks of size $B_M$ fits in SRAM with $B_M d + B_N d + (B_M + 2B_N) \lesssim M$, then Algorithm 1 achieves $\Theta\left(nd + md + \frac{nmd^2}{M}\right)$ HBM accesses for $d \leq M \leq \min\{n,m\}d$, and $\Theta(nd + md)$ for $M \geq \min\{n,m\}d$.*

The full proof is in appendix D.2. The key idea is that SRAM of size $M$ allows us cache only $B_N = \Theta(M/d)$ rows of $Q$ at a time, hence we must make $\Theta(n/B_N) = \Theta(nd/M)$ passes over $K$, each streaming $\Theta(md)$ scalars from HBM, which yields the dominant $\Theta(nmd^2/M)$ HBM-access term.

We implement each stabilized Sinkhorn update eq. (10)–(11) as a single fused Triton GPU kernel that streams tiles of $Q$ and $K$, forms the biased scores, and maintains online row-wise LSE statistics on chip, writing only the updated potentials back to HBM. This mirrors FlashAttention's IO-aware online-softmax machinery. In FlashAttention, the forward pass streams blocks of $(K, V)$ and revisits all query blocks $Q$, writing intermediate row statistics/output blocks to HBM; in FlashSinkhorn we flip the nesting (row-stationary $Q$-outer, $K/V$-inner) so each row block retains its running LSE statistics on chip across all key/value blocks and is written out only once, which better matches Sinkhorn's row-wise reductions. The kernels are detailed in algorithms 1 and 3 and illustrated in Figure 1 (middle, right), with further implementation details in Appendix G. The kernels above implement the stabilized *alternating* schedule in eq. (2)–(3); for apples-to-apples comparison to libraries that default to a different schedule, we also fuse the *symmetric* updates eq. (4)–(5) in the same **single-kernel** streaming form.

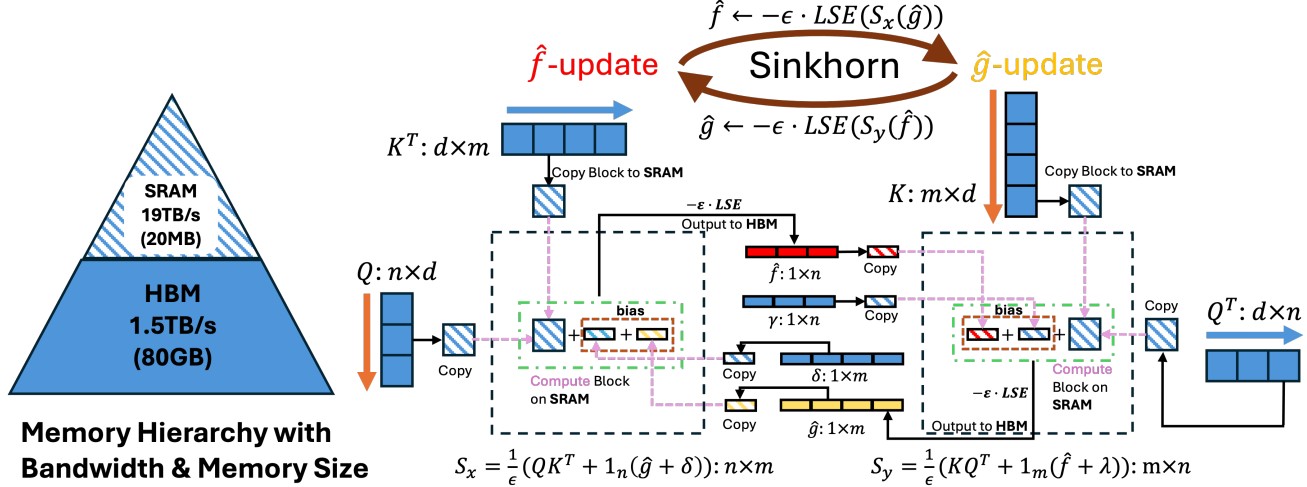

*Figure 1.* FlashSinkhorn uses tiling to avoid materializing the $n \times m$ score matrix on slow HBM. **Left:** GPU memory hierarchy. SRAM ($\sim$20 MB, $\sim$19 TB/s) is small but fast; HBM ($\sim$80 GB, $\sim$1.5 TB/s) is large but slow. **Middle:** streaming $\hat{\mathbf{f}}$ update (Algorithm 1). The outer loop (orange) stages a row block $Q_I$ in SRAM with running LSE statistics; the inner loop (blue) streams $K_J$ tiles plus bias $\hat{\mathbf{g}}_J + \boldsymbol{\delta}_J$, forms the score tile on-chip, and writes $\hat{\mathbf{f}}_I$ to HBM after the inner loop completes. **Right:** streaming $\hat{\mathbf{g}}$ update with $Q, K$ swapped (bias $\hat{\mathbf{f}}_I + \boldsymbol{\gamma}_I$). Hatched = SRAM tile, solid = HBM tensor; black arrows = HBM$\leftrightarrow$SRAM transfers, pink dashed = on-chip data flow.

**Scope of cost structure.** The same reduction applies to any cost of the form $C_{ij} = \alpha_i + \beta_j - Q_i^\top K_j$ with precomputable per-point terms $\alpha_i, \beta_j$ and explicit features $Q \in \mathbb{R}^{n \times d'}, K \in \mathbb{R}^{m \times d'}$: each Sinkhorn update then becomes a biased dot-product LSE that streams tile-by-tile without materializing the $n \times m$ cost matrix. Squared Euclidean is the primary instance ($Q = \sqrt{2}X$, $K = \sqrt{2}Y$), and cosine distance fits the same form after L2-normalization $(1 - x_i^\top y_j = \frac{1}{2}\|x_i - y_j\|_2^2$ on unit-norm inputs, with adjusted $\varepsilon$); the OTDD label-augmented cost in Section 4 adds a bounded additive lookup term (Alvarez-Melis & Fusi, 2020) evaluated on-the-fly inside the kernel. Costs without this structure (e.g., raw Euclidean $\|x - y\|_2$, where the square root breaks the affine form, or learned neural costs) do not admit this fused-streaming form and are future work.

### 3.2. Streaming Transport Matrix Application

The entropic barycentric projection admits an attention form: it is a row-wise softmax-weighted average of target data. As a result, $\nabla_X \mathrm{OT}_\varepsilon(\boldsymbol{\mu}, \boldsymbol{\nu})$ is a weighted residual between the source data and an attention output at optimality.

**Proposition 3** (Transport matrix application as an attention output)**.** *Given any shifted potentials $\hat{\mathbf{f}}$ and $\hat{\mathbf{g}}$, define the transport matrix*

$$P_{ij}(\hat{\mathbf{f}}, \hat{\mathbf{g}}) := a_i b_j \exp\left(\frac{\hat{f}_i + \hat{g}_j + (QK^\top)_{ij}}{\varepsilon}\right). \quad (12)$$

*Let $\hat{\mathbf{f}}^+ = -\varepsilon\mathrm{LSE}_{\mathrm{row}}(S_X(\hat{\mathbf{g}})), \hat{\mathbf{g}}^+ = -\varepsilon\mathrm{LSE}_{\mathrm{row}}(S_Y(\hat{\mathbf{f}})),$*

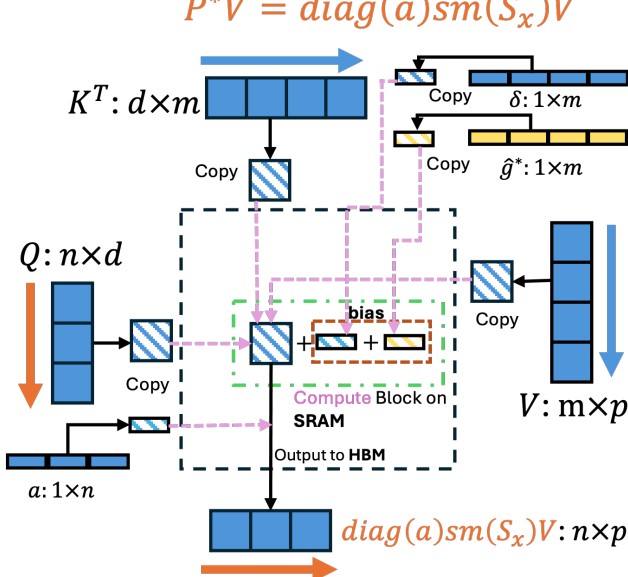

*Figure 2.* Streaming transport-matrix application $PV$ (Algorithm 2); at converged potentials this equals $P^*V$. The same tiling avoids materializing $P$ on HBM. The outer loop (orange, left) stages $Q_I, \hat{\mathbf{f}}_I, \mathbf{a}_I$ in SRAM; the inner loop (blue) streams $K_J, V_J$ plus bias $\hat{\mathbf{g}}_J + \boldsymbol{\delta}_J$ and accumulates an online weighted sum on-chip. After the inner loop, the source marginal correction is applied and $(PV)_I$ is written to HBM (orange, bottom).

*and define the row-mass vector and column-mass vector,*

$$\mathbf{r} := P\mathbb{1}_m = \mathbf{a} \odot \exp\left(\frac{\hat{\mathbf{f}} - \hat{\mathbf{f}}^+}{\varepsilon}\right), \quad (13)$$

$$\mathbf{c} := P^\top \mathbb{1}_n = \mathbf{b} \odot \exp\left(\frac{\hat{\mathbf{g}} - \hat{\mathbf{g}}^+}{\varepsilon}\right). \quad (14)$$

*Then for any $V \in \mathbb{R}^{m \times p}$ and any $U \in \mathbb{R}^{n \times p}$,*

$$P V = \operatorname{diag}(\mathbf{r}) \operatorname{Softmax}(S_X(\hat{\mathbf{g}})) V, \qquad (15)$$

$$P^\top U = \operatorname{diag}(\mathbf{c}) \operatorname{Softmax}(S_Y(\hat{\mathbf{f}})) U, \qquad (16)$$

*where softmax is applied row-wise. Moreover, if $(\hat{\mathbf{f}}^*, \hat{\mathbf{g}}^*)$ are the Sinkhorn fixed points, then $\mathbf{r} = \mathbf{a}$, $\mathbf{c} = \mathbf{b}$ and $P(\hat{\mathbf{f}}^*, \hat{\mathbf{g}}^*)$ in eq. (12) recovers the optimal transport $P^*$.*

The proof is in Appendix E.1. Algorithm 2 computes out $= P(\hat{\mathbf{f}}, \hat{\mathbf{g}})V$ with a FlashAttention-style fused *matmul–softmax–matmul* kernel (illustrated in Figure 2): it streams tiles to form biased dot-product scores, applies a row-wise softmax via online normalization, and accumulates values in a single pass. The identity holds for arbitrary potentials, so early stopping applies the induced coupling (with marginals $\mathbf{r}, \mathbf{c}$), while at convergence it recovers $P^*V$. Under the HBM/SRAM model, the IO cost is $\Theta\big((n+m)(d+p) + \frac{nm(d+p)^2}{M}\big)$ HBM accesses (Dao et al., 2022).

**Corollary 4.** *Let $(\hat{\mathbf{f}}^*, \hat{\mathbf{g}}^*)$ be converged shifted potentials of eq. (10)–(11). The barycentric projection $T_\varepsilon(X) := \operatorname{diag}(\mathbf{a})^{-1} P^* Y$ admits the attention form*

$$T_\varepsilon(X) = \operatorname{Softmax}_{\mathrm{row}}(S_X^*) Y, \qquad T_\varepsilon(\mathbf{x}_i) = [T_\varepsilon(X)]_i,$$

*and the gradient of EOT with respect to the source data has the residual-attention expression*

$$\nabla_X \operatorname{OT}_\varepsilon(\boldsymbol{\mu}, \boldsymbol{\nu}) = 2\operatorname{diag}(\mathbf{a})\Big(X - \operatorname{Softmax}(S_X^*)Y\Big). \quad (17)$$

The proof is in Appendix E.2; Table 1 summarizes the EOT–attention correspondence. Since both $P^*Y$ and (17) reduce to the same row-wise online softmax normalization and value accumulation as attention, they can be computed with the same FlashAttention-style streaming kernels.

*Table 1.* EOT to Attention Correspondence.

| EOT | Attention |
|---|---|
| Source $X$ | Queries $Q = \sqrt{2}X$ |
| Target $Y$ | Keys $K = \sqrt{2}Y$ |
| | Values $V = Y$ |
| Reg. $\varepsilon$ | Scaling $\sqrt{d}$ |
| $\hat{\mathbf{g}}^* + \varepsilon \log \mathbf{b}$ | Key bias |
| Bary. proj. $T_\varepsilon(X)$ | Attn. output $\operatorname{Softmax}(S^*)V$ |

### 3.3. Streaming Hessian-Vector Products

Second-order information is central to large-scale optimization (Shen et al., 2020; Brauer et al., 2017; Tang & Qiu, 2024), gradient flows (Yamamoto et al., 2026), and implicit layers (Eisenberger et al., 2022). Although the full Hessian $\mathcal{T} = \nabla_X^2 \operatorname{OT}_\varepsilon(\boldsymbol{\mu}, \boldsymbol{\nu})$ is infeasible to

---

**Algorithm 2** Apply Transport from Potentials, $P V$

1: **Input:** $X \in \mathbb{R}^{n \times d}$, $Y \in \mathbb{R}^{m \times d}$, $\hat{\mathbf{f}} \in \mathbb{R}^n$, $\hat{\mathbf{g}} \in \mathbb{R}^m$, $\mathbf{a} \in \Delta^n$, $\mathbf{b} \in \Delta^m$, $V \in \mathbb{R}^{m \times p}$
2: **Output:** out $= P V \in \mathbb{R}^{n \times p}$
3: $\boldsymbol{\delta} \leftarrow \varepsilon \log \mathbf{b}$         *// absorb marginal into bias*
4: **for** each row block $I$ of size $B_N$ **do**
5:      Load $X_I, \hat{\mathbf{f}}_I, \mathbf{a}_I$ to on-chip SRAM
6:      $\mathbf{m}_I \leftarrow -\infty$, $O_I \leftarrow 0$
7:      **for** each column block $J$ of size $B_M$ **do**
8:          Load $Y_J, \hat{\mathbf{g}}_J, \boldsymbol{\delta}_J, V_J$ to on-chip SRAM
9:          $S \leftarrow (2 X_I Y_J^\top + \hat{\mathbf{g}}_J + \boldsymbol{\delta}_J)/\varepsilon$
10:         $\tilde{\mathbf{m}} \leftarrow \operatorname{rowmax}(S)$        *// tile max*
11:         $\mathbf{m}_{\mathrm{new}} \leftarrow \max(\mathbf{m}_I, \tilde{\mathbf{m}})$    *// running max*
12:         $O_I \leftarrow e^{\mathbf{m}_I - \mathbf{m}_{\mathrm{new}}} \odot O_I + e^{S - \mathbf{m}_{\mathrm{new}}} V_J$   *// online weighted sum*
13:         $\mathbf{m}_I \leftarrow \mathbf{m}_{\mathrm{new}}$
14:      **end for**
15:      out$_I \leftarrow \mathbf{a}_I \odot \exp\Big(\hat{\mathbf{f}}_I/\varepsilon + \mathbf{m}_I\Big) \odot O_I$    *// source marginal correction*
16:      Write out$_I$ to HBM
17: **end for**

---

form for large $(n, m)$, many downstream routines only require Hessian–vector products (HVPs), e.g., Newton-CG, trust-region methods and Krylov eigenvalue estimation (Nocedal & Wright, 2006). We derive streaming HVPs that decompose into transport-vector, transport-matrix, and Hadamard-weighted transport applications $(P^* \odot AY^\top) Y$, avoiding materialization of both the Hessian tensor and the transport matrix. This reduces working memory from $O(n^2 d^2 + nm)$ to $O((n+m)d)$.

**Theorem 5** (Streaming HVP oracle). *Consider EOT with squared Euclidean cost. For any $A \in \mathbb{R}^{n \times d}$, the Hessian–vector product $G = \mathcal{T} A$ can be evaluated by streaming over the optimal coupling $P^*$ using $(2K_{CG} + 3)$ transport–vector products, 3 transport–matrix products, and 1 Hadamard-weighted transport, where $K_{CG}$ is the CG iteration count used to solve the Schur-complement linear system $H^* \mathbf{w} = \mathcal{R} A$. Consequently, flops $= O\big((K_{CG} + 1) nmd\big)$, memory $= O\big((n+m)d\big)$.*

*Proof sketch.* We start from the decomposition

$$G = \mathcal{T} A = \frac{1}{\varepsilon} \mathcal{R}^\top \mathbf{w} + \mathcal{E} A, \qquad \mathbf{w} = H^{*\dagger}(\mathcal{R} A).$$

Partition $\mathbf{w} = \begin{pmatrix} \mathbf{w}_1 \\ \mathbf{w}_2 \end{pmatrix} \in \mathbb{R}^{n+m}$ and $\mathbf{r} := \mathcal{R} A = \begin{pmatrix} \mathbf{r}_1 \\ \mathbf{r}_2 \end{pmatrix} \in \mathbb{R}^{n+m}$ with $\mathbf{w}_1, \mathbf{r}_1 \in \mathbb{R}^n$ and $\mathbf{w}_2, \mathbf{r}_2 \in \mathbb{R}^m$.

*(i) Explicit term.* The term $\mathcal{E} A$ reuses the cached transport-matrix product $P^*Y$ and requires one Hadamard-weighted transport application $(P^* \odot (AY^\top)) Y$.

*(ii) Implicit term.* We form $\mathbf{r} = \mathcal{R} A$ using $(P^*)^\top A$ (one

transport–vector product) and eliminate $\mathbf{w}_1$ to obtain the Schur system

$$S\,\mathbf{w}_2 = \text{rhs}, \qquad S := \text{diag}(\mathbf{b}) - (P^*)^\top \text{diag}(\mathbf{a})^{-1} P^*.$$

We solve this system by CG. Each CG iteration applies $S$ once, which amounts to one application of $P^*$ and one of $(P^*)^\top$ to vectors, plus diagonal scalings. After CG converges, we recover $\mathbf{w}_1 = \text{diag}(\mathbf{a})^{-1}(\mathbf{r}_1 - P^*\mathbf{w}_2)$, which requires one additional transport–vector product. Finally, we apply $\mathcal{R}^\top \mathbf{w}$, which uses one transport–matrix product together with the cached $P^*Y$. $\qquad\square$

In practice, we solve a damped Schur system $S_\tau := S + \tau I$ and stop CG at relative residual $\eta$, yielding a regularized HVP; the Moore–Penrose solution is recovered as $\tau \downarrow 0$ and $\eta \downarrow 0$ (see appendix H.2.3). For SPD $S_\tau$, standard CG bounds give $K_{CG} = O\big(\sqrt{\kappa(S_\tau)}\,\log(1/\eta)\big)$ with $\kappa(S_\tau)$ being the condition number of $S_\tau$. The streaming algorithm for the Hadamard-weighted transport $\big(P^* \odot (AY^\top)\big)Y$ from potentials is given in Algorithm 5; full derivations and operation counts appear in Appendix F.

# 4. Experiments

We evaluate FlashSinkhorn across three dimensions: computational speed, memory efficiency, and scalability. Our experiments span synthetic benchmarks of kernel performance comparing against GeomLoss and OTT-JAX, and two downstream tasks: (1) **OTDD** for computing distances between labeled datasets, and (2) **OT-based shuffled regression**, where our streaming HVP enables saddle detection and Newton acceleration. These experiments isolate the four main ingredients of FlashSinkhorn: the attention-style LSE reformulation (Proposition 1), the fused forward kernel (Tables 2, 5 and 6), the streamed transport/backward operator (Figure 3 forward+backward panels), and the streamed HVP (Figure 3 HVP panels).

## 4.1. Synthetic Benchmarks

We benchmark FlashSinkhorn against two widely used GPU implementations: GeomLoss (Feydy, 2019) (tensorized and KeOps) and OTT-JAX (Cuturi et al., 2022) (online) on synthetic point clouds (Figure 3 and Table 3). OTT-JAX is built on JAX/XLA and implements the stabilized log-sum-exp Sinkhorn solver via the standard alternating fixed-point updates (eq. (2)–(3)) (Cuturi et al., 2022). GeomLoss provides PyTorch operators with multiple backends: its core Sinkhorn solver uses a symmetric update schedule (eq. (4)–(5)); its tensorized mode materializes dense interactions, while its online mode represents cost and kernel matrices symbolically with KeOps LazyTensors (Charlier et al., 2021) and evaluates them via tiled map-reduce reductions.

We evaluate three computational regimes: forward pass measures time to compute dual potentials and OT cost; forward + backward reports end-to-end differentiation time for OT-based losses; and Hessian-vector product (HVP) benchmarks second-order computation via our streaming operators. Full experimental details are in Appendix H.2.

*Table 2.* NCU profiling of forward pass ($n=m=10{,}000$, $d=64$, 10 Sinkhorn iterations, A100-80GB).

|  | Tensor. | KeOps | Flash |
|---|---|---|---|
| HBM R/W (GB) | 98 | 0.14 | **0.08** |
| Runtime (ms) | 54.0 | 125.5 | **8.2** |
| SM Util. (%) | 98 | 49 | 74 |
| Mem. Stalls (%) | 79 | 8 | **3** |
| Bottleneck | Mem. | Comp. | Comp. |

**Speed.** FlashSinkhorn is the fastest method in our GPU baselines (Table 3): it achieves $9$–$32\times$ speedup over KeOps on the forward pass (and up to $161\times$ end-to-end at $d=512$), and up to $5.1\times$ over OTT-JAX. Compared to tensorized baselines, FlashSinkhorn is $1.7$–$3.5\times$ faster on the settings shown in Table 3; across a broader sweep where tensorized fits in memory, the speedup can be larger (up to $12\times$ Appendix H.2.1). At small $n$ (below ~2000) or very high $d$ with $n$ fitting in memory, tensorized baselines can be competitive because the dense cost matrix is precomputed and cached; Table 10 reports the per-$d$ crossover. Tensorized variants become impractical once $n$ grows to tens of thousands due to $O(nm)$ materialization. These gains come from kernel-level specialization rather than algorithmic differences. FlashSinkhorn fuses each update into a small number of streaming kernels that compute pairwise costs on-the-fly and maintain online LSE statistics, reducing intermediate writes and kernel-launch overhead.

On the backward pass, the gap widens at high $d$. While KeOps provides analytical gradients, realizing them entails additional all-pairs reductions and multiple kernel launches that re-evaluate the interaction kernel. In contrast, FlashSinkhorn reuses cached normalization statistics and fuses gradient accumulation into a small number of streamed passes, yielding $100$–$200\times$ speedups at $d \geq 512$ when KeOps completes (Table 9).

We report two FlashSinkhorn variants: *symmetric* (single fused update) is best at small $n$ due to fewer launches, while *alternating* (two simpler half-steps) wins at large $n$ and high $d$ where reduced register pressure improves efficiency. See Appendix H.2.1 and H.2.2 for full profiling and scaling analyses.

**Profiling insight.** Table 2 explains these trends. Tensorized appears busy (98% SM utilization) yet is bandwidth-limited: it moves 98 GB through HBM and spends 79% of cycles stalled on memory, dominated by repeatedly traversing dense $n \times m$ interactions across it-

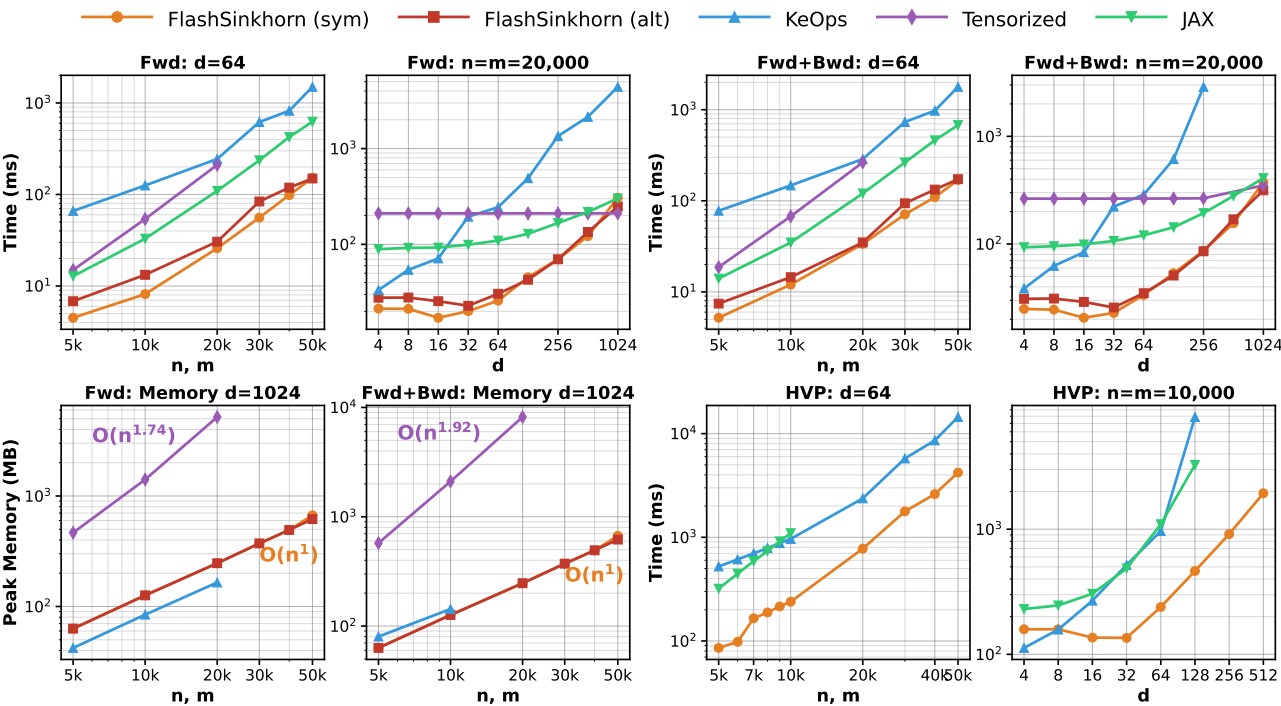

*Figure 3.* **FlashSinkhorn performance benchmarks.** (Top) Forward and forward+backward timing versus $n$ at $d{=}64$ and versus $d$ at $n{=}20k$. (Bottom left) Memory scaling at $d{=}1024$: FlashSinkhorn maintains $O(n)$ while tensorized methods scale as $O(n^{1.7}{-}n^{1.9})$. (Bottom right) HVP timing; FlashSinkhorn alone scales to $n{=}50k$ and $d{\geq}256$. A100-80GB, $\varepsilon{=}0.1$.

erations. KeOps avoids dense storage and drastically reduces HBM traffic (0.14 GB), but its generic map-reduce reductions achieve low utilization (49% SM) and high runtime (125.5 ms). FlashSinkhorn combines low HBM traffic (0.08 GB) with substantially fewer memory stalls (3%) and higher SM utilization (74%), yielding a $15.3\times$ lower runtime than KeOps in this setting (8.2 vs. 125.5 ms). The full NCU breakdown (Tables 5 to 7) attributes this gap to fused streaming: FlashSinkhorn issues $6.6\times$ fewer kernel launches (130 vs. 854) and routes $2.9\times$ more compute through the tensor pipeline (10.1 M vs. 3.5 M tensor-pipe instructions) by rewriting the squared-Euclidean interaction as tiled dot products.

**Memory.** Tensorized implementations store dense $n \times m$ intermediates, incurring $O(nm)$ memory and eventually OOM as $n$ grows. In contrast, FlashSinkhorn maintains only $O((n{+}m)d)$ resident state (points and dual potentials) and streams the interactions, enabling runs at $n{=}50k$ where tensorized baselines OOM and KeOps becomes out-of-time at high $d$ (Tables 3 and 8).

**Low-$\varepsilon$ regime.** At $\varepsilon \in \{0.10, 0.05, 0.01\}$ ($n{=}m{=}10000$, $d{=}64$, 10 iterations, TF32), FlashSinkhorn maintains its 10-iteration forward time across the range, yielding $16$–$17\times$ speedup over KeOps and $\sim 4\times$ over OTT-JAX (Table 19 in Appendix H.2.5). Numerical precision in fp32 is stable: relative error versus an fp64 dense reference

grows from $4{\times}10^{-5}$ at $\varepsilon{=}0.10$ to $7.7{\times}10^{-4}$ at $\varepsilon{=}0.01$ (Table 20). Per-iteration speed is essentially $\varepsilon$-independent; total solve time grows with the iteration budget required for convergence at smaller $\varepsilon$ (Table 21), and the streaming HVP retains $< 1\%$ relative error against a dense Moore–Penrose ground truth at $\varepsilon{=}0.01$ (Table 22).

*Table 3.* Speedup vs GeomLoss and OTT-JAX on A100. KeOps/Tensorized compared to flash(sym); JAX to flash(alt). OOM/OOT = out of memory/time (10 mins).

| | | Fwd | | | Fwd+Bwd | | |
|---|---|---|---|---|---|---|---|
| $n$ | $d$ | KeOps | Tensor. | Jax | KeOps | Tensor. | Jax |
| 10k | 128 | 9.4 | 3.2 | 2.0 | 10.3 | 3.5 | 2.0 |
| 10k | 512 | 31.7 | 1.7 | 1.3 | 161 | 1.7 | 1.2 |
| 40k | 128 | 12.5 | OOM | 3.0 | 13.5 | OOM | 2.9 |
| 40k | 512 | OOT | OOM | 1.6 | OOT | OOM | 1.6 |

**Hessian–vector products.** All methods compute HVPs using the same matrix-free Schur-complement/CG oracle from Theorem 5; differences come solely from the efficiency of the underlying transport-application operators inside the CG loop. As shown in the HVP panels of Figure 3, FlashSinkhorn achieves $3$–$6\times$ speedups at $d{=}64$ and up to $25\times$ at $d{=}128$; at $n{=}m{=}50{,}000$, $d{=}64$, FlashSinkhorn completes in 4.2s versus 14.5s for KeOps, while OTT-JAX does not complete within the time budget, making Newton–CG practical at previously infeasible scales.

## 4.2. Downstream Tasks

**OTDD Application.** We evaluate FlashSinkhorn on Optimal Transport Dataset Distance (OTDD) (Alvarez-Melis & Fusi, 2021; 2020), which compares labeled datasets using a feature–label cost

$$C(\mathbf{x}_i, \mathbf{y}_j) = \lambda_1 \|\mathbf{x}_i - \mathbf{y}_j\|_2^2 + \lambda_2 W[\ell_i, \ell_j],$$

where $W \in \mathbb{R}^{V \times V}$ stores class-to-class Wasserstein distances. OTDD uses the debiased Sinkhorn divergence (Feydy et al., 2019), requiring three OT solves per evaluation. This workload is computationally demanding at scale due to repeated OT solves including the precomputation of $W$; in particular, OTDD's analysis notes that a fully nonparametric construction of the label–label ground distances requires solving many inner OT problems and can have worst-case $O(n^5 \log n)$ complexity in the dataset size $n$ (Alvarez-Melis & Fusi, 2020).

FlashSinkhorn supports OTDD's label-augmented cost by caching the small $V \times V$ matrix $W$ and performing the lookup $W[\ell_i, \ell_j]$ on-the-fly inside the streamed kernels. This adds a modest per-iteration overhead relative to pure Euclidean cost due to increased register pressure, but avoids dense $n \times m$ score matrix materialization. In contrast, KeOps-style online backends express costs as coordinate-only formulas and do not directly support the discrete table-lookup term $W[\ell_i, \ell_j]$, so OTDD is typically run with tensorized backends.

We benchmark MNIST↔Fashion-MNIST (Deng, 2012; Xiao et al., 2017) using ResNet18 embeddings ($d=512$) in two settings: (i) OTDD distance (forward pass), and (ii) OTDD gradient flow (forward+backward) for dataset adaptation. As shown in Figure 4, FlashSinkhorn matches tensorized runtime up to the tensorized memory limit (a,b), while its peak memory grows linearly and remains under 1 GB at $n=60000$ (c,d), whereas the tensorized baseline exhibits $O(n^2)$ memory growth and OOMs beyond $n=20000$. Full details are in Appendix H.3.

**Detect Saddle Escape in OT-Based Regression.** We demonstrate a practical application of our streaming HVP oracle: detecting when saddle points are escaped in OT-based optimization, providing clear guidance on when second-order methods are beneficial.

We consider shuffled linear regression given $(X, \widetilde{Y})$ where $\widetilde{Y} = \Pi^*(XW^* + E)$ for unknown permutation $\Pi^*$, where $W^* \in \mathbb{R}^{d \times d}$ is the ground-truth linear map, and $E \in \mathbb{R}^{n \times d}$ is additive noise. Given only $(X, \widetilde{Y})$, the goal is to estimate $W^*$ by minimizing an EOT objective (Li et al., 2025b; Xie et al., 2021; Eisenberger et al., 2022),

$$\mathcal{L}(W) = \mathrm{OT}_\varepsilon \left( \frac{1}{n} \sum_i \delta_{\mathbf{y}_i}, \frac{1}{n} \sum_j \delta_{\widetilde{\mathbf{y}}_j} \right), \quad \mathbf{y}_i = \mathbf{x}_i W.$$

We use a standard flow cytometry dataset with $n = 40000$ single cells and $d = 5$ fluorescence channels (CD4, CD8, CD19, CD45, CD3), routinely used for lymphocyte immunophenotyping (Benson et al., 2014). We synthetically remove cell-level correspondences by applying an unknown permutation, modeling scenarios where two measurement sets are available only as unordered collections. Here $W^*$ represents the calibration matrix between measurement modalities.

While the parameter Hessian $H_W \in \mathbb{R}^{25 \times 25}$ is small, computing it requires the $\mathcal{T} = \nabla_Y^2 \mathrm{OT}_\varepsilon(\frac{1}{n} \sum_i \delta_{\mathbf{y}_i}, \frac{1}{n} \sum_j \delta_{\widetilde{\mathbf{y}}_j})$. Prior methods (Li et al., 2025b) materialize this Hessian, while streaming HVP computes $H_W \mathbf{v} = X^\top \mathcal{T} (X \mathbf{v})$ in $O(nd)$ memory, enabling Lanczos eigenvalue analysis at $n = 40000$ scale.

The objective $\mathcal{L}(W)$ is non-convex with saddle points that dominate random initializations (Abid & Zou, 2018). Newton fails in saddle regions (minimum eigenvalue of the Hessian $\lambda_{\min}(H_W) < 0$), and trust-region methods are prohibitively expensive at scale. The challenge is knowing when to switch. Our streaming HVP makes Lanczos cheap enough to monitor $\lambda_{\min}(H_W)$ every 5 steps throughout optimization, enabling a simple rule: full-batch adam while $\lambda_{\min} < 0.001$; Newton once $\lambda_{\min} \geq 0.001$. As shown in Figure 5, once escape is detected, Newton converges rapidly. Across 72 random initializations ($n = 40000$, $\varepsilon \in 0.1, 0.25, 0.5$), all started in saddle regions and full-batch adam successfully escaped 71 out of 72. Post-escape, Newton converged in a median of 11 steps. The full experiment details are in appendix H.4.

## 5. Related Work

**Attention as EOT.** Litman (2025) derives scaled dot-product attention (SDPA) as the exact solution of a one-sided EOT problem, i.e., a degenerate transport formulation that constrains only the row marginal and admits a closed-form of softmax solution. Litman & Guo (2026) generalize this view by replacing the implicit uniform attention prior with a learnable prior, while remaining compatible with optimized SDPA kernels. Our Sinkhorn half-steps share the algebraic pattern of a biased softmax/LSE, but the bias plays a different role: it is an iteration-dependent dual term updated to enforce the missing marginals into two-sided EOT. Accordingly, we focus on computation rather than interpretation: we target standard two-marginal EOT and show how stabilized Sinkhorn half-steps can be executed exactly via IO-aware streaming without materializing the $n \times m$ score matrix.

**Sinkhorn-style attention and online EOT.** Several transformer variants replace row-softmax with Sinkhorn balancing to impose global constraints on attention, e.g.,

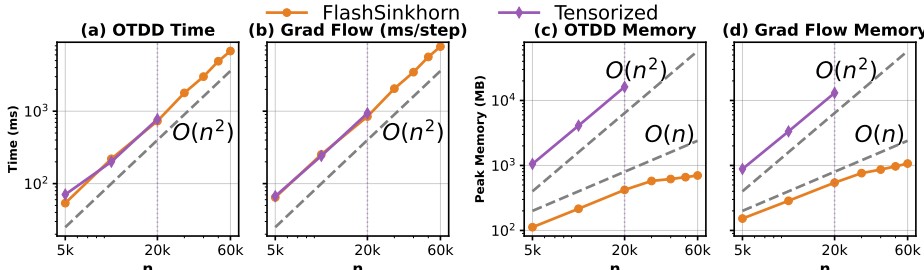

*Figure 4.* Scaling OTDD to large datasets (MNIST↔Fashion-MNIST, d=512). FlashSinkhorn vs tensorized baseline for (a,b) time and (c,d) memory on OTDD distance computation and gradient flow.

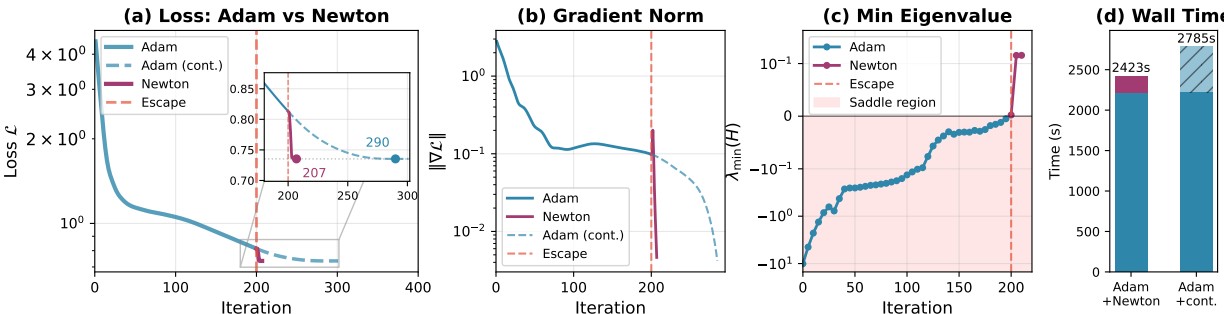

*Figure 5.* **Saddle escape trajectory comparison: Adam vs Newton** $n = 40000, \varepsilon = 0.1$ (a) Loss: Adam decay slowly in saddle region; post-escape Newton converges in 7 steps vs 90 for Adam continuation. (b) Gradient norm: Adam stalls near 0.1; Newton breaks through rapidly. (c) Minimum Hessian eigenvalue via Lanczos; negative←positive transition at step 200 triggers Newton switch. (d) Wall time: Adam+Newton (2423s) vs Adam-only (2785s), 2.8× speedup after escape.

doubly-stochastic attention in Sinkformers (Sander et al., 2022) and latent-permutation/sorting-based sparsification in Sparse Sinkhorn Attention (Tay et al., 2020). Recent works extend this line via OT-structured attention: ESPFormer (Shahbazi et al., 2025a) uses expected sliced transport plans for doubly-stochastic attention, and LOT-Former (Shahbazi et al., 2025b) obtains doubly-stochastic linear attention via a low-rank OT factorization. More broadly, Sinkhorn scaling is used as a differentiable relaxation of matchings, with Gumbel–Sinkhorn (Mena et al., 2018) framing Sinkhorn as a temperature-controlled matrix analog of softmax. Online Sinkhorn (Mensch & Peyré, 2020) addresses a complementary regime, estimating regularized OT between continuous distributions from sample streams, using stochastic-approximation updates with fresh samples and a growing nonparametric kernel-mixture representation, and can serve as a warm-start while constructing a discrete cost matrix. FlashSinkhorn assumes fixed discrete measures and speeds up stabilized Sinkhorn updates and transport applications by adopting FlashAttention's IO-aware tiling and online normalization, without ever materializing the $n \times m$ score matrix (Dao et al., 2022; Dao, 2024).

## 6. Conclusion

We introduced FlashSinkhorn, an IO-aware GPU implementation of stabilized Sinkhorn for EOT that avoids materializing the $n \times m$ matrices. By rewriting each Sinkhorn half-step as a biased dot-product LSE reduction, we leverage FlashAttention-style streaming to execute the iterations exactly while substantially reducing HBM traffic. We gain up to $32\times$ forward-pass and $161\times$ end-to-end speedups over state-of-the-art online baselines. So, large-scale EOT becomes practical in the memory-bound regime. Extending comparable fused-kernel benefits to costs outside the dot-product class of Section 3 (e.g., raw Euclidean $\|x - y\|_2$ and learned neural costs) is future work.

## Acknowledgements

FY is grateful for partial support from seed funding by the Center for Emerging Artificial Intelligence Systems at the University at Albany. This research was supported in part by SUNY System Administration using the SUNY AI Platform, and in part by the AI Computing Cluster at the University at Albany.

## Impact Statement

This paper presents work whose goal is to advance the field of Machine Learning. There are many potential societal consequences of our work, none which we feel must be specifically highlighted here.

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

# A. Notation

*Table 4.* Notation used throughout the paper.

| Symbol | Meaning |
|---|---|
| **Discrete measures and cost** | |
| $X = [\mathbf{x}_1; \ldots; \mathbf{x}_n] \in \mathbb{R}^{n \times d}$ | Source points (rows), $\mathbf{x}_i \in \mathbb{R}^d$. |
| $Y = [\mathbf{y}_1; \ldots; \mathbf{y}_m] \in \mathbb{R}^{m \times d}$ | Target points (rows), $\mathbf{y}_j \in \mathbb{R}^d$. |
| $n, m, d$ | #source points, #target points, ambient dimension. |
| $\mathbf{a} \in \Delta^n$, $\mathbf{b} \in \Delta^m$ | Probability weights on $X$ and $Y$ (simplex). |
| $\boldsymbol{\mu}$, $\boldsymbol{\nu}$ | Discrete measures induced by $(X, \mathbf{a})$ and $(Y, \mathbf{b})$. |
| $\delta_{\mathbf{x}}$ | Dirac measure at $\mathbf{x}$. |
| $c(\mathbf{x}_i, \mathbf{y}_j)$ | Ground cost; in this paper $c(\mathbf{x}_i, \mathbf{y}_j) = \|\mathbf{x}_i - \mathbf{y}_j\|_2^2$. |
| $C \in \mathbb{R}^{n \times m}$ | Cost matrix, $C_{ij} = c(\mathbf{x}_i, \mathbf{y}_j)$. |
| $\varepsilon > 0$ | Entropic regularization parameter. |
| **Entropic OT and Sinkhorn** | |
| $\mathrm{OT}_\varepsilon(\boldsymbol{\mu}, \boldsymbol{\nu})$ | EOT objective between $\boldsymbol{\mu}$ and $\boldsymbol{\nu}$. |
| $\mathbf{f} \in \mathbb{R}^n$, $\mathbf{g} \in \mathbb{R}^m$ | Dual potentials (log-domain). |
| $\boldsymbol{\alpha} \in \mathbb{R}^n$, $\boldsymbol{\beta} \in \mathbb{R}^m$ | Squared norms, $\alpha_i = \|\mathbf{x}_i\|_2^2$, $\beta_j = \|\mathbf{y}_j\|_2^2$. |
| $\hat{\mathbf{f}} := \mathbf{f} - \boldsymbol{\alpha}$, $\hat{\mathbf{g}} := \mathbf{g} - \boldsymbol{\beta}$ | Shifted (stabilized) potentials used in kernels. |
| $\boldsymbol{\gamma} := \varepsilon \log \mathbf{a}$, $\quad \boldsymbol{\delta} := \varepsilon \log \mathbf{b}$ | Precomputable log-weight biases. |
| $P(\hat{\mathbf{f}}, \hat{\mathbf{g}}) \in \mathbb{R}^{n \times m}$ | Coupling induced by potentials; $P^*$ denotes the converged/optimal coupling. |
| $\mathbf{r} := P\mathbb{1}_m$, $\mathbf{c} := P^\top \mathbb{1}_n$ | Induced row/column marginals of $P$. |
| $T_\varepsilon(X) = \mathrm{diag}(\mathbf{a})^{-1} P^* Y$ | Entropic barycentric projection. |
| **Attention-form variables** | |
| $Q := \sqrt{2}\, X$, $K := \sqrt{2}\, Y$ | Query/key matrices for squared-Euclidean dot-product form. |
| $S_X(\hat{\mathbf{g}})$ | Biased score matrix for the $\hat{\mathbf{f}}$-update (row-wise reduction). |
| $S_Y(\hat{\mathbf{f}})$ | Biased score matrix for the $\hat{\mathbf{g}}$-update (row-wise reduction). |
| $V \in \mathbb{R}^{m \times p}, U \in \mathbb{R}^{n \times p}$ | "Values" used in transport application ($PV$, $P^\top U$). |
| $\mathrm{LSE}_{\mathrm{row}}(S)$ | Row-wise LogSumExp: $[\mathrm{LSE}_{\mathrm{row}}(S)]_i = \log \sum_j e^{S_{ij}}$. |
| $\mathrm{Softmax}_{\mathrm{row}}(\mathrm{S})$ | Row-wise softmax: $[\mathrm{Softmax}_{\mathrm{row}}(\mathrm{S})]_{ij} = e^{S_{ij}} / \sum_k e^{S_{ik}}$. |
| **Streaming kernels and IO model** | |
| HBM / SRAM | Off-chip high-bandwidth memory / on-chip shared memory (per thread block). |
| $M$ | SRAM capacity (in scalars) available to a thread block. |
| $B_N$, $B_M$ | Row/column tile sizes used in streaming kernels. |
| $\mathbf{m}_I$, $\mathbf{s}_I$ | Running row-wise max and sumexp accumulators for row block $I$. |
| $O_I$ | Output accumulator for row block $I$ (e.g., for $PV$). |
| **Second-order operators (HVP)** | |
| $\mathcal{T}$ | Data-space Hessian linear operator for EOT (used in $G = \mathcal{T}A$). |
| $A \in \mathbb{R}^{n \times d}$ | Input perturbation for a Hessian–vector product. |
| $G = \mathcal{T}A \in \mathbb{R}^{n \times d}$ | Hessian–vector product output. |
| $H^* \in \mathbb{R}^{(n+m) \times (n+m)}$ | Linear system matrix arising in implicit differentiation; $H^{*\dagger}$ its pseudoinverse. |
| $\mathcal{R}$ | Linear map defining the right-hand side $\mathcal{R}A$ in the implicit solve. |
| $S$ | Schur complement system solved by CG in the HVP oracle. |
| $K_{CG}$ | Number of CG iterations used for the Schur solve. |
| $\tau$, $\eta$ | Schur damping ($S_\tau = S + \tau I$) and CG relative-residual tolerance. |
| **OTDD (label-augmented cost)** | |
| $\ell_i, \ell_j \in \{1, \ldots, V\}$ | Class labels for samples. |
| $W \in \mathbb{R}^{V \times V}$ | Class-to-class Wasserstein distances used in OTDD. |
| $\lambda_1, \lambda_2$ | Feature/label cost weights in OTDD. |

## B. Derivation of Sinkhorn algorithm

Recall from (1) that the cost function for the entropic Optimal Transport (EOT) is given by

$$\mathrm{OT}_\varepsilon(\boldsymbol{\mu}, \boldsymbol{\nu}) = \min_{P \in \Pi(\mathbf{a}, \mathbf{b})} \langle C, P \rangle + \varepsilon \mathrm{KL}\left(P \| \mathbf{a} \otimes \mathbf{b}\right)$$

where $\Pi(\mathbf{a}, \mathbf{b}) = \{P \in \mathbb{R}_{\geq 0}^{n \times m} : P \mathbb{1}_m = \mathbf{a}, P^\top \mathbb{1}_n = \mathbf{b}\}$, $C \in \mathbb{R}^{n \times m}$ is the cost matrix with entries $c_{ij} = \|\mathbf{x}_i - \mathbf{y}_j\|_2^2$, and $\mathrm{KL}(P \| \mathbf{a} \otimes \mathbf{b}) = \sum_{ij} \left(P_{ij} \log \frac{P_{ij}}{a_i b_j} - P_{ij} + a_i b_j\right)$ is the Kullback–Leibler (KL) divergence (also called relative entropy, in the generalized/Bregman form so that it remains well-defined for non-probability $P$), and $\varepsilon$ is the entropy regularization strength.

We form the Lagrangian of the EOT problem (assuming $a_i, b_j > 0$),

$$\mathcal{L}(\mathbf{f}, \mathbf{g}, P) = \sum_{ij} \left[c_{ij} P_{ij} + \varepsilon P_{ij} \log \frac{P_{ij}}{a_i b_j} - \varepsilon P_{ij}\right]$$
$$- \sum_i f_i \left(\sum_j P_{ij} - a_i\right) - \sum_j g_j \left(\sum_i P_{ij} - b_j\right) + \varepsilon.$$

where $\mathbf{f}$ and $\mathbf{g}$ are the Lagrange multipliers, or the dual potentials. The pair $(\mathbf{f}, \mathbf{g})$ is determined up to a gauge shift $(\mathbf{f} + c, \mathbf{g} - c)$ for any constant $c$.

The optimal $P^*$ is determined by the first-order optimality condition $\frac{\partial \mathcal{L}}{\partial P}|_{P^*} = 0$, and can be expressed as a function of the dual potentials $f$ and $g$,

$$P_{ij}^* := a_i b_j \exp\left(\frac{f_i^* + g_j^* - c_{ij}}{\epsilon}\right). \tag{18}$$

Recall the marginal constraints of EOT

$$\sum_j P_{ij} = a_i, \quad \sum_i P_{ij} = b_j, \tag{19}$$

and plugging the optimal $P_{ij}^*$ (18), we get

$$\sum_j a_i b_j \exp\left(\frac{f_i^* + g_j^* - c_{ij}}{\varepsilon}\right) = a_i, \quad \sum_i a_i b_j \exp\left(\frac{f_i^* + g_j^* - c_{ij}}{\varepsilon}\right) = b_j.$$

If we solve the above marginal constraints for $\mathbf{f}^*$ and $\mathbf{g}^*$ by alternatively scaling $\mathbf{f}^{(k)}$ and $\mathbf{g}^{(k)}$ in the $\log$-domain, then we obtain the Sinkhorn algorithm with the LSE function.

Note that the marginal constraints are equivalent to

$$\exp\left(\frac{f_i^*}{\varepsilon}\right) \sum_j \exp\left(\log(b_j) + \frac{g_j^* - c_{ij}}{\varepsilon}\right) = 1,$$
$$\exp\left(\frac{g_j^*}{\varepsilon}\right) \sum_i \exp\left(\log(a_i) + \frac{f_i^* - c_{ij}}{\varepsilon}\right) = 1,$$

so we take $\log$ on both sides and get

$$\frac{f_i^*}{\varepsilon} + \mathrm{LSE}_j\left(\log(b_j) + \frac{g_j^* - c_{ij}}{\varepsilon}\right) = 0,$$
$$\frac{g_j^*}{\varepsilon} + \mathrm{LSE}_i\left(\log(a_i) + \frac{f_i^* - c_{ij}}{\varepsilon}\right) = 0. \tag{20}$$

If we apply iterative schemes to solve $\mathbf{f}^*$ and $\mathbf{g}^*$, at the $k$-th iteration,

- if we use the symmetric Sinkhorn with a half step Jacobi type update, which is adopted by the geomloss (Feydy, 2020), then the iteration reads

$$f_i^{\text{new}} = \frac{f_i^{(k)}}{2} - \frac{\varepsilon}{2}\text{LSE}_j\left(\log(b_j) + \frac{g_j^{\text{old}} - c_{ij}}{\varepsilon}\right),$$

$$g_j^{\text{new}} = \frac{g_j^{(k)}}{2} - \frac{\varepsilon}{2}\text{LSE}_i\left(\log(a_i) + \frac{f_i^{\text{old}} - c_{ij}}{\varepsilon}\right).$$

- if we use the Gauss–Seidel type iteration which is adopted by the OTT (Cuturi et al., 2022), then the update reads

$$f_i^{\text{new}} = -\varepsilon\text{LSE}_j\left(\log(b_j) + \frac{g_j^{\text{old}} - c_{ij}}{\varepsilon}\right),$$

$$g_j^{\text{new}} = -\varepsilon\text{LSE}_i\left(\log(a_i) + \frac{f_i^{\text{new}} - c_{ij}}{\varepsilon}\right).$$

Note that Jacobi-type updates allow fully parallel updates, whereas Gauss–Seidel type updates require sequential updates. For small datasets (i.e., when $n, m = O(10^3)$ or smaller), Jacobi-type iterations are typically more efficient; in contrast, for large-scale problems, Gauss–Seidel-type iterations tend to be faster.

## C. Differentiation of EOT with respect to data

In this section, we derive the first- and second-order derivatives of EOT with respect to the source data $X$. The expressions of derivatives with respect to the target data $Y$ will be similar. These closed-form expressions enable optimization without implicit differentiation, substantially improving memory efficiency, computational speed, and numerical robustness.

Assume the squared Euclidean cost $c_{ij} = \|\mathbf{x}_i - \mathbf{y}_j\|_2^2$, and let $P^* \in \mathbb{R}^{n \times m}$ be the optimal transport matrix with marginals $\mathbf{a} \in \Delta^n, \mathbf{b} \in \Delta^m$, and optimal dual potentials $\mathbf{f}^* \in \mathbb{R}^n, \mathbf{g}^* \in \mathbb{R}^m$.

**The first-order derivative.** Suppose the optimal $(\mathbf{f}^*, \mathbf{g}^*)$ is a single element of the dual problem $\mathcal{L}(\mathbf{f}, \mathbf{g}, P)$, then by simply applying the Danskin's theorem (Bertsekas, 1997; Cuturi & Doucet, 2014) to the dual formulation, we obtain the first-order derivatives with respect to the source dataset $\mathbf{X}$.

$$\frac{d\text{OT}_\varepsilon(\boldsymbol{\mu}, \boldsymbol{\nu})}{d\mathbf{x}_k} = \frac{\partial\mathcal{L}(\mathbf{f}, \mathbf{g}, P)}{\partial\mathbf{x}_k} = \sum_{j=1}^m P_{kj}^* \frac{\partial c_{kj}}{\partial\mathbf{x}_k} = 2\sum_{j=1}^m P_{kj}^*\left(\mathbf{x}_k - \mathbf{y}_j\right) = 2a_k\left(\mathbf{x}_k - T_\varepsilon(\mathbf{x}_k)\right), \tag{21}$$

where $T_\varepsilon$ is the barycentric projection defined as $T_\varepsilon(\mathbf{x}_i) = \frac{1}{a_i}\sum_{j=1}^m P_{ij}^* \mathbf{y}_j$.

**The second-order derivative.** This section derives the Hessian of the entropic optimal transport (EOT) objective $\text{OT}_\varepsilon(\boldsymbol{\mu}, \boldsymbol{\nu})$ with respect to the source support $X$ in eq. (6).

Define the sensitivity matrix $H^*$,

$$H^* := \begin{pmatrix} \text{diag}(\mathbf{a}) & P^* \\ (P^*)^\top & \text{diag}(\mathbf{b}) \end{pmatrix} \in \mathbb{R}^{(n+m)\times(n+m)}.$$

For each $k \in [n]$, define $B_k \in \mathbb{R}^{d \times m}$ by

$$(B_k)_{tj} := 2(x_{k,t} - y_{j,t})P_{kj}^*, \qquad t \in [d], \ j \in [m].$$

**Lemma 6.** *For each $k \in [n]$, the Jacobians of the optimal potentials with respect to $\mathbf{x}_k$,*

$$\frac{\partial\mathbf{f}^*}{\partial\mathbf{x}_k} \in \mathbb{R}^{n \times d}, \qquad \frac{\partial\mathbf{g}^*}{\partial\mathbf{x}_k} \in \mathbb{R}^{m \times d}, \qquad \left(\frac{\partial\mathbf{f}^*}{\partial\mathbf{x}_k}\right)_{it} := \frac{\partial f_i^*}{\partial x_{k,t}}, \ \left(\frac{\partial\mathbf{g}^*}{\partial\mathbf{x}_k}\right)_{jt} := \frac{\partial g_j^*}{\partial x_{k,t}},$$

*solve the linear system*

$$H^* \begin{pmatrix} \dfrac{\partial \mathbf{f}^*}{\partial \mathbf{x}_k} \\ \dfrac{\partial \mathbf{g}^*}{\partial \mathbf{x}_k} \end{pmatrix} = \begin{pmatrix} \mathbf{e}_k \, (B_k \mathbb{1}_m)^\top \\ B_k^\top \end{pmatrix}, \tag{22}$$

*where* $\mathbf{e}_k \in \mathbb{R}^n$ *is the* $k$*-th standard basis vector.*

*Proof.* Differentiate the marginal constraints $\sum_{j=1}^m P_{ij}^* = a_i$ and $\sum_{i=1}^n P_{ij}^* = b_j$ with respect to $x_{k,t}$. Since $\mathbf{a}, \mathbf{b}$ are fixed,

$$\sum_{j=1}^m \frac{\partial P_{ij}^*}{\partial x_{k,t}} = 0, \qquad \sum_{i=1}^n \frac{\partial P_{ij}^*}{\partial x_{k,t}} = 0.$$

Using the optimality form (log-domain) of $P^*$ (cf. (18)),

$$\frac{\partial P_{ij}^*}{\partial x_{k,t}} = \frac{P_{ij}^*}{\varepsilon} \Big( \frac{\partial f_i^*}{\partial x_{k,t}} + \frac{\partial g_j^*}{\partial x_{k,t}} - \frac{\partial c_{ij}}{\partial x_{k,t}} \Big), \qquad \frac{\partial c_{ij}}{\partial x_{k,t}} = 2(x_{i,t} - y_{j,t}) \, \delta_{ik}.$$

Plugging in and using $\sum_j P_{ij}^* = a_i$, $\sum_i P_{ij}^* = b_j$ yields

$$a_i \frac{\partial f_i^*}{\partial x_{k,t}} + \sum_{j=1}^m P_{ij}^* \frac{\partial g_j^*}{\partial x_{k,t}} = \delta_{ik} \sum_{j=1}^m 2(x_{k,t} - y_{j,t}) P_{kj}^*, \qquad i \in [n],$$

$$\sum_{i=1}^n P_{ij}^* \frac{\partial f_i^*}{\partial x_{k,t}} + b_j \frac{\partial g_j^*}{\partial x_{k,t}} = 2(x_{k,t} - y_{j,t}) P_{kj}^*, \qquad j \in [m].$$

Stacking these identities over $t = 1, \ldots, d$ gives (22). $\qquad \square$

Define $\mathcal{B} \in \mathbb{R}^{n \times d \times m}$ by $\mathcal{B}_{ktj} := 2(x_{k,t} - y_{j,t}) P_{kj}^*$, and define $\mathcal{R} \in \mathbb{R}^{(n+m) \times n \times d}$ slice-wise (for each $t \in [d]$) as

$$\mathcal{R}_{:,:,t} := \begin{pmatrix} \mathrm{diag}(\mathcal{B}_{:,t,:} \mathbb{1}_m) \\ (\mathcal{B}_{:,t,:})^\top \end{pmatrix}.$$

Note that $\mathcal{R}_{:,k,:} \in \mathbb{R}^{(n+m) \times d}$ coincides with the right-hand side of (22).

**Theorem 7.** *Let* $\mathcal{T} = \nabla_X^2 \mathrm{OT}_\varepsilon(\boldsymbol{\mu}, \boldsymbol{\nu}) \in \mathbb{R}^{n \times d \times n \times d}$*. Its entries decompose as*

$$\mathcal{T}_{ktsl} = \frac{1}{\varepsilon} \sum_{i,j=1}^{n+m} \mathcal{R}_{ikt} \, H_{ij}^{*\dagger} \, \mathcal{R}_{jsl} \; + \; \mathcal{E}_{ktsl}, \tag{23}$$

*where* $H^{*\dagger}$ *is the Moore–Penrose pseudoinverse of* $H^*$*, and* $\mathcal{E}$ *is block-diagonal across source points:*

$$\mathcal{E}_{k,:,s,:} = 0 \quad (k \neq s), \qquad \mathcal{E}_{k,:,k,:} = 2a_k I_d - \frac{4}{\varepsilon} \sum_{j=1}^m P_{kj}^* \, (\mathbf{x}_k - \mathbf{y}_j)^\top (\mathbf{x}_k - \mathbf{y}_j).$$

*Proof.* For the squared Euclidean cost,

$$\frac{\partial \mathrm{OT}_\varepsilon(\boldsymbol{\mu}, \boldsymbol{\nu})}{\partial \mathbf{x}_k} = 2 \sum_{j=1}^m P_{kj}^* (\mathbf{x}_k - \mathbf{y}_j) \in \mathbb{R}^{1 \times d}.$$

Fix $s \in [n]$. Differentiating entry-wise gives, for $t, l \in [d]$,

$$\frac{\partial^2 \mathrm{OT}_\varepsilon(\boldsymbol{\mu}, \boldsymbol{\nu})}{\partial x_{s,l} \, \partial x_{k,t}} = 2 \sum_{j=1}^m (x_{k,t} - y_{j,t}) \frac{\partial P_{kj}^*}{\partial x_{s,l}} \; + \; 2 \delta_{ks} \sum_{j=1}^m P_{kj}^* \, \delta_{tl}.$$

The second term equals $2a_k\delta_{ks}\delta_{tl}$. Using

$$\frac{\partial P^*_{kj}}{\partial x_{s,l}} = \frac{P^*_{kj}}{\varepsilon}\Big(\frac{\partial f^*_k}{\partial x_{s,l}} + \frac{\partial g^*_j}{\partial x_{s,l}} - \frac{\partial c_{kj}}{\partial x_{s,l}}\Big), \qquad \frac{\partial c_{kj}}{\partial x_{s,l}} = 2(x_{k,l} - y_{j,l})\delta_{ks},$$

yields

$$\frac{\partial^2 \mathrm{OT}_\varepsilon}{\partial x_{s,l}\,\partial x_{k,t}} = \frac{1}{\varepsilon}\sum_{j=1}^m \mathcal{B}_{ktj}\Big(\frac{\partial f^*_k}{\partial x_{s,l}} + \frac{\partial g^*_j}{\partial x_{s,l}}\Big) \;+\; 2a_k\delta_{ks}\delta_{tl} \;-\; \frac{4}{\varepsilon}\delta_{ks}\sum_{j=1}^m P^*_{kj}(x_{k,t} - y_{j,t})(x_{k,l} - y_{j,l}).$$

By Lemma 6, $\begin{pmatrix}\partial \mathbf{f}^*/\partial \mathbf{x}_s \\ \partial \mathbf{g}^*/\partial \mathbf{x}_s\end{pmatrix} = H^{*\dagger}\mathcal{R}_{:,s,:}$. Therefore,

$$\frac{1}{\varepsilon}\sum_{j=1}^m \mathcal{B}_{ktj}\Big(\frac{\partial f^*_k}{\partial x_{s,l}} + \frac{\partial g^*_j}{\partial x_{s,l}}\Big) = \frac{1}{\varepsilon}\sum_{i,j=1}^{n+m} \mathcal{R}_{ikt}\,H^{*\dagger}_{ij}\,\mathcal{R}_{jsl},$$

and the remaining $\delta_{ks}$-terms form $\mathcal{E}_{ktsl}$. This proves eq. (6). $\qquad\square$

**Remark 8.** *For strictly positive couplings (as produced by entropy regularization), $H^*$ is singular with a simple zero eigenvalue; the pseudoinverse is therefore required. Moreover, the smallest positive eigenvalue of $H^*$ can be very small in low-$\varepsilon$ regimes (Li et al., 2025b), making Hessian-related computations numerically ill-conditioned. In practice, truncated SVD, Tikhonov regularization, or preconditioning may be necessary to ensure numerical stability (Cuturi et al., 2022; Kemertas et al., 2025). Although these techniques yield approximations of the Hessian–vector product (HVP), they can still provide accurate estimates of the Hessian spectrum with respect to the dataset or feature variables, depending on the chosen stabilization strategy.*

## D. Proof in Section 3.1

### D.1. Proof of Proposition 1

*Proof.* Throughout we use the identity for squared Euclidean cost

$$C_{ij} = \|\mathbf{x}_i - \mathbf{y}_j\|_2^2 = \|\mathbf{x}_i\|_2^2 + \|\mathbf{y}_j\|_2^2 - 2\langle \mathbf{x}_i, \mathbf{y}_j\rangle = \alpha_i + \beta_j - (QK^\top)_{ij},$$

since $QK^\top = (\sqrt{2}X)(\sqrt{2}Y)^\top = 2XY^\top$ and $(XY^\top)_{ij} = \mathbf{x}_i\mathbf{y}_j^\top$. We also recall that for any constant $c$, $\mathrm{LSE}_j(z_j + c) = \mathrm{LSE}_j(z_j) + c$.

Define $\boldsymbol{\delta} = \varepsilon \log \mathbf{b}$ and $\boldsymbol{\gamma} = \varepsilon \log \mathbf{a}$, so that $\log b_j = \delta_j/\varepsilon$ and $\log a_i = \gamma_i/\varepsilon$. Define shifted potentials $\hat{f}_i = f_i - \alpha_i$ and $\hat{g}_j = g_j - \beta_j$.

Starting from the stabilized update (2),

$$f_i = -\varepsilon\,\mathrm{LSE}_j\left[\frac{g_j - C_{ij}}{\varepsilon} + \log b_j\right]$$

$$= -\varepsilon\,\mathrm{LSE}_j\left[\frac{g_j - (\alpha_i + \beta_j - (QK^\top)_{ij})}{\varepsilon} + \frac{\delta_j}{\varepsilon}\right]$$

$$= -\varepsilon\,\mathrm{LSE}_j\left[\frac{(QK^\top)_{ij} + (g_j - \beta_j) + \delta_j - \alpha_i}{\varepsilon}\right]$$

$$= -\varepsilon\left(-\frac{\alpha_i}{\varepsilon} + \mathrm{LSE}_j\left[\frac{(QK^\top)_{ij} + \hat{g}_j + \delta_j}{\varepsilon}\right]\right)$$

$$= \alpha_i - \varepsilon\,\mathrm{LSE}_j\left[\frac{(QK^\top)_{ij} + \hat{g}_j + \delta_j}{\varepsilon}\right].$$

Subtracting $\alpha_i$ from both sides yields

$$\hat{f}_i = -\varepsilon\,\mathrm{LSE}_j\left[\frac{(QK^\top)_{ij} + \hat{g}_j + \delta_j}{\varepsilon}\right].$$

In matrix form, the quantity inside $\mathrm{LSE}_j$ is exactly the $i$th row of

$$S_X(\hat{\mathbf{g}}) = \frac{QK^\top + \mathbb{1}_n(\hat{\mathbf{g}} + \boldsymbol{\delta})}{\varepsilon},$$

hence $\hat{\mathbf{f}} = -\varepsilon \, \mathrm{LSE}_{\mathrm{row}}(S_X(\hat{\mathbf{g}}))$, which is (10).

For $\hat{\mathbf{g}}$ update, $\hat{\mathbf{g}} = -\varepsilon \, \mathrm{LSE}_{\mathrm{row}}(S_Y(\hat{\mathbf{f}}))$ can be proved similarly. This proves the claimed equivalence. $\qquad\square$

## D.2. Proof of Theorem 2

*Proof.* Fix one row block $I$ of size $B_N$. Algorithm 1 loads $Q_I \in \mathbb{R}^{B_N \times d}$ once from HBM and keeps it in SRAM. It maintains running vectors $m_I, s_I \in \mathbb{R}^{B_N}$ on-chip, and then scans over all column blocks $J$. Across the full scan over $J$, every element of $K \in \mathbb{R}^{m \times d}$ is loaded exactly once (just partitioned into tiles) and contributes $\Theta(md)$ HBM reads, and every element of the bias $u \in \mathbb{R}^m$ is loaded once and contributes $\Theta(m)$. No logits tile is written to HBM; only the final $\hat{\mathbf{f}}_I$ is written back, which results in a total cost $\Theta(B_N)$.

Recall $\lceil \cdot \rceil$ ceiling and $\lfloor \cdot \rfloor$ floor functions, and for a single row block,

$$\mathrm{HBM}(I) = \Theta(B_N d + md + m + B_N).$$

There are $T_N = \lceil n/B_N \rceil$ row blocks, so total HBM accesses are

$$\Theta\!\Big(T_N(B_N d + md + m + B_N)\Big) = \Theta\!\Big(nd + \frac{n}{B_N}(md + m) + n\Big).$$

From the SRAM fit condition

$$B_M d + B_N d + (B_M + 2B_N) \lesssim M,$$

we obtain

$$(d+1)B_M + (d+2)B_N \lesssim M. \tag{$\star$}$$

Hence $(d+2)B_N \lesssim M$ and therefore $B_N = O(M/d)$.

Conversely, to maximize $B_N$ under $(\star)$ we minimize $B_M$. Taking $B_M := 1$ and

$$B_N := \left\lfloor \frac{M - (d+1)}{d+2} \right\rfloor$$

(which is feasible whenever the tiling constraint is satisfiable) yields $B_N = \Omega(M/d)$. Therefore the largest feasible row tile size satisfies $B_N = \Theta(M/d)$.

In the regime $d \leq M \leq \min\{n, m\}d$ this yields

$$\Theta\!\Big(nd + \frac{n}{M/d} md\Big) = \Theta\!\Big(nd + \frac{nmd^2}{M}\Big),$$

with the $\frac{n}{B_N}m$ term absorbed since $d \geq 1$. If $M \geq \min\{n, m\}d$, we can fit the smaller of $Q$ or $K$ entirely in SRAM and stream the other once, so the IO collapses to $\Theta(nd + md)$. $\qquad\square$

In addition, we can show there is no uniform asymptotic improvement over all $M$. This follows by the same endpoint argument as FlashAttention Proposition 3: when $M = \Theta(\min\{n, m\}d)$, we have $\frac{nmd^2}{M} = \Theta(\max\{n, m\}d) = \Theta(nd + md)$. But any exact algorithm must at least read the inputs once, which already costs $\Omega(nd+md)$ HBM accesses. Therefore no algorithm can achieve $o(nmd^2/M)$ HBM accesses for all $M$ in $[d, \min\{n, m\}d]$.

## D.3. Proof of Online LSE Correctness

*Proof.* We prove correctness for a single row; the vectorized (row-wise) statement used in Algorithm 1 follows by applying the same argument independently to each row.

Fix a row and let $\{\mathbf{x}_j\}_{j=1}^m$ denote its full logits, i.e. $b\mathbf{x}_j = S_{ij}$ for that row. Partition the column indices into $K$ blocks $J_1, \ldots, J_K$ (the tiles streamed by the algorithm). For each block $k$, define the block maximum

$$\tilde{\mathbf{m}}^{(k)} := \max_{j \in J_k} \mathbf{x}_j.$$

The online LSE maintains a running maximum $\mathbf{m}^{(k)}$ and a running normalized sum $\mathbf{s}^{(k)}$ via

$$\mathbf{m}^{(k)} := \max\big(\mathbf{m}^{(k-1)}, \tilde{\mathbf{m}}^{(k)}\big),$$

$$\mathbf{s}^{(k)} := e^{\mathbf{m}^{(k-1)} - \mathbf{m}^{(k)}} \mathbf{s}^{(k-1)} + \sum_{j \in J_k} e^{\mathbf{x}_j - \mathbf{m}^{(k)}},$$

initialized with $\mathbf{m}^{(0)} = -\infty$ and $\mathbf{s}^{(0)} = 0$. We claim the following invariant holds for every $k \in \{1, \ldots, K\}$:

$$\mathbf{s}^{(k)} = \sum_{j \in J_1 \cup \cdots \cup J_k} \exp\big(\mathbf{x}_j - \mathbf{m}^{(k)}\big). \tag{24}$$

In the base case $k = 1$, since $\mathbf{m}^{(1)} = \tilde{\mathbf{m}}^{(1)} = \max_{j \in J_1} \mathbf{x}_j$ and $\mathbf{s}^{(0)} = 0$, the update gives $\mathbf{s}^{(1)} = \sum_{j \in J_1} e^{\mathbf{x}_j - \mathbf{m}^{(1)}}$, which is exactly (24) for $k = 1$.

Assume (24) holds for $k - 1$. Then

$$\mathbf{s}^{(k)} = e^{\mathbf{m}^{(k-1)} - \mathbf{m}^{(k)}} \mathbf{s}^{(k-1)} + \sum_{j \in J_k} e^{\mathbf{x}_j - \mathbf{m}^{(k)}}$$

$$= e^{\mathbf{m}^{(k-1)} - \mathbf{m}^{(k)}} \sum_{j \in J_1 \cup \cdots \cup J_{k-1}} e^{\mathbf{x}_j - \mathbf{m}^{(k-1)}} + \sum_{j \in J_k} e^{\mathbf{x}_j - \mathbf{m}^{(k)}}$$

$$= \sum_{j \in J_1 \cup \cdots \cup J_{k-1}} e^{\mathbf{x}_j - \mathbf{m}^{(k)}} + \sum_{j \in J_k} e^{\mathbf{x}_j - \mathbf{m}^{(k)}} = \sum_{j \in J_1 \cup \cdots \cup J_k} e^{\mathbf{x}_j - \mathbf{m}^{(k)}},$$

proving the invariant for $k$.

At the end of the stream ($k = K$), we have

$$\mathbf{m}^{(K)} = \max_{1 \leq j \leq m} \mathbf{x}_j, \qquad \mathbf{s}^{(K)} = \sum_{j=1}^m e^{\mathbf{x}_j - \mathbf{m}^{(K)}}.$$

Therefore,

$$\mathbf{m}^{(K)} + \log \mathbf{s}^{(K)} = \mathbf{m}^{(K)} + \log\left(\sum_{j=1}^K e^{\mathbf{x}_j - \mathbf{m}^{(K)}}\right) = \log\left(\sum_{j=1}^m e^{\mathbf{x}_j}\right) = \mathrm{LSE}(\mathbf{x}).$$

Applying this row-wise yields $\mathrm{LSE}_{\mathrm{row}}(S) = m + \log s$, and hence the algorithm computes $\hat{\mathbf{f}} = -\varepsilon \, \mathrm{LSE}_{\mathrm{row}}(S_X(\hat{\mathbf{g}}))$ as claimed. $\square$

## E. Proof in Section 3.2

### E.1. Proof of Proposition 3

*Proof.* Fix shifted potentials $(\hat{\mathbf{f}}, \hat{\mathbf{g}})$ and define

$$P_{ij} := P_{ij}(\hat{\mathbf{f}}, \hat{\mathbf{g}}) = a_i b_j \exp\left(\frac{\hat{f}_i + \hat{g}_j + (QK^\top)_{ij}}{\varepsilon}\right).$$

Recall $\boldsymbol{\delta} = \varepsilon \log \mathbf{b}$ and $\boldsymbol{\gamma} = \varepsilon \log \mathbf{a}$, so that $b_j = \exp(\delta_j/\varepsilon)$ and $a_i = \exp(\gamma_i/\varepsilon)$.

By definition,

$$S_X(\hat{\mathbf{g}})_{ij} = \frac{(QK^\top)_{ij} + \hat{g}_j + \delta_j}{\varepsilon}, \qquad \exp\big(S_X(\hat{\mathbf{g}})_{ij}\big) = \exp\left(\frac{(QK^\top)_{ij} + \hat{g}_j + \delta_j}{\varepsilon}\right).$$

The row-update definition $\hat{\mathbf{f}}^+ = -\varepsilon \operatorname{LSE}_{\text{row}}(S_X(\hat{\mathbf{g}}))$ means, entrywise,

$$\hat{f}_i^+ = -\varepsilon \log \sum_{j=1}^{m} \exp(S_X(\hat{\mathbf{g}})_{ij}) \quad \Longrightarrow \quad \sum_{j=1}^{m} \exp(S_X(\hat{\mathbf{g}})_{ij}) = \exp\left(-\frac{\hat{f}_i^+}{\varepsilon}\right).$$

Now rewrite $P_{ij}$ as

$$P_{ij} = a_i \exp\left(\frac{\hat{f}_i}{\varepsilon}\right) \exp\left(\frac{(QK^\top)_{ij} + \hat{g}_j + \delta_j}{\varepsilon}\right) = a_i \exp\left(\frac{\hat{f}_i}{\varepsilon}\right) \exp(S_X(\hat{\mathbf{g}})_{ij}).$$

Therefore the row-mass vector $r = P\mathbb{1}_m$ satisfies

$$r_i = \sum_{j=1}^{m} P_{ij} = a_i \exp\left(\frac{\hat{f}_i}{\varepsilon}\right) \sum_{j=1}^{m} \exp(S_X(\hat{\mathbf{g}})_{ij}) = a_i \exp\left(\frac{\hat{f}_i - \hat{f}_i^+}{\varepsilon}\right),$$

which is the claimed expression for $r$. Moreover,

$$\frac{P_{ij}}{r_i} = \frac{\exp(S_X(\hat{\mathbf{g}})_{ij})}{\sum_{j'=1}^{m} \exp(S_X(\hat{\mathbf{g}})_{ij'})} = \operatorname{Softmax}(S_X(\hat{\mathbf{g}}))_{ij},$$

where $\operatorname{Softmax}(\cdot)$ is applied row-wise. Hence, in matrix form,

$$P = \operatorname{diag}(r) \operatorname{Softmax}(S_X(\hat{\mathbf{g}})),$$

and multiplying by any $V \in \mathbb{R}^{m \times p}$ gives

$$PV = \operatorname{diag}(r) \operatorname{Softmax}(S_X(\hat{\mathbf{g}})) V.$$

The column form $P^\top U = \operatorname{diag}(c) \operatorname{Softmax}(S_Y(\hat{\mathbf{f}})) U$ is shown similarly.

If $(\hat{\mathbf{f}}^*, \hat{\mathbf{g}}^*)$ satisfies the Sinkhorn fixed-point relations, then $\hat{\mathbf{f}}^* = \hat{\mathbf{f}}^{*+}$ and $\hat{\mathbf{g}}^* = \hat{\mathbf{g}}^{*+}$, so $r = \mathbf{a} \odot \exp((\hat{\mathbf{f}}^* - \hat{\mathbf{f}}^{*+})/\varepsilon) = \mathbf{a}$ and $c = \mathbf{b} \odot \exp((\hat{\mathbf{g}}^* - \hat{\mathbf{g}}^{*+})/\varepsilon) = \mathbf{b}$. Hence $P(\hat{\mathbf{f}}^*, \hat{\mathbf{g}}^*)$ has the prescribed marginals and coincides with the coupling induced by the optimal dual potentials, i.e., it recovers the optimal transport matrix. $\square$

### E.2. Proof of Corollary 4

*Proof.* By Proposition 3 (row form) with $V = Y$, we have

$$P(\hat{\mathbf{f}}^*, \hat{\mathbf{g}}^*) Y = \operatorname{diag}(r) \operatorname{Softmax}_{\text{row}}(S_X^*) Y.$$

At a Sinkhorn fixed point, the transport matrix has the prescribed row marginal, i.e. $r = \mathbf{a}$, and $P(\hat{\mathbf{f}}^*, \hat{\mathbf{g}}^*) = P^*$. Therefore

$$P^* Y = \operatorname{diag}(\mathbf{a}) \operatorname{Softmax}_{\text{row}}(S_X^*) Y, \qquad T_\varepsilon(X) = \operatorname{diag}(\mathbf{a})^{-1} P^* Y = \operatorname{Softmax}_{\text{row}}(S_X^*) Y,$$

which proves the attention form of the barycentric projection.

Finally, for squared Euclidean cost the gradient identity $\nabla_{\mathbf{x}_i} \operatorname{OT}_\varepsilon(\boldsymbol{\mu}, \boldsymbol{\nu}) = 2a_i(\mathbf{x}_i - T_\varepsilon(\mathbf{x}_i))$ holds. Stacking these $n$ row-gradients gives

$$\nabla_X \operatorname{OT}_\varepsilon(\boldsymbol{\mu}, \boldsymbol{\nu}) = 2\operatorname{diag}(\mathbf{a})(X - T_\varepsilon(X)) = 2\operatorname{diag}(\mathbf{a})(X - \operatorname{Softmax}_{\text{row}}(S_X^*)Y),$$

which is (17). $\square$

## F. Streaming Hessian-Vector Products Derivation and Proof of Theorem 5

Fix $A \in \mathbb{R}^{n \times d}$. Define the HVP $G := \mathcal{T} A \in \mathbb{R}^{n \times d}$ entrywise by $G_{kt} := \sum_{s=1}^{n} \sum_{l=1}^{d} \mathcal{T}_{ktsl} A_{sl}$.

Recall the Hessian decomposition $\mathcal{T} = \frac{1}{\varepsilon} \mathcal{R}^{\top} H^{*\dagger} \mathcal{R} + \mathcal{E}$ in Equation (6). Then the HVP splits into an implicit term (through $H^{*\dagger}$) and an explicit term:

$$\mathcal{T} A = \frac{1}{\varepsilon} \mathcal{R}^{\top} \mathbf{w} + \mathcal{E} A, \qquad \mathbf{w} := H^{*\dagger}(\mathcal{R} A), \tag{25}$$

where $\mathcal{R} A \in \mathbb{R}^{n+m}$ is the contraction $(\mathcal{R} A)_i = \sum_{s,l} \mathcal{R}_{isl} A_{sl}$, and $\mathbf{w}$ is the minimum-norm solution of the consistent linear system

$$H^{*} \mathbf{w} = \mathcal{R} A. \tag{26}$$

The matrix $H^{*}$ is symmetric positive semidefinite with a simple zero eigenvalue under strictly positive couplings $P_{kj}^{*} > 0$, hence the need for $H^{*\dagger}$; see (Li et al., 2025b).

Let $\odot$ denote the Hadamard product and $\mathbb{1}_d$ the all-ones vector in $\mathbb{R}^d$. Define the rowwise dot-products

$$\langle X, A \rangle := (X \odot A) \mathbb{1}_d \in \mathbb{R}^n.$$

### F.1. Explicit term $\mathcal{E} A$

Since $\mathcal{E}$ is block-diagonal across source points, $(\mathcal{E} A)_{kt}$ depends only on the $k$-th row $A_{k:}$:

$$(\mathcal{E} A)_{kt} = \sum_{l=1}^{d} \mathcal{E}_{ktkl} A_{kl} = 2a_k A_{kt} - \frac{4}{\varepsilon} \sum_{j=1}^{m} \sum_{l=1}^{d} P_{kj}^{*} (x_{kt} - y_{jt})(x_{kl} - y_{jl}) A_{kl}.$$

Introduce

$$\mathbf{u} := \langle X, A \rangle \in \mathbb{R}^n, \qquad \mathbf{u}_P := \langle PY, A \rangle \in \mathbb{R}^n, \qquad W := A Y^{\top} \in \mathbb{R}^{n \times m} \quad (W_{kj} = \langle A_k, y_j \rangle).$$

Then the contraction $\mathcal{E} A \in \mathbb{R}^{n \times d}$ admits the streaming-friendly decomposition

$$\mathcal{E} A = B_1 - \frac{4}{\varepsilon} (B_2 - B_3 - B_4 + B_5), \tag{27}$$

with

$$\begin{aligned}
B_1 &= 2 \operatorname{diag}(\mathbf{a}) A, \\
B_2 &= \operatorname{diag}(\mathbf{a} \odot \mathbf{u}) X, \\
B_3 &= \operatorname{diag}(\mathbf{u}) (PY), \\
B_4 &= \operatorname{diag}(\mathbf{u}_P) X, \\
B_5 &= (P^{*} \odot W) Y.
\end{aligned} \tag{28}$$

Here $B_3, B_4$ use $PY$ and rowwise dot-products, while $B_5$ is a Hadamard-weighted transport $(P^{*} \odot (A Y^{\top}))Y$; all can be implemented via streaming $P^{*}$-vector products without materializing $P^{*}$ (or $W$).

### F.2. Implicit term $\mathcal{R}^{\top} \mathbf{w}$

Write $\mathbf{w} = \begin{pmatrix} \mathbf{w}_1 \\ \mathbf{w}_2 \end{pmatrix}$ with $\mathbf{w}_1 \in \mathbb{R}^n$, $\mathbf{w}_2 \in \mathbb{R}^m$. The implicit term in Equation (25) is evaluated in three steps.

**Step 1: build the right-hand side $\mathbf{r} := \mathcal{R} A$.** Using the definition of $\mathcal{R}$ with $\mathcal{B}_{ktj} = 2(x_{kt} - y_{jt}) P_{kj}^{*}$, the contraction $\mathbf{r} = \begin{pmatrix} \mathbf{r}_1 \\ \mathbf{r}_2 \end{pmatrix} \in \mathbb{R}^{n+m}$ has blocks

$$\boxed{\begin{aligned}
\mathbf{r}_1 &= 2\Big(\mathbf{a} \odot \mathbf{u} - \mathbf{u}_P\Big) \in \mathbb{R}^n, \\
\mathbf{r}_2 &= 2\Big(P^{*\top} \mathbf{u} - \langle P^{*\top} A, Y \rangle\Big) \in \mathbb{R}^m.
\end{aligned}} \tag{29}$$

Every term here is a vector product with $P^{*}$ or $P^{*\top}$, plus elementwise operations.

**Step 2: solve $H^*\mathbf{w} = \mathbf{r}$ via a (damped) Schur complement.** Write $\mathbf{w} = \begin{pmatrix} \mathbf{w}_1 \\ \mathbf{w}_2 \end{pmatrix}$ and $\mathbf{r} = \begin{pmatrix} \mathbf{r}_1 \\ \mathbf{r}_2 \end{pmatrix}$, so that

$$\begin{pmatrix} \mathrm{diag}(\mathbf{a}) & P^* \\ (P^*)^\top & \mathrm{diag}(\mathbf{b}) \end{pmatrix} \begin{pmatrix} \mathbf{w}_1 \\ \mathbf{w}_2 \end{pmatrix} = \begin{pmatrix} \mathbf{r}_1 \\ \mathbf{r}_2 \end{pmatrix}.$$

When $P^* > 0$, $H^*$ is symmetric positive semidefinite with a one-dimensional nullspace being $H^*[\mathbb{1}_n; -\mathbb{1}_m] = \mathbf{0}$. Eliminating $\mathbf{w}_1$ yields the Schur-complement system:

$$\underbrace{\left( \mathrm{diag}(\mathbf{b}) - (P^*)^\top \mathrm{diag}(\mathbf{a})^{-1} P^* \right)}_{=:S \in \mathbb{R}^{m \times m}} \mathbf{w}_2 = \mathbf{r}_2 - (P^*)^\top \mathrm{diag}(\mathbf{a})^{-1} \mathbf{r}_1, \qquad \mathbf{w}_1 = \mathrm{diag}(\mathbf{a})^{-1}(\mathbf{r}_1 - P^* \mathbf{w}_2). \tag{30}$$

Moreover, $S \succeq 0$ and $S\mathbb{1}_m = 0$ (the same mass-shift mode), so one can either (i) project onto $\mathbb{1}_m^\perp$ and run CG there, or (ii) apply Tikhonov damping $S_\tau := S + \tau I$ to obtain an SPD system. In this work, we use damping and solve

$$S_\tau \mathbf{w}_2 = \mathbf{r}_2 - (P^*)^\top \mathrm{diag}(\mathbf{a})^{-1} \mathbf{r}_1, \qquad S_\tau := \left( \mathrm{diag}(\mathbf{b}) - (P^*)^\top \mathrm{diag}(\mathbf{a})^{-1} P^* \right) + \tau I.$$

For a target relative residual $\eta$, CG satisfies the standard iteration bound $K_{CG} = O\left(\sqrt{\kappa(S_\tau)} \log(1/\eta)\right)$ for SPD systems (Golub & Loan, 2013). Consequently, the transport-dominated HVP cost scales as

$$O\left( nm\left(d + \sqrt{\kappa(S_\tau)} \log(1/\eta)\right) \right).$$

As $\varepsilon \downarrow 0$, conditioning can deteriorate: Li et al. (2025b) show that the associated sensitivity matrix has a simple zero eigenvalue and can have a smallest positive eigenvalue as small as $O(e^{-1/\varepsilon})$, implying potentially large effective condition numbers at small $\varepsilon$. Damping enforces $\lambda_{\min}(S_\tau) \geq \tau$, hence $\kappa(S_\tau) \leq (\lambda_{\max}(S) + \tau)/\tau$, which stabilizes $K$. Finally, in the strictly SPD setting the Schur complement is not worse conditioned than the full system: if $H \succ 0$, then $S(H)^{-1}$ is a principal submatrix of $H^{-1}$ and eigenvalue interlacing implies $\kappa_2(S(H)) \leq \kappa_2(H)$ (Golub & Loan, 2013).

**Step 3: apply $\mathcal{R}^\top$ to $\mathbf{w}$.** Given $\mathbf{w}_1, \mathbf{w}_2$, the contraction $\mathcal{R}^\top \mathbf{w} \in \mathbb{R}^{n \times d}$ simplifies to

$$\boxed{\mathcal{R}^\top \mathbf{w} = 2\left( \mathrm{diag}(\mathbf{a} \odot \mathbf{w}_1) X - \mathrm{diag}(\mathbf{w}_1)(PY) + \mathrm{diag}(P^*\mathbf{w}_2) X - P^*(\mathrm{diag}(\mathbf{w}_2) Y) \right).} \tag{31}$$

Again, only transport-like products $P^*v$, $P^{*\top}u$, $PY$, and $P^*(\mathrm{diag}(\mathbf{w}_2)Y)$ appear.

**Summary.** Combining Equation (25), Equation (27)–Equation (28), and Equation (29)–Equation (31), the full HVP reduces to (i) a small number of $P^*/P^{*\top}$ transport vectors, (ii) transport matrices $PY$ and $P^*(\mathrm{diag}(\mathbf{w}_2)Y)$, and (iii) one Hadamard-weighted transport $(P^* \odot (AY^\top))Y$, all of which admit FlashAttention-style streaming implementations.

**Remark 9** (Sign structure). *The implicit component $\frac{1}{\varepsilon} \mathcal{R}^\top H^{*\dagger} \mathcal{R}$ is positive semidefinite because $H^{*\dagger} \succeq 0$ and it appears as $\mathcal{R}^\top(\cdot)\mathcal{R}$. The explicit component $\mathcal{E}$ is block-diagonal with blocks $2a_k \mathbb{I}_d - \frac{4}{\varepsilon} \sum_j P_{kj}^*(\mathbf{x}_k - \mathbf{y}_j)^\top(\mathbf{x}_k - \mathbf{y}_j)$, i.e., $2a_k \mathbb{I}_d$ minus a PSD term scaled by $4/\varepsilon$. Hence $\mathcal{E}$, and therefore $\mathcal{T}$ is generally indefinite.*

### F.3. Computational complexity of one HVP

We count operations that traverse the dense pairwise interactions. Since $P^*$ is not materialized, each transport application is implemented from potentials by streaming tiles, forming biased dot-product scores, and applying online normalization (Dao et al., 2022; Milakov & Gimelshein, 2018).

Let a streamed transport application be $P^*V$ or $(P^*)^\top U$ with $V \in \mathbb{R}^{m \times p}$, $U \in \mathbb{R}^{n \times p}$. One streamed pass costs

$$\Theta\left( nm(d + p) \right) \text{ flops}$$

because it forms scores with $\Theta(nmd)$ dot products and accumulates values with $\Theta(nmp)$ multiply-adds (the remaining exp/LSE bookkeeping is $O(nm)$). In particular: (i) a *transport–vector* ($p = 1$) costs $\Theta(nm(d + 1)) = \Theta(nmd)$ flops; (ii) a *transport–matrix* with $p = d$ costs $\Theta(nm(2d)) = \Theta(nmd)$ flops. A *Hadamard-weighted transport* $(P^* \odot (AY^\top))Y$ is also $\Theta(nmd)$ flops when fused, noting it requires an additional streamed dot product $AY^\top$ but remains the same order.

$\mathcal{E} A$. Using $\mathcal{E} A = B_1 - \frac{4}{\varepsilon}(B_2 - B_3 - B_4 + B_5)$, the dense operations are: (i) one transport–matrix $P^* Y$, which is cacheable across repeated HVPs at fixed $(\hat{\mathbf{f}}^*, \hat{\mathbf{g}}^*)$, and (ii) one Hadamard-weighted transport $B_5 = (P^* \odot (AY^\top))Y$. All remaining terms are rowwise scalings and dot-products, costing $O((n+m)d)$. Hence, flops$(\mathcal{E} A) = \Theta(nmd)$.

$\mathcal{R}^\top \mathbf{w}$. Forming $\mathbf{r} = \mathcal{R} A$ requires a transport–matrix $(P^*)^\top A$ plus a constant number of transport–vector products, hence $\Theta(nmd)$ flops. Eliminating $\mathbf{w}_1$ yields the Schur system $S \mathbf{w}_2 = \text{rhs}$, which we solve by CG. Each CG iteration applies $S$ once, i.e., one $P^*$ and one $(P^*)^\top$ transport–vector application, so $K$ iterations cost $\Theta(K_{CG} nmd)$ flops. Back-substitution and the final application of $\mathcal{R}^\top \mathbf{w}$ contribute an additional $\Theta(nmd)$ flops. Therefore,

$$\text{flops}(\mathcal{R}^\top \mathbf{w}) = \Theta\big((2K_{CG} + O(1))\, nmd\big).$$

**Total.** Combining the above,

$$\text{flops}(\mathcal{T} A) = \Theta\big((2K_{CG} + O(1))\, nmd\big) = O\big((K_{CG} + 1)\, nmd\big),$$

up to constant factors. The working memory stores $X, Y, A$ and a constant number of $n \times d$ or $m \times d$ buffers (e.g., $P^* Y$, $(P^*)^\top A$) and $O(n+m)$ vectors, hence in total $O((n+m)d)$ memory without materializing $P^*$.

# G. Implementation Details

## G.1. Algorithms

A direct implementation of (10) would invoke separate kernels to (i) compute the score tiles $QK^\top$, (ii) add biases, and (iii) perform a row-wise LSE reduction, typically incurring extra HBM traffic and kernel-launch overhead due to intermediate score materialization. Instead, we fuse these stages into a single streaming kernel: each thread block loads a tile of $Q$ and iterates over tiles of $K$, updates the running row-wise max and rescaled sum-exp statistics $(\mathbf{m}_I, \mathbf{s}_I)$ (online LSE), and emits $\hat{\mathbf{f}}_I$ (or $\hat{\mathbf{g}}_I$) once per row block. For transport matrix application, we fuse the corresponding softmax-weighted value accumulation into the same pass, analogous to fused attention kernels that compute $\text{Softmax}(QK^\top)V$ without storing $QK^\top$ or $\text{Softmax}(QK^\top)$ explicitly.

In addition to Algorithm 1 and Algorithm 2, we will have three more algorithms: streaming $\hat{\mathbf{g}}$-update in algorithm 3; apply adjoint transport from potentials $P^\top V$ in algorithm 4 and Hadamard-weighted transport from potentials $(P \odot W) V$ in algorithm 5.

To maintain mathematical consistency for the current transport matrix $P$ (and hence for early-stopped Sinkhorn), our implementation uses induced marginals throughout:

- **Gradient.** For squared Euclidean cost,

$$\nabla_X \text{OT}_\varepsilon(\boldsymbol{\mu}, \boldsymbol{\nu}) = 2\big(\text{diag}(\hat{\mathbf{a}})\, X - PY\big),$$

  i.e., the row-scaling term uses $\hat{\mathbf{a}} = P\mathbb{1}_m$ which is available for free from the same streaming normalizations used to evaluate $PY$.

- **HVP.** The sensitivity matrix and its Schur complement are constructed using $\hat{\mathbf{a}}, \hat{\mathbf{b}}$:

$$H(P) = \begin{pmatrix} \text{diag}(\hat{\mathbf{a}}) & P \\ P^\top & \text{diag}(\hat{\mathbf{b}}) \end{pmatrix}, \qquad S(P) = \text{diag}(\hat{\mathbf{b}}) - P^\top \text{diag}(\hat{\mathbf{a}})^{-1} P,$$

This ensures that the gradient and HVP are consistent objects derived from the same transport matrix $P$, rather than the optimal coupling $P^*$ with the target marginals $(\mathbf{a}, \mathbf{b})$. When Sinkhorn is fully converged, $\hat{\mathbf{a}} = \mathbf{a}$ and $\hat{\mathbf{b}} = \mathbf{b}$, and the distinction disappears.

## G.2. Loop order

Both FlashAttention and FlashSinkhorn rely on the same online-softmax/online-LSE recurrence (running row-wise max and rescaled exp-sums) to avoid materializing the score matrix in HBM. The difference is which operand is held stationary

---

**Algorithm 3** FlashSinkhorn: streaming $\hat{\mathbf{g}}$-update

---

1: **Input:** $X \in \mathbb{R}^{n \times d}, Y \in \mathbb{R}^{m \times d}, \hat{\mathbf{f}} \in \mathbb{R}^n, \mathbf{a} \in \Delta^n$
2: **Output:** $\hat{\mathbf{g}} \in \mathbb{R}^m$
3: $Q \leftarrow \sqrt{2}X, K \leftarrow \sqrt{2}Y, \boldsymbol{\gamma} \leftarrow \varepsilon \log \mathbf{a}$              *// scale on load*
4: **for** each row block $J$ of size $B_M$ **do**
5:     Load $K_J$ to on-chip SRAM
6:     $\mathbf{m}_J \leftarrow -\infty, \mathbf{s}_J \leftarrow 0$              *// running max / sumexp*
7:     **for** each column block $I$ of size $B_N$ **do**
8:         Load $Q_I, \hat{\mathbf{f}}_I$ and $\boldsymbol{\gamma}_I$ to on-chip SRAM
9:         $S \leftarrow (K_J Q_I^\top + \hat{\mathbf{f}}_I + \boldsymbol{\gamma}_I)/\varepsilon$              *// scores + bias*
10:         $\tilde{\mathbf{m}} \leftarrow \text{rowmax}(S)$              *// max over col*
11:         $\mathbf{m}_{\text{new}} \leftarrow \max(\mathbf{m}_J, \tilde{\mathbf{m}})$              *// update elementwise*
12:         $\mathbf{s}_J \leftarrow e^{\mathbf{m}_J - \mathbf{m}_{\text{new}}} \odot \mathbf{s}_J + \text{rowsum}\left(e^{S - \mathbf{m}_{\text{new}}}\right)$
13:         $\mathbf{m}_J \leftarrow \mathbf{m}_{\text{new}}$
14:     **end for**
15:     $\hat{\mathbf{g}}_J \leftarrow -\varepsilon \cdot \left(\mathbf{m}_J + \log \mathbf{s}_J\right)$              *// $\hat{\mathbf{g}} = -\varepsilon\,\text{LSE}_{\text{row}}$*
16:     Write $\hat{\mathbf{g}}_J$ to HBM
17: **end for**

---

in SRAM. FlashAttention Algorithm 1 chooses a $(K, V)$-stationary traversal: it loops over key/value blocks $j$ in the outer loop, loads $(K_j, V_j)$ into on-chip SRAM once, and then sweeps over all query blocks $Q_i$ to update $(O_i, \ell_i, m_i)$, writing these intermediate quantities back to HBM each pass. This ordering is motivated by IO: with the block-size choice $B_r$, the number of query blocks can be large, and streaming $(K, V)$ outermost avoids repeatedly reloading the same $(K, V)$ blocks from HBM.

In contrast, FlashSinkhorn's half-steps are fundamentally row-wise reduction. We therefore adopt a row-stationary traversal: fix a query row block $Q_I$ in SRAM, stream all key/value blocks $(K_J, V_J)$, and maintain the running statistics and accumulators $(\mathbf{m}_I, \mathbf{s}_I)$ (and $O_I$ when computing $PV$) on-chip until the row block is complete, then write out $\hat{\mathbf{f}}_I$ or $(PV)_I$ once. This strategy substantially reduces HBM traffic for intermediate accumulators and is particularly critical for OT transport-matrix/vector products, where the column dimension of $V$, whenever it is a matrix or a vector ,can be much larger than an attention head, making standard FlashAttention-style online normalization impractical. This row-stationary nesting aligns with the refinement adopted in FlashAttention-2 (Dao, 2024), which similarly moves the queries to the outer loop for the same row-wise efficiency reasons.

## H. Full Experiment Results

### H.1. Experimental Setup

All experiments were conducted on an NVIDIA A100-80GB GPU with CUDA 12.1. We use PyTorch 2.5.1+ with Triton 2.1+ for FlashSinkhorn. We benchmark against GeomLoss 0.2.6 (Feydy et al., 2019) with PyKeOps 2.3 backend and OTT-JAX 0.5.1 (Cuturi et al., 2022) with JAX 0.8.2.

### H.2. Synthetic Benchmarks

We benchmark on balanced optimal transport with uniform marginals ($\mathbf{a} = \mathbb{1}_n/n$, $\mathbf{b} = \mathbb{1}_m/m$) and squared Euclidean cost $C(\mathbf{x}, \mathbf{y}) = \|\mathbf{x} - \mathbf{y}\|_2^2$. Points are sampled uniformly from $[0, 1]^d$ with $n \in [5\text{k}, 50\text{k}]$, $d \in [4, 1024]$ for forward/backward benchmarks and $d \in [4, 512]$ for HVP examples. All methods use regularization $\varepsilon = 0.1$ and 10 Sinkhorn iterations for forward/backward benchmarks, or 100 Sinkhorn iterations for HVP benchmarks. For HVP, the conjugate gradient solver uses $K = 50$ fixed iterations with damping $\tau = 10^{-5}$ and no early stopping (convergence tolerance disabled for fair comparison).

For PyTorch and KeOps, we use CUDA events for precise GPU timing. For JAX, we use wall-clock time with `block_until_ready()` synchronization. All benchmarks include 10 warmup iterations before timing, with 50 repetitions for forward, 30 for backward, and 20 for HVP. We report mean timing across repetitions.

---

**Algorithm 4** Apply Adjoint Transport from Potentials, $P^\top V$

---

1: **Input:** $X \in \mathbb{R}^{n \times d}, Y \in \mathbb{R}^{m \times d}, \hat{\mathbf{f}} \in \mathbb{R}^n, \hat{\mathbf{g}} \in \mathbb{R}^m, \mathbf{a} \in \Delta^n, \mathbf{b} \in \Delta^m, V \in \mathbb{R}^{n \times p}$
2: **Output:** out $= P^\top V \in \mathbb{R}^{m \times p}$
3: $Q \leftarrow \sqrt{2}X, K \leftarrow \sqrt{2}Y, \boldsymbol{\gamma} \leftarrow \varepsilon \log \mathbf{a}$         *// scale on load*
4: **for** each row block $J$ of size $B_M$ **do**
5:     Load $K_J, \hat{\mathbf{g}}_J, \mathbf{b}_J$ to on-chip SRAM
6:     $\mathbf{m}_J \leftarrow -\infty, \mathbf{s}_J \leftarrow 0, O_J \leftarrow 0$
7:     **for** each column block $I$ of size $B_N$ **do**
8:         Load $Q_I, \hat{\mathbf{f}}_I, \boldsymbol{\gamma}_I, V_I$ to on-chip SRAM
9:         $S \leftarrow (K_J Q_I^\top + \hat{\mathbf{f}}_I + \boldsymbol{\gamma}_I)/\varepsilon$         *// score tile + bias*
10:         $\tilde{\mathbf{m}} \leftarrow \text{rowmax}(S)$         *// tile max*
11:         $\mathbf{m}_{\text{new}} \leftarrow \max(\mathbf{m}_J, \tilde{\mathbf{m}})$         *// update max row-wise*
12:         $\mathbf{s}_J \leftarrow e^{\mathbf{m}_J - \mathbf{m}_{\text{new}}} \odot \mathbf{s}_J + \text{rowsum}(e^{S - \mathbf{m}_{\text{new}}})$
13:         $O_J \leftarrow e^{\mathbf{m}_J - \mathbf{m}_{\text{new}}} \odot O_J + e^{S - \mathbf{m}_{\text{new}}} V_I$
14:         $\mathbf{m}_J \leftarrow \mathbf{m}_{\text{new}}$
15:     **end for**
16:     $\hat{\mathbf{g}}_J^+ \leftarrow -\varepsilon \cdot (\mathbf{m}_J + \log \mathbf{s}_J)$         *// one-step $\hat{\mathbf{g}}$-update*
17:     $\mathbf{c}_J \leftarrow \mathbf{b}_J \odot \exp\big((\hat{\mathbf{g}}_J - \hat{\mathbf{g}}_J^+)/\varepsilon\big)$         *// $\mathbf{c} = (P^\top \mathbf{1})_J$*
18:     out$_J \leftarrow \text{diag}(\mathbf{c}_J) \text{diag}(\mathbf{s}_J)^{-1} O_J$
19:     Write out$_J$ to HBM
20: **end for**

---

**Algorithm 5** Hadamard-Weighted Transport from Potentials, $(P \odot W) V$ with $W_{kj} = A_k B_j^\top$

---

1: **Input:** $X \in \mathbb{R}^{n \times d}, Y \in \mathbb{R}^{m \times d}, A \in \mathbb{R}^{n \times r}, B \in \mathbb{R}^{m \times r}, \hat{\mathbf{f}} \in \mathbb{R}^n, \hat{\mathbf{g}} \in \mathbb{R}^m, \mathbf{a} \in \Delta^n, \mathbf{b} \in \Delta^m, V \in \mathbb{R}^{m \times p}$
2: **Output:** out $= (P \odot W) V \in \mathbb{R}^{n \times p}$
3: $Q \leftarrow \sqrt{2}X, K \leftarrow \sqrt{2}Y, \boldsymbol{\delta} \leftarrow \varepsilon \log \mathbf{b}$         *// scale on load*
4: **for** each row block $I$ of size $B_N$ **do**
5:     Load $Q_I, A_I, \hat{\mathbf{f}}_I, \mathbf{a}_I$ to on-chip SRAM
6:     $\mathbf{m}_I \leftarrow -\infty, \mathbf{s}_I \leftarrow 0, O_I \leftarrow 0$         *// running max/sumexp, output accumulator*
7:     **for** each column block $J$ of size $B_M$ **do**
8:         Load $K_J, B_J, \hat{\mathbf{g}}_J, \boldsymbol{\delta}_J, V_J$ to on-chip SRAM
9:         $S \leftarrow (Q_I K_J^\top + \hat{\mathbf{g}}_J + \boldsymbol{\delta}_J)/\varepsilon$         *// score tile + bias*
10:         $W \leftarrow A_I B_J^\top$         *// Hadamard weights: $W_{kj} = A_k B_j^\top$*
11:         $\tilde{\mathbf{m}} \leftarrow \text{rowmax}(S)$
12:         $\mathbf{m}_{\text{new}} \leftarrow \max(\mathbf{m}_I, \tilde{\mathbf{m}})$
13:         $\mathbf{s}_I \leftarrow e^{\mathbf{m}_I - \mathbf{m}_{\text{new}}} \odot \mathbf{s}_I + \text{rowsum}(e^{S - \mathbf{m}_{\text{new}}})$
14:         $O_I \leftarrow e^{\mathbf{m}_I - \mathbf{m}_{\text{new}}} \odot O_I + (e^{S - \mathbf{m}_{\text{new}}} \odot W) V_J$
15:         $\mathbf{m}_I \leftarrow \mathbf{m}_{\text{new}}$
16:     **end for**
17:     $\hat{\mathbf{f}}_I^+ \leftarrow -\varepsilon \cdot (\mathbf{m}_I + \log \mathbf{s}_I)$         *// one-step $\hat{\mathbf{f}}$-update*
18:     $\mathbf{r}_I \leftarrow \mathbf{a}_I \odot \exp\big((\hat{\mathbf{f}}_I - \hat{\mathbf{f}}_I^+)/\varepsilon\big)$         *// $\mathbf{r} = (P\mathbf{1})_I$*
19:     out$_I \leftarrow \text{diag}(\mathbf{r}_I) \text{diag}(\mathbf{s}_I)^{-1} O_I$
20:     Write out$_I$ to HBM
21: **end for**

---

**Methods Compared.**

- **FlashSinkhorn**: Our method with symmetric (GeomLoss-style) and alternating (OTT-style) updates

- **KeOps**: GeomLoss with `backend='online'` — streaming $O(nd)$ memory

- **Tensorized**: GeomLoss with `backend='tensorized'` — dense $O(n^2)$ memory

- **OTT-JAX**: OTT-JAX with `use_danskin=True` for gradients

Another method **POT**: Python Optimal Transport toolbox (primarily for NumPy/SciPy CPU) (Flamary et al., 2021) is not benchmarked here since our evaluation targets GPU implementations. GeomLoss also provides `backend='multiscale'`, a fast octree-style multiscale routine intended for very large problems in low ambient dimension, but it is restricted to inputs in dimension $d \in \{1, 2, 3\}$; since our benchmarks focus on higher-dimensions, we do not include this mode, either.

A closely related line is Lindbäck et al. (2023), which proposes RDROT, a Douglas–Rachford splitting solver adapted to GPUs for regularized OT with a broad class of sparsity-promoting regularizers. However, our benchmarks focus on EOT, for which the optimal coupling is typically dense; notably, Lindbäck et al. (2023) explicitly excludes negative Shannon entropy from its target regularizer class. As a result, RDROT is not a like-for-like baseline for our setting, so we discuss it as complementary rather than including it in the timing comparisons.

**Benchmark Details** To isolate implementation efficiency from algorithmic differences, our default benchmark configuration is: (1) fixed iteration count with no early stopping; (2) $\varepsilon = 0.1$ with no epsilon-scaling; (3) TF32 precision enabled for forward and backward benchmarks, and strict FP32 for HVP benchmarks due to higher numerical sensitivity of second-order computations; (4) no debiasing for simpler comparison. These settings ensure all methods perform identical arithmetic operations. The low-$\varepsilon$ study in Section H.2.5 varies $\varepsilon \in \{0.10, 0.05, 0.01\}$ and the iteration budget, and additionally compares fp32 against an fp64 reference; the remaining settings are held fixed.

We discard the first 10 runs to exclude JIT compilation and cache warm-up, then measure 50 repetitions using the framework-appropriate timing protocol described above (CUDA events for PyTorch/KeOps; `block_until_ready()` for JAX). For Triton kernels, we run problem sizes from large to small to avoid autotuning cache pollution (Dao et al., 2022; Ringlein et al., 2025; Li et al., 2025a), which can cause up to 50% performance degradation if sizes are run in ascending order.

FlashSinkhorn and GeomLoss (same symmetric algorithm) differ by $< 0.1\%$ in loss values due to floating-point non-associativity in parallel reductions. FlashSinkhorn and OTT-JAX differ by $\sim 2\%$ at 10 iterations due to different initial $\mathbf{f}$ and $\mathbf{g}$, but converge to $< 0.1\%$ difference at 100 iterations.

### H.2.1. FLASHSINKHORN VS GEOMLOSS: PERFORMANCE ANALYSIS

Tables 8 and 9 show that KeOps speedup scales with $d$: At small $d$ ($\leq 16$), KeOps' lazy evaluation is competitive (1-7×). At large $d$ ($\geq 64$), FlashSinkhorn's fused kernels dominate (9-47×). Peak speedup occurs at $d = 512$ where FlashSinkhorn achieves 32-47× speedup. KeOps backward is extremely slow: FlashSinkhorn achieves 137-212× speedup at $d$=512-1024 because KeOps must recompute the kernel matrix for gradient computation, while FlashSinkhorn uses custom analytical gradient kernels.

Tables 10 and 11 show that FlashSinkhorn beats Tensorized at small-to-medium d: At $d \leq 256$, FlashSinkhorn is 2-12× faster because Tensorized's $O(n^2)$ cost matrix allocation dominates. Tensorized wins at $d = 1024$: FlashSinkhorn is 0.5-0.7× slower because the on-the-fly recomputation overhead exceeds Tensorized's memory bandwidth cost when $d$ is very large. The real advantage is memory: Even when Tensorized is faster ($d = 1024$), it uses 39-71× more memory and OOMs at $n \geq 30k$. FlashSinkhorn trades some speed at large $d$ for dramatically better scalability.

We profile FlashSinkhorn, GeomLoss KeOps, and GeomLoss Tensorized using NVIDIA Nsight Compute. Table 5 reports the **forward** pass ($n$=$m$=10000, $d$=64, 10 Sinkhorn iterations, A100-80GB). Table 7 reports the **forward+backward** pass ($n$=$m$=10000, $d$=128, 10 Sinkhorn iterations, A100-80GB). They reveal fundamentally different execution regimes.

**Forward: Tensorized is memory-bound despite high SM utilization.** Tensorized achieves 98% SM utilization and 89% occupancy—metrics that would typically indicate efficient GPU usage. However, 79% of cycles are spent stalled on

*Table 5.* NCU profiling of the forward pass ($n=m$=10000, $d$=64, 10 Sinkhorn iterations, A100-80GB). **Note:** Working set (5 MB) fits in A100 L2 cache (40 MB); HBM traffic reflects L2-resident execution.

| Metric | Tensor. | KeOps | Flash |
|---|---|---|---|
| *Performance* | | | |
| Runtime (ms) | 54.0 | 125.5 | **8.2** |
| *Memory* | | | |
| HBM Read | 59 GB | 140 MB | 79 MB |
| HBM Write | 39 GB | 0 | 0 |
| Shared Mem Traffic (GB) | 6 | 23 | 53 |
| L1 Hit (%) | 18 | 82 | 8 |
| L2 Hit (%) | 56 | 97 | 99 |
| *Compute* | | | |
| SM Util. (%) | 98 | 49 | 74 |
| Occupancy (%) | 89 | 9 | 11 |
| Instructions (B) | 10 | 16 | 7 |
| *Stalls* | | | |
| Memory (%) | 79 | 8 | 3 |
| Math (%) | 2 | 8 | 6 |
| *Launch* | | | |
| Regs/Thread | 16 | 96 | 255 |
| Bottleneck | Memory | Compute | Compute |

*Table 6.* Per-kernel and tensor-pipe breakdown of the forward pass ($n=m$=10000, $d$=64, 10 Sinkhorn iterations, A100-80GB). KeOps issues 96 `GpuConv1D` reductions and 758 elementwise auxiliary kernels per evaluation; FlashSinkhorn fuses each Sinkhorn update into a small number of streaming Triton kernels ($\sim$5 per iteration, 48 Sinkhorn-related in total) and routes the squared-Euclidean interaction through tiled `tl.dot` on the tensor pipeline rather than elementwise CUDA-core operations.

| Metric | KeOps | Flash | Ratio |
|---|---|---|---|
| Total kernel launches | 854 | **130** | 6.6× fewer |
| Tensor-pipe instructions (M) | 3.5 | **10.1** | 2.9× more |

memory. The root cause is the $O(n^2)$ memory footprint: Tensorized transfers 98 GB through HBM, repeatedly reading and writing the $n \times n$ cost matrix across iterations. High SM utilization here is misleading, the SMs are active but stalled, waiting for data.

**Forward: FlashSinkhorn and KeOps are compute-bound, but FlashSinkhorn is 3× more efficient.** Both online methods shift the bottleneck from memory to compute. However, FlashSinkhorn executes 2.3× fewer instructions (7B vs 16B) due to fused kernels that perform distance computation, log-sum-exp reduction, and potential updates in a single pass. Two further counters (Table 6) account for the remaining gap: FlashSinkhorn issues 6.6× fewer kernel launches (130 vs. 854), avoiding dispatch overhead and intermediate global-memory traffic between KeOps's 96 `GpuConv1D` reductions and 758 elementwise auxiliaries, and routes 2.9× more compute through the tensor pipeline (10.1 M vs. 3.5 M tensor-pipe instructions) by rewriting the squared-Euclidean interaction as tiled `tl.dot` dot products instead of CUDA-core elementwise operations. The counters are complementary rather than multiplicative — instruction count reflects total work, SM utilization and tensor-pipe activity reflect compute efficiency, and launch count reflects fragmentation — jointly explaining the 15.3× KeOps gap.

**Forward: FlashSinkhorn trades occupancy for fusion.** FlashSinkhorn uses 255 registers per thread—the GPU maximum—enabling complex fused operations within a single kernel launch. This limits occupancy to 11% (vs Tensorized's 89%), but the reduced memory traffic more than compensates. FlashSinkhorn transfers only 0.08 GB through HBM, a **1225× reduction** versus Tensorized, by streaming intermediate results through 53 GB of shared memory. Memory stalls drop from 79% to just 3%. Streaming methods show zero HBM write because the $O(nd)$ working set fits entirely in L2 cache, whereas Tensorized must write 39GB to materialize the cost matrix.

**Backward: The memory bottleneck intensifies for Tensorized.** The backward pass requires computing gradients with respect to both point clouds, amplifying Tensorized's memory problem. HBM traffic reaches 109 GB for for-

*Table 7.* NCU profiling of the forward and backward pass ($n=m$=10000, $d$=128, 10 Sinkhorn iterations, A100-80GB).

| Metric | Tensor. | KeOps | Flash |
|---|---|---|---|
| *Performance* | | | |
| Runtime (ms) | 67.6 | 197.0 | **19.2** |
| *Memory* | | | |
| HBM Read | 65 GB | 250 MB | 199 MB |
| HBM Write | 44 GB | 4 MB | 48 MB |
| Shared Mem Traffic (GB) | 14 | 330 | 354 |
| L1 Hit (%) | 17 | 85 | 17 |
| L2 Hit (%) | 56 | 98 | 100 |
| *Compute* | | | |
| SM Util. (%) | 30 | 38 | 38 |
| Occupancy (%) | 84 | 5 | 11 |
| Instructions (B) | 11 | 35 | 18 |
| *Stalls* | | | |
| Memory (%) | 68.2 | 0.8 | 0.5 |
| Math (%) | 2.0 | 0.0 | 0.5 |
| *Launch* | | | |
| Regs/Thread | 122 | 255 | 255 |
| Bottleneck | Memory | Compute | Compute |

*Table 8.* Speedup of FlashSinkhorn over KeOps (forward pass only, averaged over 50 runs). OOT indicates exceeded time limit of 10 mins. Bold indicates at least 10x speed up.

| $n$ | $d$=4 | $d$=8 | $d$=16 | $d$=32 | $d$=64 | $d$=128 | $d$=256 | $d$=512 | $d$=1024 |
|---|---|---|---|---|---|---|---|---|---|
| 50000 | 1.2 | 1.9 | 4.1 | 8.2 | 9.9 | **12.8** | **17.8** | OOT | OOT |
| 40000 | 1.0 | 1.7 | 3.5 | 6.4 | 8.3 | **12.5** | **16.2** | OOT | OOT |
| 30000 | 1.4 | 2.2 | 4.7 | 8.4 | **11.0** | **11.4** | **18.7** | OOT | OOT |
| 20000 | 1.6 | 2.6 | 4.2 | 9.7 | 9.4 | **10.8** | **19.3** | **17.6** | **14.4** |
| 10000 | 2.7 | 4.3 | 5.5 | **17.0** | **15.4** | 9.4 | **23.3** | **32.0** | **24.2** |
| 5000 | 3.4 | 5.6 | 7.3 | **19.2** | **14.7** | **14.3** | **32.8** | **46.6** | **32.7** |

ward+backward at $d = 128$, with 68% of cycles stalled on memory. In contrast, FlashSinkhorn transfers only 0.25 GB through HBM, a 444× reduction, by fusing the gradient computation into a single streaming kernel that recomputes the transport matrix on-the-fly rather than materializing it.

**Backward: FlashSinkhorn achieves 10× speedup over KeOps.** Both online methods are compute-bound, yet FlashSinkhorn is **10.3× faster** (19.2 ms vs 197.0 ms). The key difference is instruction efficiency: FlashSinkhorn executes 1.9× fewer instructions (18B vs 35B) through kernel fusion. KeOps suffers from extremely low occupancy (5%) due to its 255 registers per thread and fine-grained kernel launch patterns, preventing the GPU from hiding latency across warps. FlashSinkhorn uses the same register count but achieves 2× higher occupancy (11%) through aggressive kernel fusion, demonstrating that even compute-bound kernels benefit dramatically from reduced instruction count and improved warp-level parallelism.

**OTT-JAX profiling limitations.** We exclude OTT-JAX from NCU profiling because XLA's kernel fusion produces opaque CUDA kernels that conflict with NCU's replay mechanism, returning incomplete metrics. Wall-clock benchmarks show FlashSinkhorn achieves 2-4× speedup over OTT-JAX.

### H.2.2. FLASHSINKHORN VS OTT-JAX: PERFORMANCE ANALYSIS

Both FlashSinkhorn and OTT-JAX implement online Sinkhorn with $O(nd)$ memory complexity, never materializing the $n \times m$ cost matrix. We analyze why FlashSinkhorn achieves 1.1–5.1× speedup despite identical algorithmic complexity.

Table 12 and 13 report the ratio of OTT-JAX to FlashSinkhorn wall-clock time on an NVIDIA A100-80GB GPU. Both methods use identical convergence thresholds and iteration counts.

*Table 9.* Speedup of FlashSinkhorn over KeOps (forward+backward, averaged over 30 runs). OOT indicates exceeded time limit of 10 mins. Bold indicates at least 10x speed up.

| $n$ | $d=4$ | $d=8$ | $d=16$ | $d=32$ | $d=64$ | $d=128$ | $d=256$ | $d=512$ | $d=1024$ |
|---|---|---|---|---|---|---|---|---|---|
| 50000 | 1.2 | 2.0 | 4.1 | 8.5 | **10.5** | **13.6** | **32.4** | OOT | OOT |
| 40000 | 1.1 | 1.7 | 3.6 | 6.7 | 8.9 | **13.5** | **30.9** | OOT | OOT |
| 30000 | 1.4 | 2.3 | 4.7 | 8.6 | **10.3** | **12.0** | **31.4** | OOT | OOT |
| 20000 | 1.5 | 2.6 | 4.1 | 9.7 | 8.5 | **11.5** | **33.3** | OOT | OOT |
| 10000 | 2.8 | 4.3 | 5.6 | **17.1** | **12.3** | **10.3** | **32.8** | **161.4** | **212.3** |
| 5000 | 3.8 | 4.9 | 6.0 | **17.6** | **15.0** | **13.5** | **40.9** | **136.7** | **188.6** |

*Table 10.* Speedup of FlashSinkhorn over Tensorized (forward pass only). Tensorized OOMs at $n \geq 30000$. Values $< 1$ indicate Tensorized is faster. Bold indicates at least 10x speed up.

| $n$ | $d=4$ | $d=8$ | $d=16$ | $d=32$ | $d=64$ | $d=128$ | $d=256$ | $d=512$ | $d=1024$ |
|---|---|---|---|---|---|---|---|---|---|
| 20000 | 9.9 | 9.9 | **12.3** | **10.5** | 8.1 | 4.6 | 3.0 | 1.7 | 0.7 |
| 10000 | 7.4 | 7.6 | 7.6 | 9.1 | 6.6 | 3.2 | 2.1 | 1.7 | 0.7 |
| 5000 | 3.8 | 4.5 | 4.8 | 5.4 | 3.3 | 2.6 | 1.6 | 1.4 | 0.5 |

The speedup ratio can be modeled as: $\text{Speedup}(n, d) \approx \frac{\alpha(n)}{\beta(d)}$, where $\alpha(n)$ captures FlashSinkhorn's advantage and $\beta(d)$ captures OTT-JAX's advantage.

**Factor 1: Kernel fusion ($\alpha$ increases with $n$).** FlashSinkhorn's Triton kernels fuse distance computation, log-sum-exp, and potential updates into a single GPU kernel. This eliminates intermediate memory writes and enables: online softmax accumulation without materializing the $n \times m$ logits matrix; coalesced memory access patterns optimized for streaming; reduced kernel launch overhead (one launch vs. multiple XLA kernels). These advantages scale with $n$ due to better GPU occupancy and amortized overhead.

**Factor 2: cuBLAS efficiency ($\beta$ increases with $d$).** OTT-JAX computes pairwise distances via matrix multiplication. The dominant $XY^\top$ term is dispatched to cuBLAS GEMM, which achieves near-peak TFLOPS for large inner dimension $d$ and benefits from highly optimized tiling and register allocation. The speedup peaks at $d = 32$ due to GPU hardware alignment: the warp size is 32 threads, and Triton's default `BLOCK_K` $= 32$ or $64$. At $d = 32$, the inner reduction loop perfectly fills one warp, maximizing parallelism. The breakeven line (speedup $\approx 1$) follows approximately $n \propto d^2$.

H.2.3. HVP: PARITY AND PERFORMANCE ANALYSIS

**HVP Parity.** We first validate the streaming HVP (damped Schur solve + CG tolerance $\eta$) against a dense reference $G$ computed with the Moore–Penrose pseudoinverse. We report the relative error $\|G_{\tau,\eta} - G_\star\|_F / \|G_\star\|_F$.

We test the configurations $n = m = 512$, $d = 4$, $X, Y \sim \mathcal{N}(0, I)$, $\mathbf{a}, \mathbf{b}$ are random simplex weights, and $\varepsilon \in \{0.1, 0.25, 0.5\}$. We form $P^*$ from converged Sinkhorn potentials. The reference uses an eigendecomposition-based pseudoinverse (threshold $10^{-10}$). The streaming implementation solves the damped Schur system $S_\tau \mathbf{w}_2 = \text{rhs}$ with $S_\tau = S + \tau I$ and CG tolerance $\eta$. From Table 14, small $\tau$ yields high-fidelity parity; tightening $\eta$ improves accuracy until reaching a $\tau$-dependent bias floor.

For high-precision verification, use $\tau = 10^{-7}$ with $\eta = 10^{-7}$. For standard optimization where $\sim 0.5\%$ error is acceptable, the default $\tau = 10^{-5}$ with $\eta = 10^{-6}$ provides a good balance between numerical stability and accuracy.

**Performance** FlashSinkhorn's streaming HVP kernel achieves consistent speedups over both OTT-JAX and OTT-PyTorch/KeOps baselines, with gains increasing at higher feature dimensions. At moderate dimensions ($d = 64$), FlashSinkhorn is 4–6× faster than both baselines. At high dimensions ($d = 128$), the gap widens: FlashSinkhorn achieves 7× speedup over OTT-JAX and 17–27× over KeOps. At $d = 256$, FlashSinkhorn reaches 10–12× over JAX and 35–52× over KeOps at moderate problem sizes ($n \leq 7000$); beyond this, the baselines exceed the 10-minute time limit.

FlashSinkhorn's fused Triton kernels compute the HVP without materializing the $n \times n$ transport matrix, achieving $O(nd)$ memory complexity. As shown in Figure 6, peak memory scales linearly with problem size: from 30 MB at $n = 5,000$ to just 219 MB at $n = 50000$ (with $d = 64$).

*Table 11.* Speedup of FlashSinkhorn over Tensorized (forward+backward, averaged over 30 runs). Tensorized OOMs at $n \geq 30{,}000$. Values $< 1$ indicate Tensorized is faster.

| $n$ | $d{=}4$ | $d{=}8$ | $d{=}16$ | $d{=}32$ | $d{=}64$ | $d{=}128$ | $d{=}256$ | $d{=}512$ | $d{=}1024$ |
|---|---|---|---|---|---|---|---|---|---|
| 20000 | **10.6** | **10.7** | **12.8** | **11.5** | 7.9 | 4.9 | 3.1 | 1.4 | 0.9 |
| 10000 | 8.1 | 8.3 | 8.3 | **10.0** | 5.6 | 3.5 | 2.2 | 1.7 | 0.7 |
| 5000 | 4.5 | 4.3 | 4.2 | 5.3 | 3.6 | 2.5 | 1.9 | 1.4 | 0.5 |

*Table 12.* Speedup of FlashSinkhorn over OTT-JAX (forward pass only, averaged over 50 runs).

| $n$ | $d{=}4$ | $d{=}8$ | $d{=}16$ | $d{=}32$ | $d{=}64$ | $d{=}128$ | $d{=}256$ | $d{=}512$ | $d{=}1024$ |
|---|---|---|---|---|---|---|---|---|---|
| 50000 | 3.9 | 3.9 | 4.5 | 5.1 | 4.2 | 3.4 | 2.3 | 2.0 | 1.5 |
| 40000 | 3.7 | 3.7 | 4.2 | 4.4 | 3.6 | 3.0 | 2.2 | 1.6 | 1.4 |
| 30000 | 2.8 | 2.7 | 3.1 | 3.4 | 2.8 | 2.3 | 1.7 | 1.3 | 1.2 |
| 20000 | 3.2 | 3.3 | 3.6 | 4.3 | 3.6 | 3.0 | 2.4 | 1.6 | 1.2 |
| 10000 | 2.4 | 2.5 | 2.8 | 3.2 | 2.5 | 2.0 | 1.6 | 1.3 | 0.8 |
| 5000 | 1.7 | 1.7 | 1.8 | 1.9 | 1.9 | 1.4 | 1.1 | 0.9 | 0.6 |

### H.2.4. SYMMETRIC VS ALTERNATING SINKHORN: PERFORMANCE ANALYSIS

We compare two update strategies for the Sinkhorn iteration:

- **Symmetric (GeomLoss-style):** A single fused kernel computes both $\mathbf{f}$ and $\mathbf{g}$ updates per iteration with symmetric averaging.

- **Alternating (OTT-style):** Two separate kernels update $\mathbf{f}$ then $\mathbf{g}$ sequentially.

We profile both strategies at $n{=}10000$ with $d = 1024$ on an NVIDIA A100 using NVIDIA Nsight Compute (NCU). Table 17 reports kernel-level metrics. Both strategies saturate the register file at 255 registers per thread, yielding identical theoretical occupancy of 12.5%. However, achieved occupancy and memory stall rates differ across kernels. The symmetric kernel achieves 8.3% occupancy with 3.0% memory stalls. For alternating, the $\hat{\mathbf{f}}$-update kernel performs worse (6.2% occupancy, 5.4% stalls), but the $\hat{\mathbf{g}}$-update kernel achieves full theoretical occupancy (12.5%) with near-zero stalls (0.3%). This efficient $\hat{\mathbf{g}}$ kernel, which uses a larger block size (256 vs. 128) and exploits a more streaming-friendly memory access pattern, compensates for the weaker $\hat{\mathbf{f}}$ kernel. Despite launching twice as many kernels (80 vs. 48), alternating is 1.16× faster at $n{=}10{,}000$.

Table 18 shows wall-clock performance across problem sizes. The crossover occurs around $n \approx 15000$ at $d = 1024$.

The crossover is governed by two competing factors:

- **Kernel launch overhead:** Alternating requires 2× more kernel launches. At small $n$, this overhead dominates total runtime.

- **Memory throughput:** The alternating kernel sustains near-zero memory stalls (0.3%) due to a more streaming-friendly access pattern.

At small $n$ where kernel time is modest ($<$20 ms), launch overhead dominates and symmetric wins. At large $n$ where kernel time exceeds hundreds of milliseconds, the throughput advantage of outweighs the launch overhead, and alternating wins by up to 1.36×.

### H.2.5. LOW-$\varepsilon$ REGIME

We extend the synthetic benchmarks to smaller regularization strengths $\varepsilon \in \{0.10, 0.05, 0.01\}$, reporting forward timing, fp32 precision, iteration budget, and HVP parity.

*Table 13.* Speedup of FlashSinkhorn over OTT-JAX (forward+backward, averaged over 30 runs).

| $n$ | $d=4$ | $d=8$ | $d=16$ | $d=32$ | $d=64$ | $d=128$ | $d=256$ | $d=512$ | $d=1024$ |
|---|---|---|---|---|---|---|---|---|---|
| 50000 | 3.8 | 3.9 | 4.9 | 5.0 | 4.0 | 3.0 | 2.3 | 1.6 | 1.0 |
| 40000 | 4.1 | 4.2 | 5.3 | 5.2 | 4.2 | 3.1 | 2.2 | 1.6 | 1.2 |
| 30000 | 3.9 | 4.0 | 5.1 | 5.2 | 3.7 | 2.7 | 2.2 | 1.7 | 1.0 |
| 20000 | 3.7 | 3.9 | 4.8 | 4.7 | 3.6 | 2.7 | 2.2 | 1.8 | 1.1 |
| 10000 | 3.4 | 3.7 | 3.6 | 4.8 | 2.9 | 2.1 | 1.6 | 1.8 | 1.1 |
| 5000 | 2.9 | 2.6 | 2.7 | 3.6 | 2.7 | 1.9 | 1.7 | 1.7 | 0.9 |

*Table 14.* Parity summary: best achievable error (no damping, tight CG) and error at the default setting.

| $\varepsilon$ | $\tau = 0,\ \eta = 10^{-7}$ | $\tau = 10^{-7},\ \eta = 10^{-7}$ | default $\tau = 10^{-5},\ \eta = 10^{-6}$ |
|---|---|---|---|
| 0.10 | $1.20 \times 10^{-5}$ | $5.02 \times 10^{-5}$ | $4.59 \times 10^{-3}$ |
| 0.25 | $8.33 \times 10^{-6}$ | $4.39 \times 10^{-5}$ | $4.24 \times 10^{-3}$ |
| 0.50 | $6.74 \times 10^{-6}$ | $5.08 \times 10^{-5}$ | $4.89 \times 10^{-3}$ |

**Forward time across $\varepsilon$.** Table 19 reports the 10-iteration forward time at $\varepsilon \in \{0.10, 0.05, 0.01\}$ ($n=m=10000$, $d=64$, TF32, A100-80GB): FlashSinkhorn stays within $\sim 5\%$ of itself across the range, yielding 16–17$\times$ speedup over KeOps and $\sim 4\times$ over OTT-JAX.

**fp32 stability vs fp64 reference.** At fixed iteration count, FlashSinkhorn in fp32 matches a pure-PyTorch dense fp64 Sinkhorn implementation to within $\sim 0.1\%$ relative error even at $\varepsilon=0.01$ (Table 20). The online max-subtraction (Algorithm 1, lines 10–13) keeps each tile's softmax numerically safe: a local row-max is computed per tile, the running global max rescales accumulated statistics, so no tile ever sees unshifted exponents. The remaining error at $\varepsilon=0.01$ comes from fp32 accumulation across many iterations.

**Iteration budget at fixed convergence.** Per-iteration cost is essentially $\varepsilon$-independent ($\sim 3.81\,\mathrm{ms}$ in strict fp32 at $n=m=10000$, $d=64$), but the number of iterations needed for convergence grows as $\varepsilon$ shrinks. Total solve time therefore scales with the iteration budget rather than the per-iteration cost.

**HVP parity at low $\varepsilon$.** The Schur complement underlying our HVP becomes ill-conditioned as $\varepsilon \to 0$: the smallest positive eigenvalue of $H^*$ shrinks as $O(e^{-1/\varepsilon})$ (Li et al., 2025b). We extend the dense ground-truth parity test to $\varepsilon=0.01$ ($n=m=512$, $d=4$, dense Moore–Penrose reference). With our default Tikhonov regularization $\tau=10^{-5}$, CG converges at all tested $\varepsilon$ with $<1\%$ relative error even at $\varepsilon=0.01$ (Table 22). CG requires more iterations as $\varepsilon$ shrinks (78 at $\varepsilon=0.10$, 195 at $\varepsilon=0.01$). The error floor is dominated by $\tau$; tightening it (e.g., $\tau=10^{-6}$, $\eta=10^{-5}$) gives marginally lower error at the same CG cost. Improving the conditioning through preconditioning is an active research direction.

### H.2.6. RECTANGULAR $n \neq m$ REGIME

The streaming kernel handles rectangular shapes natively without algorithmic changes; our main experiments focus on $n \approx m$ because that is the dominant regime in dataset-comparison workloads (OTDD, shuffled regression). Table 23 reports forward times on rectangular pairs at $d=128$, $\varepsilon=0.1$, 10 iterations, TF32 (A100-80GB). Relative to the square case, the FlashSinkhorn-over-KeOps speedup remains large but degrades as the aspect ratio becomes extreme: $13.3\times$ at $10\mathrm{k} \times 10\mathrm{k}$, $11.1\times$ at $1\mathrm{k} \times 10\mathrm{k}$, and $8.3\times$ at $500 \times 50\mathrm{k}$.

At moderate aspect ratios ($10\times$), runtime drops substantially relative to the square case, but not in proportion to $nm$ because fixed overheads remain. At extreme ratios ($100\times$), parallelism along the short dimension becomes limited: with $n=500$ and a row-block size of $B_N=64$, only $\sim 8$ row blocks are exposed, which underutilizes the GPU. FlashAttention-2-style work partitioning along the long dimension (Dao, 2024) is a natural direction to recover throughput in this regime.

### H.3. OTDD Benchmarks

We evaluate FlashSinkhorn on Optimal Transport Dataset Distance (OTDD) (Alvarez-Melis & Fusi, 2020), which measures distances between labeled datasets using a combined feature-label cost.

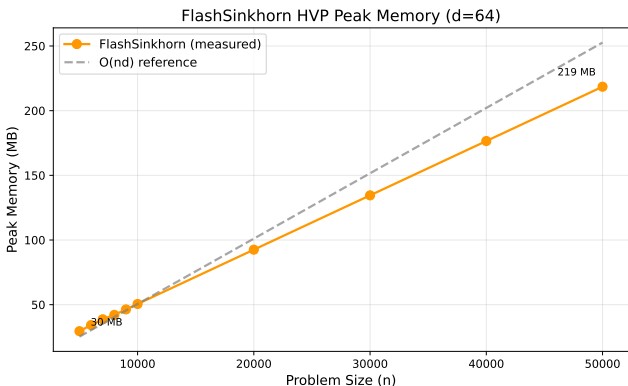

*Figure 6.* FlashSinkhorn HVP peak memory vs. problem size ($d = 64$). Linear scaling confirms $O(nd)$ memory complexity.

*Table 15.* Speedup of FlashSinkhorn HVP over OTT-JAX Hessian (50 CG iterations, averaged over 50 runs). OOT indicates exceeded time limit of 10 mins.

| $n$ | $d{=}4$ | $d{=}8$ | $d{=}16$ | $d{=}32$ | $d{=}64$ | $d{=}128$ | $d{=}256$ | $d{=}512$ |
|---|---|---|---|---|---|---|---|---|
| 10000 | 1.5 | 1.6 | 2.2 | 3.6 | 4.6 | 7.0 | OOT | OOT |
| 9000 | 1.4 | 1.5 | 2.1 | 3.4 | 4.2 | 6.5 | OOT | OOT |
| 8000 | 1.2 | 1.3 | 2.0 | 3.1 | 3.9 | 6.0 | OOT | OOT |
| 7000 | 1.2 | 1.3 | 1.9 | 2.8 | 3.6 | 5.5 | 9.9 | OOT |
| 6000 | 1.1 | 1.2 | 1.7 | 2.6 | 4.5 | 7.0 | **12.5** | OOT |
| 5000 | 1.0 | 1.0 | 1.5 | 2.2 | 3.7 | 5.7 | **10.3** | OOT |

OTDD uses a label-augmented cost combining Euclidean feature distance and class-to-class Wasserstein distance:

$$C(x_i, y_j) = \lambda_1 \|\mathbf{x}_i - \mathbf{y}_j\|_2^2 + \lambda_2 W[\ell_i, \ell_j] \tag{32}$$

where $\mathbf{x}_i, \mathbf{y}_j \in \mathbb{R}^d$ are feature embeddings, $\ell_i, \ell_j \in \{1, \ldots, V\}$ are class labels, and $W \in \mathbb{R}^{V \times V}$ is the matrix of class-to-class Wasserstein distances. We use $\lambda_1 = \lambda_2 = 1/2$.

For debiased Sinkhorn divergence, $W$ must include both within-dataset and cross-dataset class distances. Given datasets with $V_1$ and $V_2$ classes respectively, $W$ is constructed as:

$$W = \begin{pmatrix} W_{11} & W_{12} \\ W_{12}^\top & W_{22} \end{pmatrix} \in \mathbb{R}^{(V_1+V_2) \times (V_1+V_2)}, \tag{33}$$

where $W_{11}$ contains within-dataset distances for dataset 1, $W_{22}$ for dataset 2, and $W_{12}$ contains cross-dataset distances. For MNIST and Fashion-MNIST ($V_1 = V_2 = 10$), this yields a $20 \times 20$ matrix (1.6 KB).

FlashSinkhorn supports this custom cost by storing $W$ and computing the combined cost on-the-fly within the Triton kernel: (1) Load labels $\ell_i, \ell_j$ for the current block; (2) Gather $W[\ell_i, \ell_j]$ via indexed lookup; (3) Combine: $C_{ij} = \lambda_1 \cdot C_{\text{euclidean}} + \lambda_2 \cdot W[\ell_i, \ell_j]$. This per-iteration lookup incurs at most $\sim$30% overhead compared to pure Euclidean cost due to register pressure forcing smaller thread blocks (`BLOCK_M`=64 vs 128). However, tensorized backends must materialize the full $n \times n$ augmented cost matrix, requiring $O(n^2)$ memory. FlashSinkhorn trades per-iteration overhead for $O(nd + V^2)$ memory.

**Dataset and Embeddings.** We use MNIST as the source dataset and Fashion-MNIST as the target, with $V = 10$ classes each. Images (grayscale, $28 \times 28$) are preprocessed by replicating the single channel to 3-channel RGB (required by ResNet18's ImageNet-pretrained weights), resizing to $224 \times 224$, and normalizing with ImageNet statistics. We extract $d = 512$-dimensional embeddings from ResNet18's penultimate layer. Sample sizes range from $n = 5000$ to $n = 60000$, with entropy regularization $\varepsilon = 0.1$ and debiased Sinkhorn divergence (requiring 3 OT computations per evaluation). We compare against the official OTDD implementation (Alvarez-Melis & Fusi, 2020), which uses GeomLoss's tensorized

*Table 16.* Speedup of FlashSinkhorn HVP over OTT-PyTorch/KeOps (50 CG iterations, averaged over 50 runs). OOT indicates exceeded time limit of 10 mins.

| $n$ | $d=4$ | $d=8$ | $d=16$ | $d=32$ | $d=64$ | $d=128$ | $d=256$ | $d=512$ |
|---|---|---|---|---|---|---|---|---|
| 10000 | 0.7 | 1.0 | 2.0 | 3.8 | 4.0 | **17.0** | OOT | OOT |
| 9000 | 0.7 | 1.0 | 2.0 | 3.9 | 4.1 | **17.2** | OOT | OOT |
| 8000 | 0.8 | 1.0 | 2.1 | 3.9 | 4.2 | **17.7** | OOT | OOT |
| 7000 | 0.8 | 1.1 | 2.2 | 4.0 | 4.2 | **18.0** | 35.6 | OOT |
| 6000 | 0.9 | 1.2 | 2.3 | 4.1 | 6.2 | **26.7** | 52.2 | OOT |
| 5000 | 1.0 | 1.2 | 2.3 | 4.1 | 6.1 | **25.3** | 45.4 | OOT |

*Table 17.* NCU profiling of symmetric vs. alternating kernels ($n=m$=10,000, $d$=1024, 10 iterations, A100-80GB).

| | Symmetric | Alternating $\hat{\mathbf{f}}$ | Alternating $\hat{\mathbf{g}}$ | Alternating total |
|---|---|---|---|---|
| Occupancy (%) | 8.3 | 6.2 | **12.5** | 9.4 (avg) |
| Mem. stalls (%) | 3.0 | 5.4 | **0.3** | 3.2 (avg) |
| Runtime (ms) | 1531 | 758 | 558 | **1316** |

backend. The OTDD library has a hardcoded `GPU_LIMIT=20000`; beyond this threshold it falls back to CPU computation. For fair comparison at $n \leq 20000$: (1) both methods use the same pre-computed label cost matrix $W$; (2) timing measures only the outer OT computation, excluding $W$ computation; and (3) we set `maxsamples=n` to ensure no subsampling (OTDD defaults to 10000).

**Gradient Flow Setup**   We evaluate gradient-based dataset adaptation, where the source dataset is moved toward the target distribution by following the Sinkhorn gradient:

$$X^{(t+1)} = X^{(t)} - \eta \cdot \nabla_X S_\varepsilon(X^{(t)}, Y). \tag{34}$$

We run 20 optimization steps with learning rate $\eta = 0.1$ using the OTDD label-augmented cost. Each gradient flow step consists of three phases: (1) a *forward pass* that computes the debiased Sinkhorn divergence $S_\varepsilon(X, Y)$ via three Sinkhorn solves; (2) a *backward pass* that computes $\nabla_X S_\varepsilon$ using our custom Triton gradient kernel; and (3) an *update* $X \leftarrow X - \eta \cdot \nabla_X S_\varepsilon$. We report both total wall-clock time and per-step time (ms/step).

**Additional Experiment.**   To isolate the kernel performance from label cost overhead, we benchmark pure Euclidean cost (no labels) using the same MNIST↔Fashion-MNIST dataset. This enables comparison with GeomLoss's online backend (KeOps), which only supports Euclidean cost. The results are summarized in Figure 7.

### H.4. Detect Saddle Escape in OT-Based Regression

**Experiment Setup**   We use the Cornell Flow Cytometry dataset (Benson et al., 2014), containing 40,000 single cells with 5 fluorescent markers (FITC/CD4, PE/CD8, ECD/CD19, PC5/CD45, PC7/CD3). Each marker measures the expression level of a cell surface antigen, commonly used in immunophenotyping.

Given normalized features $X \in \mathbb{R}^{n \times 5}$, we generate:

1. Ground truth transformation: $W^* \in \mathbb{R}^{5 \times 5}$ with entries $W_{ij}^* \sim \mathcal{N}(0, 1/5)$.

2. Clean targets: $Y_{\text{clean}} = XW^*$.

3. Noisy targets: $Y = Y_{\text{clean}} + E$, where $E_{ij} \sim \mathcal{N}(0, \sigma^2)$ with $\sigma = 0.05 \cdot \text{std}(Y_{\text{clean}})$.

4. Shuffled observations: $\widetilde{Y} = \Pi^*(Y)$ for unknown permutation $\Pi^*$.

The optimization objective is $\min_W \mathcal{L}(W)$, where

$$\mathcal{L}(W) = \text{OT}_\varepsilon \left( \frac{1}{n} \sum_{i=1}^n \delta_{\mathbf{y}_i}, \frac{1}{n} \sum_{j=1}^n \delta_{\widetilde{\mathbf{y}}_j} \right) = \min_{P \in \Pi(\frac{1}{n}\mathbb{1}_n, \frac{1}{n}\mathbb{1}_n)} \langle C(W), P \rangle + \varepsilon \text{KL} \left( P \| \left( \frac{\mathbb{1}_n}{n} \otimes \frac{\mathbb{1}_n}{n} \right) \right)$$

*Table 18.* Wall-clock comparison of symmetric vs. alternating (10 iterations). Ratio >1 favors alternating.

| $d$ | $n$ | Symmetric (ms) | Alternating (ms) | Ratio | Winner |
|---|---|---|---|---|---|
| | 10k | 14.0 | 17.2 | 0.81 | Sym. |
| 64 | 20k | 37.4 | 39.6 | 0.94 | Sym. |
| | 50k | 188.1 | 192.2 | 0.98 | Sym. |
| | 10k | 103.7 | 134.6 | 0.77 | Sym. |
| 1024 | 20k | 377.5 | 303.7 | **1.24** | Alt. |
| | 50k | 2248.2 | 1647.0 | **1.36** | Alt. |

*Table 19.* Forward time in ms at low $\varepsilon$ ($n=m=10000$, $d=64$, 10 iterations, TF32, A100-80GB). Parenthetical values give speedup of FlashSinkhorn over the corresponding baseline.

| $\varepsilon$ | Flash | KeOps | OTT-JAX |
|---|---|---|---|
| 0.10 | **7.75** | 125.37 (16.2×) | 31.82 (4.1×) |
| 0.05 | **7.81** | 125.31 (16.0×) | 33.24 (4.3×) |
| 0.01 | **7.60** | 125.27 (16.5×) | 31.46 (4.1×) |

where $\mathbf{y}_i = \mathbf{x}_i W$ and $C_{ij}(W) = \|\mathbf{y}_i - \widetilde{\mathbf{y}}_j\|_2^2$

**Optimizer Configuration**   We test three regularization strengths $\varepsilon \in 0.1, 0.25, 0.5$. The solver uses epsilon scaling with factor 0.9 (66 annealing steps from diameter to final $\varepsilon$), followed by 60 extra iterations at the final regularization for convergence.

1. **Adam phase.** In the saddle region ($\lambda_{\min} < 0.001$), we use full-batch Adam with learning rate 0.03 and momentum parameters $(\beta_1, \beta_2) = (0.9, 0.999)$. Adam's momentum helps traverse the flat regions near saddle points by accumulating gradient information over steps, while the adaptive per-parameter learning rates handle the ill-conditioned curvature typical of saddle neighborhoods.

2. **Newton phase.** Once $\lambda_{\min} \geq 0.001$ confirms a local minimum, we switch to Newton with initial step size 10.0 and Armijo backtracking line search (reduction factor $\beta = 0.5$, sufficient decrease constant $c = 0.1$). The Newton direction is computed via conjugate gradient with maximum 100 iterations and tolerance $10^{-6}$. The inner Hessian (Schur complement) uses Tikhonov regularization $\tau = 10^{-5}$ for numerical stability.

Optimization terminates when the gradient norm falls below $5 \times 10^{-3}$, with early stopping patience of 3 consecutive steps without improvement.

**Why full-batch Adam?**   We use full-batch Adam to obtain a deterministic optimization trajectory and a stable curvature signal for saddle-escape detection. Minibatch SGD introduces gradient noise that can itself facilitate escape from strict saddles—stochastic gradients exhibit variance aligned with negative-curvature directions and can effectively replace explicit perturbations—which would confound our goal of detecting when the full OT objective becomes locally convex via $\lambda_{\min}(H_W)$ monitoring (Daneshmand et al., 2018; Ge et al., 2015; Jin et al., 2017). Moreover, our detector relies on repeated Hessian–vector products at the same iterate, and in the full-batch setting we can amortize the Sinkhorn solve by reusing the current dual potentials and cached normalization statistics across many HVP evaluations; changing minibatches would require repeatedly recomputing these operators and yields a noisier $\lambda_{\min}(H_W)$ signal. This mirrors standard practice in CG-based second-order methods, where multiple $Hv$ queries at fixed parameters benefit from caching shared computations (Martens, 2010).

**Lanczos Iteration for Minimum Eigenvalue**   We estimate the smallest eigenvalue of the parameter Hessian $H_W \in \mathbb{R}^{25 \times 25}$ using ARPACK's implicitly restarted Lanczos method as implemented by `scipy.sparse.linalg.eigsh`. Lanczos requires only matrix–vector products $v \mapsto H_W v$ and avoids explicit Hessian construction. We expose $H_W$ as a LinearOperator whose matvec is a streaming HVP, and call `eigsh` with `which='SA'` (smallest algebraic) and a small Krylov subspace (`ncv=6`). Each Krylov step triggers one HVP, so an eigenvalue check costs $O(N_{\mathrm{mv}}\mathrm{cost}(\mathrm{HVP}))$ time and $O(nd)$ memory—independent of storing the $n \times n$ transport matrix or the data-space Hessian. At $n = 40,000$, this adds $\approx 11$ seconds per check, whereas materializing the data-space Hessian $\nabla_Y^2 \mathrm{OT}_\varepsilon$ would require $O((nd)^2)$ storage ( $> 100\,\mathrm{GB}$ at this scale). Since our HVP oracle is inexact (about 0.5% relative error due to damping and early stopping of the inner solve), we use a modest eigensolver tolerance and treat $\lambda_{\min}$ as a coarse diagnostic (sign/margin) rather than a

*Table 20.* Numerical precision: FlashSinkhorn fp32 vs. pure-PyTorch dense fp64 reference at fixed iteration count ($n$=$m$=10000, $d$=64, A100-80GB).

| $\varepsilon$ | OT value (fp32) | OT value (fp64) | rel. err. (fp32 vs fp64) |
|---|---|---|---|
| 0.10 | 72.99633 | 72.999261 | $4.02 \times 10^{-5}$ |
| 0.05 | 72.571186 | 72.574516 | $4.59 \times 10^{-5}$ |
| 0.01 | 72.173378 | 72.22889 | $7.69 \times 10^{-4}$ |

*Table 21.* Iteration budget to reach convergence at low $\varepsilon$ (FlashSinkhorn, strict fp32, $n$=$m$=10000, $d$=64, A100-80GB).

| $\varepsilon$ | iterations | total time | ms / iter |
|---|---|---|---|
| 0.10 | 2000 | 7.6 s | 3.82 |
| 0.05 | 4000 | 15.2 s | 3.81 |
| 0.01 | 5000 | 19.1 s | 3.81 |

high-precision estimate, consistent with inexact projection methods for eigencomputation. So we set the threshold of $\lambda_{\min}$ switching at 0.001.

**Additional Experiment: Multi-Saddle Escape Trajectory** Figure 8 shows a configuration ($n = 40{,}000$, $\varepsilon = 0.25$, seed=0) that reveals the complex landscape of the shuffled regression objective, characterized by numerous shallow saddle points and local minima in close proximity.

Starting from loss 3.76, the optimization traverses multiple saddle-local transitions before converging at loss 1.77:

1. **Steps 0-105 (Adam):** Descends through a saddle region until $\lambda_{\min} = +0.0047$ signals escape into a shallow local minimum.

2. **Steps 106-110 (Newton):** Newton's large steps cross into an adjacent shallow saddle ($\lambda_{\min} = -0.0027$), triggering fallback.

3. **Steps 111-145:** The optimizer oscillates between shallow saddles and local minima, with two more escape-reentry cycles.

4. **Steps 155-164 (Newton):** Finally reaches a deeper local minimum basin ($\lambda_{\min} = +0.024$), where Newton converges in 9 steps.

Landscape interpretation. The small magnitude of eigenvalues at transition points ($|\lambda_{\min}| < 0.005$ for the first two escapes) suggests the optimizer navigates a region densely populated with shallow saddles and local minima separated by low barriers. Newton's aggressive steps easily cross these barriers, while Adam's smaller updates can remain within a basin. The final successful convergence occurs when the optimizer reaches a deeper basin with $\lambda_{\min} = 0.024$, five times larger than earlier transitions, indicating stronger local convexity that contains Newton's iterates.

This behavior motivates the fallback mechanism: near shallow critical points, Newton may destabilize, but the eigenvalue-based switching rule automatically recovers by reverting to Adam until a more stable basin is found.

*Table 22.* HVP parity at low $\varepsilon$ vs. dense Moore–Penrose ground truth ($n=m=512$, $d=4$, balanced; $\tau$ is the Schur–complement Tikhonov damping; $\eta$ is the CG residual tolerance).

| $\varepsilon$ | $\tau$ | $\eta$ | HVP rel. err. | CG iters | converged |
|---|---|---|---|---|---|
| 0.10 | $10^{-5}$ | $10^{-6}$ | $4.75 \times 10^{-3}$ | 78 | Y |
| 0.05 | $10^{-5}$ | $10^{-6}$ | $4.35 \times 10^{-3}$ | 83 | Y |
| 0.01 | $10^{-5}$ | $10^{-6}$ | $8.54 \times 10^{-3}$ | 195 | Y |
| 0.01 | $10^{-6}$ | $10^{-5}$ | $7.69 \times 10^{-3}$ | 195 | Y |

*Table 23.* Forward time on rectangular point clouds ($d=128$, $\varepsilon=0.1$, 10 iterations, TF32, A100-80GB). Parenthetical values give the FlashSinkhorn-over-KeOps speedup.

| $n \times m$ | ratio | Flash (ms) | KeOps (ms) |
|---|---|---|---|
| $10k \times 10k$ | $1\times$ | **12.01** | $160.24 \ (13.3\times)$ |
| $1k \times 10k$ | $10\times$ | **8.18** | $91.10 \ (11.1\times)$ |
| $2k \times 20k$ | $10\times$ | **18.07** | $184.26 \ (10.2\times)$ |
| $10k \times 1k$ | $10\times$ | **8.36** | $91.16 \ (10.9\times)$ |
| $500 \times 50k$ | $100\times$ | **49.14** | $406.65 \ (8.3\times)$ |

*Table 24.* **Method support for OTDD benchmarks.** Label-augmented cost requires custom cost function support: $C(x_i, y_j) = \lambda_1 \|x_i - y_j\|_2^2 + \lambda_2 W[\ell_i, \ell_j]$. GeomLoss (KeOps) does not support custom cost functions, limiting it to no-label experiments. Prior methods run out of memory (OOM) beyond $n = 20,000$ on a 40 GB A100 GPU.

| Method | With Labels | Without Labels | Memory | Max $n$ |
|---|---|---|---|---|
| FlashSinkhorn | ✓ | ✓ | $O(nd)$ | $\geq 60,000$ |
| GeomLoss (KeOps) | ✗ | ✓ | $O(nd)$ | 20,000 |
| GeomLoss (Tensorized) | ✓ | ✓ | $O(n^2)$ | 20,000 |

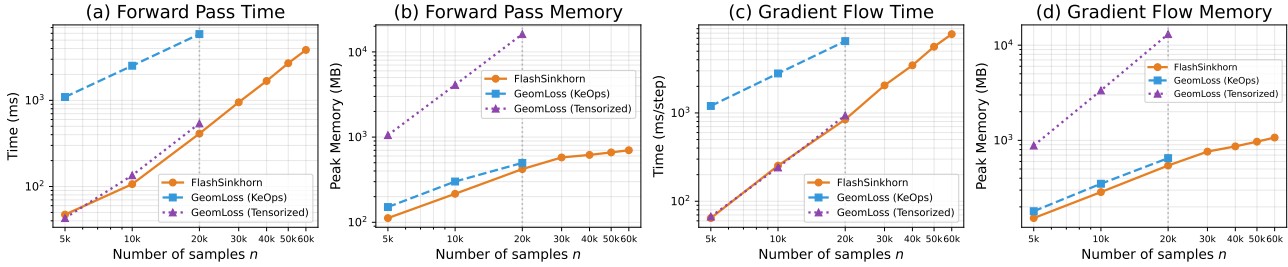

*Figure 7.* **No-label Sinkhorn divergence benchmark.** Comparison of FlashSinkhorn against GeomLoss with KeOps (online) and Tensorized backends on MNIST ($d = 784$, debiased Sinkhorn, $\varepsilon = 0.1$). **(a,b)** Forward pass: FlashSinkhorn matches Tensorized speed while using $38\times$ less memory. KeOps is 14-26$\times$ slower. **(c,d)** Gradient flow (forward + backward per step): FlashSinkhorn scales to $n = 60000$ while KeOps and Tensorized run out of memory beyond $n = 20000$ (gray dashed line). Tensorized's $O(n^2)$ memory grows to 16 GB at $n = 20000$, whereas FlashSinkhorn's $O(nd)$ memory stays below 1.1 GB even at $n = 60000$.

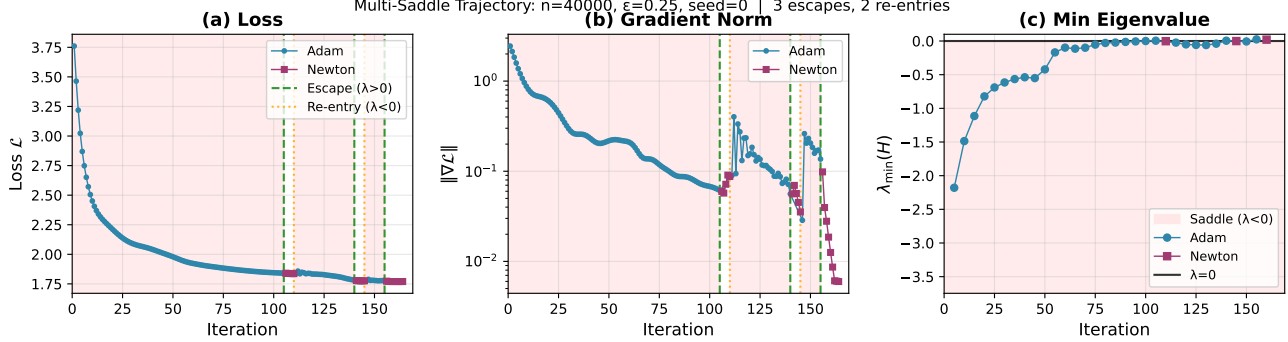

*Figure 8.* **Multi-saddle escape in shuffled regression** ($n=40,000$, $\varepsilon=0.25$, seed=0). (a) loss, (b) gradient norm, (c) estimated $\lambda_{\min}(H)$. Green dashed lines mark escapes ($\lambda_{\min}>0.001$; switch to Newton) and orange dotted lines mark re-entries ($\lambda_{\min}<0.001$; fallback to Adam). Overall: 3 escapes, 2 re-entries; loss $3.76 \rightarrow 1.77$.

