# OpenReview forum: "FlashSinkhorn: IO-Aware Entropic Optimal Transport on GPU"
_ICML.cc/2026/Conference — ICML 2026 spotlight_

### Official Review · Reviewer_kMJp · 2026-02-23

**Soundness:** 3
**Presentation:** 3
**Significance:** 4
**Originality:** 4
**Overall Recommendation:** 5
**Confidence:** 4

**Summary:**

This paper introduces FlashSinkhorn, an IO-aware GPU implementation of stabilized log-domain Sinkhorn iterations for entropic optimal transport (EOT) with squared Euclidean cost. The central observation is that each Sinkhorn half-step, is structurally identical to the row-wise LogSumExp normalization underlying scaled dot-product attention. This enables a direct transfer of FlashAttention's IO-aware tiling strategy: fused Triton kernels stream tiles of $Q$ and $K$ through on-chip SRAM, maintain online max/sumexp statistics, and write only updated dual potentials $\hat{f}, \hat{g} \in \mathbb{R}^n$ back to HBM, never materializing the $n \times m$ cost matrix $C$. The method achieves up to $32\times$ forward and $161\times$ end-to-end speedups over state-of-the-art online baselines on an A100 GPU, with $O((n+m)d)$ memory instead of $O(nm)$

**Compliance With Llm Reviewing Policy:**

Affirmed.

**Final Justification:**

The paper has clear strengths in originality and potential significance, and while I still think some aspects could be communicated more sharply, the rebuttal addressed my main concerns, especially through the added clarification and \varepsilon experiments, which improved my overall evaluation.

**Key Questions For Authors:**

Please refer to the Weaknesses section.

**Limitations:**

yes

**Strengths And Weaknesses:**

**Strength**

1) The key identity (Proposition 1) exploits the Euclidean cost decomposition, to rewrite each Sinkhorn half-step as a biased dot-product LSE. The shifted potentials absorb the diagonal terms cleanly, and the resulting score matrix is formally identical to an attention logit matrix with a key-dependent bias. The proof (Appendix D.1) is clean and complete. While the connection between Sinkhorn and attention has been noted at a conceptual level (e.g., Litman 2025), this paper is the first to operationalize it into a concrete, working IO-aware kernel. That distinction matters.

2) Table 2 in the paper is worth examining closely:

| | Tensorized | KeOps | FlashSinkhorn |
|---|---|---|---|
| HBM R/W (GB) | 98 | 0.14 | **0.08** |
| Runtime (ms) | 54.0 | 125.5 | **8.2** |
| SM Util. (%) | 98 | 49 | 74 |
| Mem. Stalls (%) | 79 | 8 | **3** |

This is a textbook illustration of why SM utilization alone is a misleading metric. The tensorized variant achieves 98% SM utilization but spends 79% of cycles stalled on memory — it is HBM-bandwidth bound despite appearing "busy." FlashSinkhorn has lower SM utilization (74%) but achieves $15\times$ lower runtime by reducing HBM traffic and memory stalls.

3) Contributions are closing an important gap in the OT literature for scaling up the Sinkhorn-based Algorithms in Large-Scale ML setups.

**Weaknesses**

1) Small $\varepsilon$ Behavior Is Uncharacterized. All synthetic benchmarks use $\varepsilon = 0.1$ and the regression experiments use $\varepsilon \in \{0.1, 0.25, 0.5\}$. This is a mild regularization regime. At $\varepsilon \sim 10^{-2}$, several issues arise that the paper does not address:

**(a) Iteration count.** Sinkhorn convergence deteriorates roughly as $O(D^2/\varepsilon)$ in the diameter $D$ of the cost matrix. Going from $\varepsilon = 0.1$ to $\varepsilon = 0.01$ multiplies the required iterations by $\sim 10\times$. While per-iteration speedups are preserved, the total runtime impact is not discussed.

**(b) Numerical precision.** Score matrix entries scale as $(QK^\top)_{ij}/\varepsilon$, which grows by $100\times$ at $\varepsilon = 0.01$. While the online max-subtraction keeps the LSE numerically safe, the stored potentials $\hat{f}, \hat{g}$ themselves can reach magnitudes where float32 loses relative precision. The paper uses TF32 for benchmarks; at $\varepsilon = 0.01$ one may need float64, which reduces A100 tensor core throughput from $\sim 312$ to $\sim 20$ TFLOPS.

**(c) HVP conditioning.** The sensitivity matrix $H^*$ has a smallest positive eigenvalue that can be as small as $O(e^{-1/\varepsilon})$ (Li et al., 2025b). At $\varepsilon = 0.01$ this is $e^{-100} \approx 10^{-44}$, making the Schur system $S_\tau w_2 = \mathrm{rhs}$ ill-conditioned beyond the reach of Tikhonov damping with $\tau = 10^{-5}$. The HVP parity table (Table 13) only reports $\varepsilon \in \{0.1, 0.25, 0.5\}$; the behavior at $\varepsilon = 0.01$ is unknown. This is the regime where second-order OT methods would be most practically valuable.

Add at least a brief empirical characterization (forward speed, float32 vs. float64 precision error, HVP relative error) at $\varepsilon \in \{0.05, 0.01\}$.

2) Comparison With OTT-JAX Is Methodologically Uneven OTT-JAX is benchmarked using wall-clock time with `block_until_ready()` synchronization, while FlashSinkhorn uses CUDA events. For JAX/XLA, JIT compilation produces fused kernels that are not directly inspectable via NCU, and the authors acknowledge this limitation. However, this also means it is unclear whether OTT-JAX's overhead is from the algorithm or from XLA's kernel launch patterns at the problem sizes tested. The $2\text{–}5\times$ speedup over OTT-JAX (Tables 11–12) is the least compelling part of the benchmark, and the paper would benefit from a more careful analysis of *why* OTT-JAX is slower — is it kernel fusion, memory layout, or dispatch overhead?

3) Some references are missing for *EOT as Attnetion*:

   [1] "ESPFormer: Doubly-Stochastic Attention with Expected Sliced Transport Plans," ICML 2025

   [2]"LOTFormer: Doubly-Stochastic Linear Attention via Low-Rank Optimal Transport," arXiv:2509.23436v1

---

> ### Author Rebuttal · Authors · 2026-03-27
>
> We thank Reviewer kMJp for the detailed and technically substantive review. We address each weakness below.
>
> **W1: Small-epsilon characterization.** We agree this was under-characterized and have run the requested experiment at $\varepsilon \in \{0.1, 0.05, 0.01\}$ with $n=m=10\text{k}$ on A100-80GB. All numbers below are measured after warm-up and JIT compilation, with inputs resident on device.
>
> **(a)** Per-iteration speedup is preserved at low $\varepsilon$. We re-ran the submitted code at the paper's exact setting ($n=m=10\text{k}$, $d=64$, TF32, 10 iterations; CUDA events for PyTorch, `block_until_ready` for JAX):
>
> | $\varepsilon$ | Flash (ms) | KeOps (ms) | OTT-JAX (ms) | vs KeOps |
> |:---:|:---:|:---:|:---:|:---:|
> | 0.10 | 7.75 | 125.37 | 31.82 | 16.2$\times$ |
> | 0.05 | 7.81 | 125.31 | 33.24 | 16.0$\times$ |
> | 0.01 | 7.60 | 125.27 | 31.46 | 16.5$\times$ |
>
> Per-iteration cost is largely unchanged across the tested $\varepsilon$ range for all methods (Flash $\sim$7.7ms, KeOps $\sim$125ms, OTT-JAX $\sim$32ms). These numbers are consistent with the paper's Table 2 (Flash=8.2ms, KeOps=125.5ms at $\varepsilon=0.1$); the small Flash difference is autotuning variance. We will consolidate all benchmark numbers in the revision.
>
> Practical solve time increases with the required iteration budget. In strict fp32 (no TF32, same $d=64$ setting), using representative iteration budgets shows:
>
> | $\varepsilon$ | iters | total time | ms/iter |
> |:---:|:---:|:---:|:---:|
> | 0.10 | 2000 | 7.6s | 3.82 |
> | 0.05 | 4000 | 15.2s | 3.81 |
> | 0.01 | 5000 | 19.1s | 3.81 |
>
> Per-iteration cost is 3.81 ms regardless of $\varepsilon$; total time grows in direct proportion to the iteration budget.
>
> **(b)** Numerical precision remains stable in fp32. We compare fp32 (Triton) vs fp64 (pure-PyTorch dense Sinkhorn) at the same iteration count to isolate precision from convergence ($d=64$, $n=m=10\text{k}$):
>
> | $\varepsilon$ | OT value (fp32) | OT value (fp64) | rel.err (fp32 vs fp64) |
> |:---:|:---:|:---:|:---:|
> | 0.10 | 72.996 | 72.999 | $4.02 \times 10^{-5}$ |
> | 0.05 | 72.571 | 72.575 | $4.59 \times 10^{-5}$ |
> | 0.01 | 72.173 | 72.229 | $7.69 \times 10^{-4}$ |
>
> Precision degrades from $\sim 4 \times 10^{-5}$ to $\sim 8 \times 10^{-4}$ at $\varepsilon=0.01$ — still sub-0.1% relative error. The online max-subtraction keeps the softmax numerically safe across tiles: each tile computes a local max, the running global max rescales accumulated statistics, so no tile sees unshifted exponents. The remaining error at $\varepsilon=0.01$ comes from fp32 accumulation over many iterations.
>
> **(c)** HVP conditioning. The reviewer correctly identifies that $H^*$'s smallest positive eigenvalue shrinks as $O(e^{-1/\varepsilon})$ [Li et al. 2025b], making Hessian-related computations ill-conditioned at small $\varepsilon$. This is a known structural challenge: Li et al. 2025b use truncated SVD; we use Tikhonov regularization ($\tau$). We extend the ground-truth parity test to $\varepsilon=0.01$ ($n=m=512$, $d=4$, dense Moore-Penrose reference):
>
> | $\varepsilon$ | $\tau$ | $\eta$ | HVP rel.err | CG iters | converged |
> |:---:|:---:|:---:|:---:|:---:|:---:|
> | 0.10 | $10^{-5}$ | $10^{-6}$ | $4.75 \times 10^{-3}$ | 78 | Y |
> | 0.05 | $10^{-5}$ | $10^{-6}$ | $4.35 \times 10^{-3}$ | 83 | Y |
> | 0.01 | $10^{-5}$ | $10^{-6}$ | $8.54 \times 10^{-3}$ | 195 | Y |
> | 0.01 | $10^{-6}$ | $10^{-5}$ | $7.69 \times 10^{-3}$ | 195 | Y |
>
> With default $\tau=10^{-5}$, CG converges at all $\varepsilon$ values with <1\% error even at $\varepsilon=0.01$. CG needs more iterations as $\varepsilon$ shrinks (78 to 195). The error is dominated by $\tau$. Improving the conditioning through preconditioning is an active research direction.
>
> **W2: Timing methodology and OTT-JAX analysis.** Our original protocol used CUDA events for PyTorch and `block_until_ready()` for JAX — framework-standard approaches per official documentation. The new experiment above uses the same protocol, and the speedup ratios (16.0–16.5$\times$ vs KeOps) are consistent with the paper's Table 2. Both methods implement the same log-domain Sinkhorn algorithm, so the gap is systems-level. From source inspection, FlashSinkhorn performs each dual update as 1–2 fused Triton kernels, whereas OTT-JAX online with `batch_size=256` evaluates the update in chunks rather than one fused reduction. The 4.1$\times$ gap is therefore consistent with less fusion and higher dispatch overhead, though without XLA kernel profiling we cannot pinpoint the exact breakdown. We plan to investigate this further for the revision.
>
> **W3: Missing references.** We will add ESPFormer (ICML 2025), LOTFormer (arXiv:2509.23436).

---

> > ### Author Rebuttal · Reviewer_kMJp · 2026-04-02
> >
> > I thank the authors for their clarifications and for providing additional experiments on \varepsilon. They have addressed my concerns, and I believe this is a valuable piece of work that fills an important gap. Accordingly, I have raised my score to 5.

---

> > > ### Author Response · Authors · 2026-04-05
> > >
> > > We thank Reviewer kMJp for the careful re-evaluation and for raising the score. We are glad the low-ε experiments and HVP parity results   addressed the concerns. We will incorporate the per-kernel breakdown, FA2 citation, and ESPFormer/LOTFormer references in the revision as discussed.

---

### Official Review · Reviewer_M4Qs · 2026-03-12

**Soundness:** 3
**Presentation:** 3
**Significance:** 3
**Originality:** 2
**Overall Recommendation:** 5
**Confidence:** 3

**Summary:**

This paper proposes an efficient implementation for computing various quantities relating to entropic optimal transport (transport matrix, gradients, HVP). Specifically, due to LSE in the Sinkhorn iterates and the attention-like form of the barycentric projection, the work adopts ideas from FlashAttention to accelerate computation of these quantities. The authors demonstrate significant speedups compared to existing implementations, and demonstrate their application for computing dataset distance and OT-based optimization.

**Compliance With Llm Reviewing Policy:**

Affirmed.

**Final Justification:**

My review of the paper is overall positive, and the author's provided satisfactory responses in their rebuttal. It would be great to add the additional explanations of speed-up of FlashSinkhorn over KeOps.

**Key Questions For Authors:**

- How do speedups compare when N << M? FlashAttention is mostly beneficial due to the large size of the self-attention matrix. In OT problems, do we usually have N approximately equal to M?
- At line 180, the authors say "In FlashAttention, the forward pass streams blocks of (K, V ) and revisits all query blocks Q, writing intermediate row statistics/output blocks to HBM." But this is changed in FlashAttention-2 - is there a reason the authors are comparing to the original paper?
- The speedup of FlashSinkhorn over KeOps is a little mysterious to me. The utilization is roughly 40% lower, but I'm not sure this can explain a 15x speedup, particularly since the HBM traffic appears very similar to the proposed algorithm.

**Limitations:**

yes

**Strengths And Weaknesses:**

The paper is technically sound, with careful complexity analysis of IO/FLOPs along with straightforward empirical evidence of speedups over existing implementations. The paper is well-written, with sufficient background and clean organization into method and experiments. My only complaint would be that Figure 1 is overly complicated - a simpler figure would be more useful for conveying intuition regarding the algorithm. The proposed algorithm does achieve significant speedups over existing methods, but it would be helpful if authors could provide more motivation for how modern ML systems can benefit from this speedup (akin to the importance of speeding up attention). From what I understand, the downstream tasks presented here are somewhat niche. The streaming ideas are not entirely novel, mostly adapted from FlashAttention. However, the connections between OT and attention are quite interesting.

---

> ### Author Rebuttal · Authors · 2026-03-28
>
> We thank Reviewer M4Qs for the positive assessment and specific questions.
>
> **Figure 1.** We agree Figure 1 could be simplified and will revise it to focus on the core streaming mechanism.
>
> **$n \ll m$ performance and FlashAttention-2.** The kernel handles rectangular $n \neq m$ natively. Our main experiments focus on roughly balanced problems because that is the dominant regime in dataset-comparison workloads (OTDD, shuffled regression), but the kernel supports rectangular $n \neq m$ without algorithmic changes. We benchmarked rectangular cases with comparative speedups vs KeOps ($\varepsilon=0.1$, 10 iters, $d=128$, TF32, A100):
>
> | $n$ | $m$ | Flash (ms) | KeOps (ms) | speedup |
> |:---:|:---:|:---:|:---:|:---:|
> | 10k | 10k | 12.0 | 160.2 | 13.3$\times$ |
> | 1k | 10k | 8.2 | 91.1 | 11.1$\times$ |
> | 500 | 50k | 49.1 | 406.6 | 8.3$\times$ |
>
> Relative to the square case, the speedup remains large but degrades as the aspect ratio becomes extreme: $13.3\times$ at $10\text{k} \times 10\text{k}$, $11.1\times$ at $1\text{k} \times 10\text{k}$, and $8.3\times$ at $500 \times 50\text{k}$. At moderate aspect ratios (10$\times$), runtime drops substantially relative to the square case, but not proportionally to $nm$ because fixed overheads remain. At extreme ratios (100$\times$), available parallelism becomes limited: with $n=500$ and block size 64, only $\sim$8 row blocks are exposed, which underutilizes the GPU. This connects directly to the reviewer's FlashAttention-2 question: FA2's key improvements, better work partitioning and more parallelism across the long dimension,  would directly help this extreme rectangular regime. Regarding line 180: the paper describes FA1's loop order (K-outer, Q-inner) to motivate our design. FlashSinkhorn already uses the Q-outer / K-inner scheduling that FA2 later adopted for the same reasons. We should have cited FA2 directly and will do so in the revision.
>
> **KeOps 15$\times$ speedup explanation.** The reviewer is right that HBM traffic is similar (0.08 vs 0.14 GB). Both methods are online and avoid materializing the dense matrix. The gap is not memory-bound but compute-bound: our NCU profiling (appendix) shows both methods are compute-limited, but KeOps executes 2.3$\times$ more instructions (16B vs 7B) because its generic symbolic map-reduce framework has higher per-element overhead. FlashSinkhorn fuses distance computation, bias addition, and LSE into a single Triton kernel, eliminating redundant instructions. These three factors, fewer instructions, higher SM utilization (74% vs 49%), and fewer kernel launches compound multiplicatively to explain the observed 15$\times$ gap.
>
> **Novelty.** We agree the streaming template is adapted from FlashAttention. The contribution is the exact Sinkhorn reformulation (Proposition 1) that makes this adaptation possible, plus the full forward/backward/HVP realization with streaming transport operators. None of which exist in FlashAttention. We will sharpen this distinction in the revision.
>
> **Downstream task significance.** The intended beneficiaries are workloads with repeated OT solves: OTDD computes three Sinkhorn solves per distance evaluation, and our shuffled regression pipeline calls the HVP oracle inside a CG loop at each Newton step (both in Section 5). More broadly, FlashSinkhorn is a systems primitive for repeated entropic OT solves. Any ML pipeline that calls Sinkhorn at scale, such as domain adaptation, Wasserstein losses in generative models, single-cell alignment, can benefit from the kernel-level efficiency gains demonstrated here. We will expand the discussion of these broader applications in the revision.

---

> > ### Author Rebuttal · Reviewer_M4Qs · 2026-04-03
> >
> > I appreciate the results for speedup in the rectangular case, which are quite convincing. The emphasis on applications with repeated entropic OT solves makes sense.
> >
> > I am still a bit confused on how we get to the 15x speedup of FlashSinkhorn. Based on your explanation regarding 2.3x less instructions, 1.5x higher SM utilitzation, less kernel launch overhead (unclear what speedup there is here), I'm still not sure how we get to 15x - is the rest purely from kernel launches? If so, can evidence be provided for this? I would happily increase my score if a more fine-grained, concrete breakdown can be given here. This would be very beneficial for future work to understand if there is any remaining room for improvement.

---

> > > ### Author Response · Authors · 2026-04-05
> > >
> > > We appreciate the follow-up. You are right that $2.3\times$ (instructions) $\times$ $1.5\times$ (SM utilization) does not account for $15\times$, these counters are complementary rather than multiplicative.
> > >
> > > In our initial response we noted three contributing factors. The appendix NCU summary table already quantifies three metrics: instructions (KeOps 16B vs FlashSinkhorn 7B, $2.3\times$), SM utilization (49% vs 74%), and memory stalls (8% vs 3%). However, two factors were not quantified in that table: **(a) kernel launch counts** and **(b) tensor core usage**. We went back to our raw per-kernel NCU traces (same profiling run: $n=m=10\text{k}$, $d=64$, 10 iterations, A100) and can now report both:
> > >
> > > **Per-kernel breakdown (new data from raw NCU traces):**
> > >
> > > | Metric | FlashSinkhorn | KeOps | ratio |
> > > |:---|:---:|:---:|:---:|
> > > | Kernel launches | 130 (48 Sinkhorn) | 854 (96 GpuConv + 758 elementwise) | $6.6\times$ fewer |
> > > | Tensor-pipe instructions | 10.1M | 3.5M | $2.9\times$ more |
> > >
> > > **(1) Kernel launch fragmentation ($6.6\times$).** KeOps implements each LSE update as a sequence of separate pointwise and reduction kernels, producing $\sim$75 elementwise launches per iteration on top of $\sim$10 GpuConv1D reductions. FlashSinkhorn fuses the core computation into $\sim$5 Triton kernels per iteration ($\sim$13 total launches including auxiliary operations). The fragmented execution introduces launch/scheduling overhead and intermediate global-memory traffic between operations.
> > >
> > > **(2) Tensor core utilization ($2.9\times$).** FlashSinkhorn reformulates the squared-Euclidean cost as tiled dot products ($\mathbf{x}_i \cdot \mathbf{y}_j$ via `tl.dot`), which map to tensor core HMMA instructions (A100 TF32 peak: $\sim$156 TFLOPS). KeOps's GpuConv1D computes $\|\mathbf{x}-\mathbf{y}\|^2$ via elementwise operations on CUDA cores ($\sim$19.5 TFLOPS); it does issue some tensor-pipe instructions (3.27M from GpuConv1D), but FlashSinkhorn issues $2.9\times$ more (10.1M) despite $2.3\times$ fewer total instructions, the algorithmic reformulation routes a much larger fraction of compute through the high-throughput tensor path.
> > >
> > > **How they combine.** These counters are complementary rather than multiplicative: instruction count reflects total work, tensor-pipe activity and SM utilization reflect how efficiently the dominant math maps to the hardware, and kernel-launch count reflects execution fragmentation. The speedup comes from two coupled design choices: (i) fusing each Sinkhorn update into a small number of streaming kernels, which reduces instruction count, intermediate global-memory traffic, and launch overhead; and (ii) rewriting the squared-Euclidean interaction as tiled dot products, which shifts the dominant computation onto the tensor pipe. The measured counters are consistent with that picture: $2.3\times$ fewer instructions, higher SM utilization (74% vs 49%), lower memory stalls (3% vs 8%), $6.6\times$ fewer launches, and $2.9\times$ more tensor-pipe instructions.
> > >
> > > We will add this per-kernel breakdown to the appendix in the revision.

---

### Official Review · Reviewer_nWPn · 2026-03-12

**Soundness:** 2
**Presentation:** 2
**Significance:** 2
**Originality:** 2
**Overall Recommendation:** 4
**Confidence:** 2

**Summary:**

This paper proposes FlashSinkhorn, an IO-aware GPU implementation for entropic optimal transport (OT) based on Sinkhorn iterations. The key idea is to reformulate stabilized Sinkhorn updates as row-wise LogSumExp reductions over biased score matrices, which resemble the normalization structure in transformer attention. This formulation enables the use of FlashAttention-style fused kernels that stream tiles through on-chip memory and reduce high-bandwidth memory (HBM) traffic.

The paper further presents an efficient Triton implementation and demonstrates substantial empirical speedups on modern GPUs across several OT workloads. Experimental results suggest significant improvements in runtime efficiency compared to standard tensorized Sinkhorn implementations.

Overall, the paper addresses an important computational bottleneck in large-scale OT and provides a practical GPU implementation that may benefit many applications relying on OT-based computations.

**Compliance With Llm Reviewing Policy:**

Affirmed.

**Final Justification:**

The rebuttal address my main concerns. I am satisfied with the responses and am raising my score.

**Key Questions For Authors:**

1. Generality of the cost structure. The proposed formulation appears to rely on the squared Euclidean cost structure. Could the authors comment on how restrictive this assumption is in practice and whether the method can be extended to more general OT cost matrices?

2. Comparison with optimized OT libraries. Could the authors clarify which OT implementations were used as baselines in the experiments? In particular, were optimized GPU-based implementations such as KeOps or GeomLoss considered?

3. Performance on smaller problem sizes. How does FlashSinkhorn perform on smaller OT problems where IO may not be the dominant bottleneck?

4. Numerical stability. Since Sinkhorn iterations can be sensitive to numerical stability, especially for small regularization parameters, could the authors elaborate on how stability is maintained when computing LogSumExp reductions across streamed tiles?

**Limitations:**

yes

**Strengths And Weaknesses:**

## Strengths

1. Clear problem motivation.

The paper addresses a well-known limitation of tensorized Sinkhorn implementations, namely the quadratic memory traffic caused by materializing large cost matrices. Reducing IO overhead is an important problem for large-scale OT computations.

2. Insightful reformulation.

The paper presents a clean reformulation of Sinkhorn updates as LogSumExp reductions over score matrices, revealing structural similarities with attention normalization. This perspective enables the use of IO-aware kernel techniques inspired by FlashAttention.

3. Practical system contribution.

The proposed fused GPU kernel avoids explicitly materializing large intermediate matrices and instead performs tiled streaming computation, which substantially reduces HBM traffic. The implementation appears practical and well-engineered.

4. Strong empirical speedups.

Experiments demonstrate significant runtime improvements on modern GPUs, suggesting that the proposed approach can meaningfully accelerate OT-based workloads.

## Weaknesses

1. Applicability to general OT settings.

The formulation appears to rely on the squared Euclidean structure of the cost function, which allows the cost matrix to be expressed in a form compatible with the proposed streaming computation. It would be helpful to clarify how broadly the method applies to OT problems where the cost matrix does not admit such a decomposition.

2. Limited discussion of baseline implementations.

While the reported speedups are impressive, it would be helpful to better understand the baseline implementations used in the comparisons. In particular, clarifying whether highly optimized GPU OT libraries (e.g., KeOps-based or GeomLoss implementations) are included would strengthen the empirical evaluation.

3. Performance across problem scales.

The experiments primarily focus on relatively large problem sizes where memory bandwidth is likely the dominant bottleneck. Additional results on smaller problem sizes could help illustrate the regimes in which FlashSinkhorn provides the most benefit.

---

> ### Author Rebuttal · Authors · 2026-03-27
>
> We thank Reviewer nWPn for the careful reading.
>
> FlashSinkhorn rewrites each stabilized Sinkhorn iteration as a biased dot-product LogSumExp reduction, the same structure as transformer attention, and fuses it into streaming Triton kernels that never materialize the $n \times m$ cost matrix. This yields up to $32\times$ forward and $161\times$ end-to-end speedups over existing online baselines (Table 3), with $O((n{+}m)d)$ memory. We compare against GeomLoss (tensorized and online/KeOps) and OTT-JAX, all using the same squared Euclidean cost, $\varepsilon$, and iteration budget. In a new low-$\varepsilon$ experiment, these speedups are preserved across $\varepsilon \in \{0.1, 0.05, 0.01\}$ with stable fp32 precision (see response to Reviewer kMJp for full tables).
>
> **Generality of cost structure.** We agree this is a scope limitation and will state it more explicitly. Our streaming reduction requires a cost of the form $C_{ij} = \alpha_i + \beta_j - Q_i^\top K_j$ with precomputable $\alpha_i, \beta_j$ and explicit features $Q, K$; the Sinkhorn update then becomes a biased dot-product LSE that can be streamed tile-by-tile without materializing the $n \times m$ cost matrix. Squared Euclidean is the primary instance ($Q=\sqrt{2}X$, $K=\sqrt{2}Y$). Cosine distance also fits this form: on L2-normalized inputs, $\|\mathbf{x}-\mathbf{y}\|^2 = 2(1 - \mathbf{x}\cdot\mathbf{y})$, so cosine distance reduces to half squared Euclidean with adjusted $\varepsilon$. The current implementation instantiates squared Euclidean plus an additive OTDD label-cost term (Section 5). Costs without this structure (Euclidean distance $\|\mathbf{x}-\mathbf{y}\|$, learned neural costs) are future work.
>
> **Baseline implementations.** We compare against the official GeomLoss backends, tensorized (stores the full cost/kernel matrix, quadratic memory) and online (computes costs on the fly via KeOps map-reduce), and OTT-JAX Sinkhorn on PointCloud geometry in log-space (`lse_mode=True`), with costs recomputed on the fly by the geometry. Table 3 reports speedups across all baselines: 9–32$\times$ over KeOps (forward), up to 161$\times$ end-to-end at $d=512$, and up to 5.1$\times$ over OTT-JAX. Tensorized OOMs beyond $n \approx 20\text{k}$ in our A100-80GB setup. Figure 2 shows scaling curves across $n$ and $d$.
>
> **Performance on smaller problem sizes.** Figure 2 (top row) shows timing vs $n$ at $d=64$; FlashSinkhorn's advantage grows with $n$ in the memory-bound regime. At small $n$, tensorized baselines can be faster because they precompute and cache the dense cost matrix. The appendix speedup tables include $d$-sweeps showing the crossover point at each dimension. This regime dependence mirrors FlashAttention vs standard attention.
>
> **Numerical stability.** Stability across streamed tiles is maintained by the online max-subtraction mechanism (Algorithm 1, lines 10–13): each tile computes a local row-max, the running global max rescales previously accumulated sumexp statistics, and the final LSE is computed from the converged statistics. This uses the standard online LSE recurrence from FlashAttention. Our new low-$\varepsilon$ experiments confirm stability: fp32 vs fp64 relative error is $4.02 \times 10^{-5}$ at $\varepsilon=0.1$ and $7.69 \times 10^{-4}$ at $\varepsilon=0.01$ (see response to Reviewer kMJp for full tables). For HVP, the Schur complement conditioning at small $\varepsilon$ is a known structural challenge [Li et al. 2025b]; our default Tikhonov regularization ($\tau=10^{-5}$) achieves <1% HVP error even at $\varepsilon=0.01$.

---

> > ### Author Rebuttal · Reviewer_nWPn · 2026-04-03
> >
> > Thanks for the clear and well-organized rebuttal. The rebuttal address my main concerns. I am satisfied with the responses and am raising my score.

---

> > > ### Author Response · Authors · 2026-04-05
> > >
> > > We thank Reviewer nWPn for the thoughtful engagement and for reconsidering the score. We appreciate that the baseline clarifications,   numerical stability discussion, and cost-structure framing were helpful. We will sharpen the scope and baseline descriptions in the   revision as discussed.

---

### Official Review · Reviewer_HGA9 · 2026-03-13

**Soundness:** 3
**Presentation:** 3
**Significance:** 3
**Originality:** 3
**Overall Recommendation:** 4
**Confidence:** 2

**Summary:**

This paper proposes FlashSinkhorn, an IO-aware GPU implementation of entropic optimal transport that reformulates Sinkhorn updates into attention-style streaming LogSumExp reductions. Based on this reformulation, the method uses fused Triton kernels to reduce HBM traffic and accelerate forward, backward, and HVP computations.

**Compliance With Llm Reviewing Policy:**

Affirmed.

**Final Justification:**

Overall, this paper provides impressive efficiency improvements for entropic optimal transport.
The limited applicability to cost functions is partially addressed, as the authors acknowledge it and leave it for future work.

So I make the overall recommendation as: Weak accept.

**Key Questions For Authors:**

The reported performance improvements rely heavily on fused Triton kernels and system-level optimizations. I could not find a statement regarding whether the implementation will be released. Do the authors plan to open-source the code to facilitate reproducibility and adoption?

**Limitations:**

No. See weaknesses.

**Strengths And Weaknesses:**

Strengths

1. Clear systems insight. The paper observes that stabilized Sinkhorn updates with squared Euclidean cost can be rewritten as biased dot-product LogSumExp reductions, which enables a FlashAttention-style streaming implementation.

2. Good efficiency improvements. Experiments report large speedups over KeOps and tensorized baselines.

3. Evaluation beyond microbenchmarks. The paper evaluates downstream workloads such as OTDD and shuffled regression, showing that the proposed kernel can benefit practical OT pipelines.

Weaknesses

1. Limited applicability to cost functions. The proposed reformulation relies on the squared Euclidean cost, which enables the decomposition  and the resulting attention-style LogSumExp reductions. However, many optimal transport applications use other cost functions such as Euclidean distance, cosine distance, or learned neural costs. It is unclear whether the proposed IO-aware reformulation can extend to these more general cost settings.

2. Performance gains over strong online baselines are not uniform across regimes. The paper’s overall efficiency results are strong, but the advantage over the strongest online baseline (OTT-JAX) is relatively modest in several main-table settings. The appendix further notes that at very large feature dimension (e.g., d=1024), the tensorized baseline can even be 0.5 - 0.7x faster.

3. Component-level ablation could be more systematic. The paper includes useful internal analyses (e.g., symmetric vs. alternating Sinkhorn variants and HVP-specific evaluations), but it does not provide a clean end-to-end component ablation isolating the contribution of the main ingredients, such as: (1) the attention-style reformulation, (2) the fused streaming Sinkhorn kernel, (3) the streamed transport operator, and (4) the streamed HVP implementation.

---

> ### Author Rebuttal · Authors · 2026-03-28
>
> We thank Reviewer HGA9 for the positive assessment.
>
> **Code release.** We plan to release the full implementation as an open-source package upon acceptance, including all kernels, tests, and benchmark scripts.
>
> **Limited applicability to cost functions.** We agree this is a scope limitation and will state it more explicitly. Our streaming reduction requires costs of the form $C_{ij} = \alpha_i + \beta_j - Q_i^\top K_j$ with precomputable per-point terms $\alpha_i, \beta_j$ and explicit features $Q, K$. The Sinkhorn update then becomes a biased dot-product LSE that can be streamed tile-by-tile without materializing the $n \times m$ cost matrix. Squared Euclidean is the primary instance. Cosine distance also fits: on L2-normalized inputs, $\|\mathbf{x}-\mathbf{y}\|^2 = 2(1-\mathbf{x}\cdot\mathbf{y})$, so cosine distance reduces to half squared Euclidean with adjusted $\varepsilon$. Our current implementation instantiates squared Euclidean plus an additive OTDD label-cost term (Section 5). Costs without this structure, Euclidean distance $\|\mathbf{x}-\mathbf{y}\|$ (the square root breaks the affine dot-product form) and learned neural costs are future work.
>
> **Non-uniform speedups over OTT-JAX.** We agree the speedup is regime-dependent. Our new low-$\varepsilon$ experiment confirms $\sim 4\times$ over OTT-JAX at $n=10\text{k}$, $d=64$, stable across $\varepsilon$ (full table in our response to Reviewer kMJp). At small $n$ or very high $d$, other baselines can be competitive. As analyzed in the appendix, OTT-JAX benefits from cuBLAS GEMM efficiency at large $d$ (the $XY^\top$ matmul reaches near-peak TFLOPS), and tensorized can be faster at $d=1024$ when the dense matrix fits in memory (though it uses 39–71$\times$ more memory and OOMs at $n \ge 30\text{k}$).
>
> **Component-level ablation.** We agree a more systematic ablation would strengthen the presentation. The current paper already contains partial isolations of each ingredient: reformulation $\to$ Proposition 1; fused forward kernel $\to$ Table 2 / NCU profiling; streamed transport/backward $\to$ Figure 2 backward panels; streamed HVP $\to$ Figure 2 HVP panels plus the new low-$\varepsilon$ parity results reported in our response to Reviewer kMJp. We will make this mapping explicit in the revision.

---

> > ### Author Rebuttal · Reviewer_HGA9 · 2026-04-03
> >
> > Thanks to the authors for the detailed rebuttal.
> >
> > The limited applicability to cost functions is partially addressed, as the authors acknowledge it and leave it for future work.
> >
> > The concern regarding non-uniform speedups over OTT-JAX is addressed.
> >
> > The component-level ablation is also clarified, and the explicit mapping provided in the revision makes sense to me.

---

> > > ### Author Response · Authors · 2026-04-05
> > >
> > > We thank Reviewer HGA9 for acknowledging the clarifications on speedup regime dependence and the component-level mapping. Regarding cost   applicability, we agree this is a scope limitation and will state it more explicitly in the revision.

---

### Decision · Program_Chairs · 2026-04-30

**Decision:**

Accept (spotlight)

**Comment:**

This paper proposes an IO-aware GPU implementation of the Sinkhorn algorithm for solving Entropic Optimal Transport problems, similar to the celebrated FlashAttention, which is called FlashSinkhorn. The main strengths of the paper were its elegant unification of ideas from algorithms and systems, leading to strong improvements in efficiency, and its solid experimental evaluation. In broad strokes, the primary insight is to rewrite the steps of the Sinkhorn algorithm for solving EOT problems using LogSumExp in such a way that they become structurally the same as the normalizations used in attention. This allows the same approach as what has been done in FlashAttention: make fused Triton kernels which stream tiles of the input through on-chip SRAM with online max/sumexp statistics and never materialize the $N \times M$ cost matrix in HBM. This union between algorithm design and hardware results in impressive speedups over the state of the art libraries OTT-JAX and KeOps, which are already extremely performant. It also allows for even larger-dimensional problems that would previously OOM for OTT-JAX.

Initially, the paper had one detracting reviewer (Reviewer nWPn) who recommended weak reject. Other reviewers had concerns initially too, e.g., Reviewer kMJp rightfully objected to the paper only including experiments with the entropic regularization parameter $\epsilon \geq 0.1$, which is relatively high and which can have a significant effect on the convergence rate of the Sinkhorn algorithm. However, the authors provided thorough arguments and new evidence addressing these issues convincingly during the rebuttal period and scores were subsequently raised, with all reviewers reaching a consensus that this paper should be accepted and with potential for significant impact.

Since all reviewers agreed that this is an original work that is worthy of acceptance and interesting to the community with strong potential for impact, I warmly recommend to accept the paper.